# Numerical experiments on vapor diffusion in polar snow and firn and its impact on isotopes using the multi-layer energy balance

## 3 model Crocus in SURFEX V8.0

4

5

12

Alexandra Touzeau<sup>1</sup>, Amaëlle Landais<sup>1</sup>, Samuel Morin<sup>2</sup>, Laurent Arnaud<sup>3</sup>, Ghislain Picard<sup>3</sup>

8 <sup>2</sup>Météo-France - CNRS, CNRM UMR3589, Centre d'Etudes de la Neige, Grenoble, France

<sup>3</sup>IGE, CNRS UMR5183, Université Grenoble Alpes, Grenoble, France

11 *Correspondence to*: Alexandra Touzeau (alexandra.touzeau@uib.no)

Abstract

To evaluate the impact of vapor diffusion onto isotopic composition variations in the snow pits and then in ice cores,

we introduced water isotopes in the detailed snowpack model Crocus. At each step and for each snow layer, 1) the

initial isotopic composition of vapor is taken at equilibrium with solid phase, 2) a kinetic fractionation is applied during

transport, and 3) vapor is condensed or snow is sublimated to compensate deviation to vapor pressure at saturation.

We study the different effects of temperature gradient, compaction, wind compaction and precipitation on the final

vertical isotopic profiles. We also run complete simulations of vapor diffusion along isotopic gradients and of vapor

diffusion driven by temperature gradients at GRIP, Greenland and at Dome C, Antarctica over periods of 1 or 10 years.

The vapor diffusion tends to smooth the original seasonal signal, with an attenuation of 7 % to 12 % of the original

- signal over 10 years at GRIP. This is smaller than the observed attenuation in ice cores, indicating that the model
- attenuation due to diffusion is underestimated or that other processes, such as ventilation, influence attenuation. At
- Dome C, the attenuation is stronger (18 %), probably because of the lower accumulation and stronger  $\delta^{18}$ O gradients.

#### 25 1 Introduction

- The isotopic ratios of oxygen or deuterium measured in ice cores have been used for a long time to reconstruct the
- evolution of temperature over the Quaternary (EPICA comm. members, 2004; Johnsen et al., 1995; Jones et al., 2018;
- Jouzel et al., 2007; Kawamura et al., 2007; Lorius et al., 1985; Petit et al., 1999; Schneider et al., 2006; Stenni et al.,
- 2004; Stenni et al., 2011; Uemura et al., 2012; WAIS-Divide members, 2013). They are however subject to alteration

 <sup>&</sup>lt;sup>1</sup>LSCE, CNRS UMR8212, UVSQ, Université Paris-Saclay, Gif-sur-Yvette, 91191, France

during post-deposition through various processes. Consequently, even if the link between temperature and isotopic 31 composition of the precipitations is quantitatively determined from measurements and modelling studies (Stenni et al., 32 2016; Goursaud et al., 2017), it cannot faithfully be applied to reconstruction of past temperature. Nevertheless, ice 33 cores remain a primary climatic archive for the Southern Hemisphere where continental archives are rare (Mann and 34 Jones, 2003). In Antarctica, where meteorological records only started in the 1950s (Genthon et al., 2013), they provide 35 useful information for understanding climate variability (e.g. EPICA comm. members, 2006; Shaheen et al., 2013; 36 Steig, 2006; Stenni et al., 2011) and recent climate change (e.g. Altnau et al., 2015; Schneider et al., 2006). When 37 using ice cores for past climate reconstruction, other parameters than temperature at condensation influence the isotopic 38 compositions and must be considered. Humidity and temperature in the region of evaporation (Landais et al., 2008; 39 Masson-Delmotte et al., 2011), or the seasonality of precipitation (Delmotte et al., 2000; Sime et al., 2008; Laepple et 40 al., 2011) should be taken into account. In addition, uneven accumulation in time and space introduces stratigraphic 41 noise (Ekaykin et al., 2009). Indeed, records from adjacent snow pits have been shown to be markedly different, under 42 the influence of decameter-scale local effects such as wind redeposition of snow, erosion, compaction, and 43 metamorphism (Ekaykin et al., 2014; Petit et al., 1982). These local effects reduce the signal/noise ratio. Then only 44 stacking a series of records from snow pits can eliminate this local variability and yield information relevant to recent 45 climate variations (Fisher and Koerner, 1994; Hoshina et al., 2014; Ekaykin et al., 2014; Altnau et al., 2015). This 46 concern is particularly significant in central regions of east Antarctica characterized by accumulation rates lower than 47 100 mm water equivalent per year (van de Berg et al., 2006). There, strong winds can scour and erode snow layer over 48 depths larger than the annual accumulation (Frezzotti et al., 2005; Morse et al., 1999; Libois et al., 2014). There is thus 49 a strong need to study post-deposition effects in these cold and dry regions.

Additionally to mechanical reworking of the snow, the isotopic compositions are further modified in the snowpack. First, diffusion along isotopic gradients can occur within the snow grains due to solid diffusion (Ramseier et al., 1967). Second, within the porosity, the vapor isotopic composition can change due to: 1) diffusion along isotopic gradients in gaseous state, 2) thermally induced vapor transport caused by vapor pressure gradients, 3) ventilation in gaseous state, or 4) exchanges between the gas phase and the solid phase i.e. sublimation and condensation. In the porosity, the combination of diffusion along isotopic gradients in the vapor and of exchange between vapor and the solid phase has been suggested to be the main explanation to the smoothing of the isotopic signal in the solid phase (Ebner et al., 2016, 57 2017; Gkinis et al., 2014; Johnsen et al., 2000). The isotopic compositions in the solid phase is also modified by 'dry 58 metamorphism' and 'ventilation' but in a less predictable way. In both cases, the vapor transport exerts an influence 59 on the isotopic compositions in the solid phase because of permanent exchanges between solid and vapor. During 'dry 60 metamorphism' (Colbeck et al., 1983), vapor transport is driven by vapor pressure gradients, themselves caused by 61 temperature gradients. During ventilation (Town et al., 2008), vapor moves as part of the air in the porosity, because 62 of pressure variations at the surface. Last, at the top of the snowpack, the isotopic composition of snow may also be 63 modified through direct exchange with atmospheric vapor (Ritter et al., 2016).

To elucidate the impact of these various post-deposition processes on the snow isotopic compositions, numerical 65 models are powerful tools. They allow one to discriminate between processes and test their impact one at a time. 66 Indeed, Johnsen et al. (2000) were able to simulate and deconvolute the influence of diffusion along isotopic gradients 67 in the vapor at two Greenland ice-core sites, GRIP and NGRIP, using a numerical model. To do this, they define a 68 quantity named 'diffusion length' which is the mean displacement of a water molecule during its residence time in the 69 porosity. Using a thinning model and an equation of diffusivity of the water isotopes in snow, they compute this 70 diffusion length as a function of depth. It is then used to compute the attenuation ratio A/Ao, and in the end retrieve 71 the original amplitude Ao. Additionally, the effect of forced ventilation was investigated by Neumann (2003) and 72 Town et al. (2008) using similar multi-layer numerical models. In these models, wind-driven ventilation forces 73 atmospheric vapor into snow. There, the vapor is condensed especially in layers colder than the atmosphere.

We focus on the movement of water isotopes in the vapor phase in the porosity, in the absence of macroscopic air movement. In that situation, the movement of vapor molecules in the porosity is caused by vapor pressure gradients, or by diffusion along isotopic gradients. Note that in the first case, the vapor transport is 'thermally induced' i.e. the vapor pressure gradients directly result from temperature gradients within the snowpack. Thus, the first prerequisite of our model is to correctly simulate macroscopic energy transfer within the snowpack and energy exchange at the surface.

The transport of vapor molecules will affect the isotopic composition in the solid phase only if exchanges between vapor and solid are also implemented. Thus, the second prerequisite is that the model includes a description of the snow microstructure, and of its evolution in time. Snow microstructure is typically represented by its emerging scalar properties such as density, specific surface area and higher order terms often referred to as "shape parameters" (e.g. Krol and Löwe, 2016). While the concept of "grain" bears ambiguity, it is a widely used term in snow science and
glaciology which we here employ as a surrogate for "elementary microstructure element", without explicit reference
to a formal definition, be it crystallographic or geometrical.

Crocus is a unidimensional multi-layer model of snowpack with a typically centimetric resolution initially dedicated to the numerical simulation of snow in temperate regions (Brun et al., 1992). It describes the evolution of the snow 88 89 microstructure driven by temperature and temperature gradients during dry snow metamorphism, using semi-empirical 90 variables and laws. It has been used for ice-sheets conditions in polar regions, both Greenland and Antarctica (Brun et 91 al., 2011; Lefebre et al., 2003; Fréville et al., 2013; Libois et al., 2014, 2015). In these regions, it gives realistic 92 predictions of density and snow type profiles (Brun et al., 1992; Vionnet et al., 2012), snow temperature profile (Brun 93 et al., 2011) and snow specific surface area and permeability (Carmagnola et al., 2014; Domine et al., 2013). It has 94 been recently optimized for application to conditions prevailing at Dome C, Antarctica (Libois et al., 2014). This was 95 necessary to account for specific conditions such as high snow density values at the surface and low precipitation 96 amounts.

The Crocus model has high vertical spatial resolution and also includes interactive simulation of snow metamorphism 98 in near-surface snow and firn. Therefore, it is a good basis for the study of post-deposition effects in low accumulation 99 regions. For the purpose of this study, we thus implemented vapor transport resulting from temperature gradients and 100 the water isotopes dynamics into the Crocus model. This article presents this double implementation, and a series of 101 sensitivity tests. A perfect match of observations is not anticipated, in part because not all relevant processes are 102 represented in the model. This study represents thus a first step towards better understanding the impact of diffusion 103 driven by temperature gradients on the snow isotopic composition.

#### 104 2 Physical basis

The isotopic composition of the snow can evolve after deposition due to several processes. Here, we first give a brief overview of such processes at the macroscopic level. Section 2.1 thus deals with modification of the isotopic composition of a centimetric/decametric snow layer after exchanges with the other layers. Second, we consider the evolution of the isotopic composition at the microscopic level, i.e. at the level of the microstructure. Indeed, the macroscopic change of the isotopic composition results from both large scale and small-scale processes. For instance, dry metamorphism includes both vapor transport from one layer to another, and vapor/ice grain exchange inside alayer.

#### 112 2.1 Evolution of the snow layers composition at the macroscopic scale

Several studies address the evolution of the isotopic compositions in the snow column after deposition. Here we 114 describe first processes leading only to attenuation of the original amplitude (Sect. 2.1.1). Then we describe processes 115 which lead to other types of signal modifications (Sect. 2.1.2). These modifications result from transportation and 116 accumulation of heavy or light isotopes in some layers without any link to the original isotopic signal. In some cases, 117 the mean  $\delta^{18}$ O value of the snow deposited can also be modified.

#### 118 2.1.1 Signal attenuation on a vertical profile: smoothing

In this Section we consider processes leading only to attenuation of the original amplitude of the  $\delta^{18}$ O signal by 120 smoothing. We define the mean local pluriannual value as the average isotopic composition in the precipitation taken 121 over 10 years. The smoothing processes, which act only on signal variability, do not modify this average value. Within 122 the snow layers, the smoothing of isotopic compositions is caused by diffusion along isotopic gradients in vapor phase 123 and in solid phase. The magnitude of smoothing depends on site temperature, and on accumulation. Indeed, higher 124 temperatures correspond to higher vapor concentrations, and higher diffusivities in the vapor and solid phases. Oppositely, high accumulation rates ensure a greater separation between seasonal  $\delta^{18}$ O peaks (Ekaykin et al., 2009; 125 Johnsen et al., 1977) thereby limiting the impact of diffusion. They also result in increased densification rates, and 126 127 therefore reduced diffusivities (Gkinis et al., 2014). Because sites with high accumulation rates also usually have higher 128 temperatures, the resulting effect on diffusion is still unclear. These two competing effects should be thoroughly 129 investigated and Johnsen et al. (2000) displays the damping amplitude of a periodic signal depending on wavelength 130 and on diffusion length, strongly driven by temperature.

In Greenland, Johnsen et al. (1977) indicate that annual cycles generally disappear at depths shallower than 100 m for sites with accumulation lower than 200 kg m<sup>-2</sup> yr<sup>-1</sup>. Diffusion along isotopic gradients exists throughout the entire snow/ice column. It occurs mainly in the vapor phase in the firn, especially in the upper layers with larger porosities. After pore closure, it takes place mostly in the solid phase, at a much slower rate. Note that in the solid phase, all isotopes have the same diffusion coefficient.

#### 136 2.1.2 Signal shift caused by processes leading to oriented vapor transport

We consider here the oriented movement of water molecules forced by external variables such as temperature or 137 138 pressure. We use the term 'oriented' here to describe an overall movement of water molecules that is different from 139 their molecular agitation, and externally forced. Three processes can contribute to oriented vapor transport and hence 140 possible isotopic modification within the snowpack: diffusion, convection, and ventilation (Albert et al., 2002). Brun 141 et Touvier (1987) have demonstrated that convection of dry air within the snow occurs only in case of very low snow 142 density of the order of  $\sim 100 \text{ kg/m}^3$ . These conditions are generally not encountered in Antarctic snow and therefore 143 convection is not considered here. Bartelt et al. (2004) also indicate that energy transfer by advection is negligible 144 compared to energy transfer by conduction in the first meters of the snowpack. The two other processes, ventilation 145 and diffusion are forced respectively by variations of the surface pressure and surface temperature. In the first case, 146 the interaction between wind and surface roughness is responsible for wind-pumping, i.e. renewal of the air of the 147 porosity through macroscopic air movement (Albert et al., 2002; Colbeck, 1989, Neumann et al., 2004). In the second 148 case, air temperature diurnal or seasonal variations generate vertical temperature gradients within the snow (Albert and McGilvary, 1992; Colbeck, 1983). They result into vertical vapor pressure gradients, responsible for vapor diffusion. 149 150 These two processes are largely exclusive (Town et al., 2008) because strong ventilation homogeneize the air and vapor 151 in the porosity and therefore prevents diffusion. Diffusion as a result of temperature gradients can coexist with 152 ventilation only at very low air velocities (Calonne et al., 2015). It becomes the main process of vapor transport when 153 air is stagnant in the porosity. During diffusion, lighter molecules move more quickly in the porosity, leading to a 154 kinetic fractionation of the various isotopologues.

#### 155

#### 2.2 Evolution of the isotopic composition at the microscopic scale

#### 156 2.2.1 Conceptual representation of snow microstructure as spherical grains

The term "snow grain" as used classically is an approximation. In reality, 'snow grains' are very diverse in size, shape, degree of metamorphism and may also be made of several snow crystals agglomerated. Moreover, they are often connected to each other, forming an ice matrix, or 'snow microstructure'. However, several studies addressing snow metamorphism physical processes have relied on spherical ice elements to represent snow grains and snow microstructure (Legagneux and Domine, 2005; Flanner and Zender, 2006). Here, we consider that the snow grains are 162 made of two concentric layers, one internal and one external, with different isotopic compositions. In terms of snow 163 microstructure, this could correspond to inner vs. outer regions of the snow microstructure.

Indeed, the snow grain or microstructure is not necessarily homogeneous in terms of isotopic composition. On the one 165 hand, the central part of the grain or of the microstructure is relatively insulated. This central part becomes even more 166 insulated as the grain grows, or as the structure gets coarser. On the other hand, outer layers are not necessarily formed 167 at the same time as the central part, or in the same environment (Lu and DePaolo, 2016). They are prone to subsequent 168 sublimation or condensation of water molecules, implying that their composition varies more frequently than for the inner layers. Of course, only the bulk  $\delta^{18}$ O value of the snow grain can be measured by mass spectrometry. But 169 170 considering the heterogeneity of the grain may be required to get a fine understanding of the processes. In the following, 171 we propose to split the ice grain compartment into two sub-compartments: grain surface and grain center. Thus, the 172 grain surface isotopic composition evolves because of exchange with two compartments: 1) with vapor in the porosity 173 through sublimation or condensation, and 2) with grain center through solid diffusion, or grain center translation. The 174 grain center composition evolves at the time scale of week/month, as opposed to the grain surface, where the 175 composition changes at the time scale of the vapor diffusion, i.e. over minutes.

#### 176 2.2.2 Solid diffusion within snow grains

177The grain center isotopic composition may change either as a result of crystal growth/sublimation or as a result of178solid diffusion within the grain. For solid diffusion, water molecules move in the crystal lattice through a vacancy179mechanism, in a process of self-diffusion that has no particular direction, and that is very slow. The diffusivity of180water molecules in solid ice  $D_{ice}$  in m<sup>2</sup>·s<sup>-1</sup> follows Arrhenius law. Thus, it can be expressed as a function of ice

temperature T (Gkinis et al., 2014; Johnsen et al., 2000; Ramseier, 1967) using Eq. (1):

$$D_{ice} = 9.2 \cdot 10^{-4} \times \exp\left(\frac{-7186}{T}\right)$$
 (1)

where symbols are listed in Table 1.

Thus at 230 K, the diffusivity is  $2.5 \times 10^{-17} \text{ m}^2 \cdot \text{s}^{-1}$ . Gay et al. (2002) indicate that in the first meter at Dome C, a typical snow grain has a radius of 0.1 mm. Across this typical snow grain, the characteristic time for diffusion is given by Eq. (2):

$$\Delta t_{sol} = \frac{R_{moy}^2}{D_{ice}} = 4.03 \times 10^8 \, s, \, \text{or} \sim 13 \, \text{years}$$
 (2)

Therefore, the solid diffusion within the grain is close to zero at the time scales considered in the model. For Dome C, if we use the average temperature T of 248 K for the summer months (Dec. to Jan., Table 2), the characteristic time becomes 15 months. Thus, within a summer period, the snow grain is only partially refreshed through this process. At Summit the grain size is typically larger, from 0.2 to 0.25 mm in wind-blown and wind pack and from 0.5 to 2 mm in the depth hoar layer (Albert and Shultz, 2002). The summer temperature is also higher, with an average value T of 259 K at Summit from July to Sept, after Shuman et al. (2001). Using a grain size of 0.25 mm, the resulting characteristic time is of the order of 30 months.

#### 195 2.2.3 Snow grain recrystallization

During snow metamorphism, the number of snow grains tends to decrease with time, while the snow grain size tends 197 to increase (Colbeck, 1983). Indeed, each grain experiences continuous recycling through sublimation/condensation, 198 but the small grains are more likely to disappear completely. Then, there is no more nucleus for condensation at the 199 grain initial position. Oppositely, the bigger grains do not disappear and accumulate the vapor released by the smaller 200 ones. Concurrently to this change of grain size, the grain shape also tends to evolve. In conditions of maintained/stable 201 temperature gradient, facets appear at the condensing end of snow grains, while the sublimating end becomes rounded 202 (Colbeck, 1983). In that case, the center of the grain moves toward the warm air region. This migration causes a renewal 203 of the grain center, on a proportion that can be estimated from the apparent grain displacement (Pinzer et al., 2012). 204 Pinzer et al. (2012) use this method to obtain an estimation of vapor fluxes.

The asymmetric recrystallization of snow grains implies that the surface layer of the snow grain is eroded at one end and buried at the other end. Therefore, the composition of the grain center changes more often than if the surface layer was thickening through condensation or thinning through sublimation homogeneously over the grain surface. This means that the 'inner core' of the grain gets exposed more often. Implementing this process is thus very important to have a real-time evolution of the snow grain center isotopic composition. Here, we reverse the method of Pinzer et al. (2012). Therefore, we use the fluxes of isotopes in vapor phase computed by the model to assess the renewal of the grain center (Sect. 3.1.3.).

#### 212 3 Material and Methods

#### 213 3.1 Description of the model SURFEX/Crocus V8.0

We first present the model structure and second describe the new module of vapor transport (diffusion forced by

temperature gradients). Third, we present the integration of water isotopes in the model.

#### 216 3.1.1 Model structure

The Crocus model is a one-dimensional detailed snowpack model, consisting of a series of snow layers with variable and evolving thicknesses. Each layer is characterized by its density, heat content, and by parameters describing snow microstructure such as sphericity and specific surface area (Vionnet et al., 2012, Carmagnola et al., 2014). In the model, the profile of temperature evolves with time in response to 1) surface temperature and 2) energy fluxes at the surface and at the base of the snowpack. To correctly compute energy balance, the model integrates albedo calculation, deduced from surface microstructure and impurity content (Brun et al., 1992, Vionnet et al., 2012).

The successive components of the Crocus model have been described by Vionnet et al. (2012). Here we only list them to describe those modified to include water stable isotopes and water vapor transfer. Note that the Crocus model has a typical internal time step of 900 s (15 min), corresponding to the update frequency of layers properties. We only refer here to processes occurring in dry snow.

1) Snow fall: The presence/absence of precipitation at a given time is determined from the atmospheric forcing
inputs. When there is precipitation, a new layer of snow may be formed. Its thickness is deduced from the precipitation
amount.

2) Update of snow layering: At each step, the model may split one layer into two or merge two layers together
to get closer to a target vertical profile for optimal calculations. This target profile has high resolution in the first layers
to correctly simulate heat and matter exchanges. The layers that are merged together are the closest in terms of
microstructure variables.

3) Metamorphism: The microstructure variables evolution follows empirical laws. These laws describe the
change of grain parameters as a function of temperature, temperature gradient, snow density and liquid water content.
4) Snow compaction: Layer thickness decreases, and layer density increases under the burden of the overlying
layers and resulting from metamorphism. In the original module, snow viscosity is parameterized using the layer

density and also using information on the presence of hoar or liquid water. However, this parameterization of the 239 viscosity was designed for alpine snowpack (Vionnet et al., 2012) and may not be adapted to polar snow packs. 240 Moreover, since we are considering only the first 12 m of the snowpack in the present simulations, the compaction in 241 the considered layers does not compensate the yearly accumulation, leading to rising snow level with time. To maintain 242 a stable surface level in our simulations, we used a simplified compaction scheme, where the compaction rate  $\varepsilon$  is the 243 same for all the layers. The compaction rate is obtained by dividing the accumulation rate at the site (see Sect. 3.3) by 244 the total mass of the snow column (Eq. 3). It is then applied to all layers to obtain the density change per time step 245 using Eq. 4.

$$\varepsilon = \frac{dm_{sn}}{dt} / \sum_{1}^{nmax} (\rho_{sn}(t,n) \times dz(t,n))$$
(3)

$$\frac{\rho_{sn}(t+dt,n)-\rho_{sn}(t,n)}{dt} = \varepsilon \times \rho_{sn}(t,n)$$
(4)

Wind drift events: They modify the properties of the snow grains which tend to become more rounded. They
also increase the density of the first layers through compaction. An option allows snow to be partially sublimated
during these wind drift events (Vionnet et al., 2012).

6) Snow albedo and transmission of solar radiation: In the first 3 cm of snow, snow albedo and absorption coefficient are computed from snow microstructure properties and impurity content. The average albedo value in the first 3 cm is used to determine the part of incoming solar radiation reflected at the surface. The rest of the radiation penetrates into the snowpack. Then, the absorption coefficient is used to describe the rate of decay of the radiation as it is progressively absorbed by the layers downward, following an exponential law.

7) Latent and sensible surface energy and mass fluxes: The sensible heat flux and the latent heat flux are257 computed using the aerodynamic resistance and the turbulent exchange coefficients.

8) Vertical snow temperature profile: It is deduced from the heat diffusion equation, using the snow conductivity,as well as the energy balance at the top and at the bottom of the snowpack.

Snow sublimation and condensation at the surface: The amount of snow sublimated/condensed is deduced
from the latent heat flux, and the thickness of the first layer is updated. Other properties of the first layer such as density
and SSA are kept constant.

#### 263 3.1.2 Implementation of water transfer

The new vapor transport subroutine has been inserted after the compaction (4) and wind drift (5) modules, and before the solar radiation module (6). In this section, the term 'interface' is used for the horizontal surface of exchange between two consecutive layers. The flux of vapor at the interface between two layers is obtained using the Fick's law of diffusion (Eq. (5)):

$$F(n+1 \to n) = \frac{-2 D_{eff}(t,n \to n+1)(C_v(t,n) - C_v(t,n+1))}{dz(t,n) + dz(t,n+1)}$$
(5)

where dz(t, n) and dz(t, n+1) are the thicknesses of the two layers considered in meters,  $C_v(t, n)$  and  $C_v(t, n+1)$  are the local vapor mass concentrations in the two layers in kg m<sup>-3</sup>, and  $D_{eff}(t, n \rightarrow n + 1)$  in m<sup>2</sup> s<sup>-1</sup> is the effective diffusivity of water vapor in the snow at the interface. The thicknesses are known from the previous steps of the Crocus model, but the vapor mass concentrations and the interfacial diffusivities must be computed.

The effective diffusivity at the interface is obtained in two steps: first the effective diffusivities (Deff(t,n) and Deff(t,n+1)) 274 in each layer are calculated (Eq. (6)), second, the interfacial diffusivity  $(D_{eff}(t, n \rightarrow n + 1))$  is computed as their 275 harmonic mean (Eq. (7)). Effective diffusivity can be expressed as a function of the snow density using the relationship 276 proposed by Calonne et al. (2014), for layers with relatively low density. In these circumstances, the compaction occurs 277 by 'boundary sliding', meaning that the grains slide on each other, but that their shape is not modified. It is therefore 278 applicable to our study where density is always below 600 kg m<sup>-3</sup>. The equation of Calonne et al. (2014) is based on 279 the numerical analysis of 3D tomographic images of different types of snow. It relates normalized effective diffusivity 280  $D_{eff}/D_v$  to the snow density  $\rho_{sn}$  in the layer (Eq. (6)).  $D_v$  is the vapor diffusivity in air and has a value that varies 281 depending on the air pressure and air temperature (Eq. (19) in Johnsen et al., 2000). pice corresponds to the density of 282 ice and has a value of 917 kg m<sup>-3</sup>.

$$\frac{D_{eff}(t,n)}{D_{v}} = \frac{3}{2} \left( 1 - \frac{\rho_{sn}(t,n)}{\rho_{ice}} \right) - \frac{1}{2}$$
(6)

$$D_{eff}(t,n \to n+1) = \frac{1}{\frac{1}{D_{eff}(t,n)^+} \frac{1}{D_{eff}(t,n+1)}}$$
 (7)

We assume that vapor is in general at saturation in the snow layers (Neumann et al., 2008; Neumann et al., 2009). The local mass concentration of vapor  $C_{\nu}$  in kg m<sup>-3</sup> in each layer is given by the Clausius-Clapeyron equation (Eq. (8)):

$$C_{\nu}(t,n) = C_{\nu 0} \exp\left(\frac{L_{sub}}{R_{\nu}\rho_{ice}}\left(\frac{1}{T_0} - \frac{1}{T(t,n)}\right)\right)$$
 (8)

where  $C_{\nu 0}$  is the mass concentration of vapor at 273.16 K and is equal to 2.173 10<sup>-3</sup> kg m<sup>-3</sup>,  $L_{sub}$  is the latent heat of sublimation and has a value of 2.6 10<sup>9</sup> J m<sup>-3</sup>,  $R_v$  is the vapor constant and has a value of 462 J kg<sup>-1</sup> K<sup>-1</sup>,  $\rho_{ice}$  is the density of ice and has a value of 917 kg m<sup>-3</sup>,  $T_0$  is the temperature of the triple point of water and is equal to 273.16 K and T is the temperature of the layer.

All layers are treated identically, except the first layer at the top and the last layer at the bottom. For the uppermost layer, the exchange of vapor occurs only at the bottom boundary. Indeed, exchanges with the atmosphere are described elsewhere in Crocus at step 9 where surface energy balance is realized. For the lowermost layer, only exchanges taking place at the top boundary are considered, the flux of vapor to/from the underlying medium being set to zero.

For each layer, the mass concentration of vapor in air and effective diffusivity are computed within the layer and in the 297 neighboring layers. Fluxes at the top and bottom of each layer are deduced from Fick's law of diffusion (Eq. (5)). They 298 are integrated over the subroutine time step, and the new mass of the layer is computed. It is used at the beginning of 299 the next subroutine step. We use a 1 s time step within the subroutine, smaller than the main routine time step of 900 s. This ensures that vapor fluxes remain small relative to the amount of vapor present in the layers. Note that the 300 temperature profile, which controls the vapor pressure profile, is not modified within the subroutine. Physically, 301 temperature values should change as a result of the transfer of sensible heat from one layer to another associated with 302 303 vapor transport. They should also evolve due to the loss or gain of heat caused by water sublimation or condensation 304 (Albert and McGilvary, 1992; Kaempfer et al., 2005). However, vapor transport is only a small component to heat 305 transfer between layers (Albert and Hardy, 1995; Albert and McGilvary, 1992). In the absence of ventilation, with or 306 without vapor diffusion, the steady-state profile for temperature varies by less than 2% (Calonne et al., 2014). Thus, 307 the effect can be neglected at first order.

#### **308 3.1.3 Implementation of water isotopes**

In the model, the isotopic composition of snow in each layer is represented by the triplicate ( $\delta^{18}$ O, d-excess, <sup>17</sup>O-310 excess). Only the results of  $\delta^{18}$ O are presented and discussed here. For each parameter, two values per layer are 311 considered independently, corresponding to the 'snow grain center' and the 'snow grain surface', respectively. Water 312 vapor isotopic composition is deduced at each step from the 'snow grain surface' isotopic composition. It is not stored 313 independently to limit the number of prognostic variables. The isotopic compositions are used at step 1, i.e. for

snowfall, and after step 5, within the new module of vapor transfer.

In the snow fall subroutine, a new layer of snow may be added, depending on the weather, at the top of the snowpack.

At this step of the routine, the snow grains being deposited are supposed to be homogenous, i.e. they have the same

composition in the "grain surface" compartment and in the "grain center" compartment. Their composition is deducedfrom the air temperature (see Sect. 3.2).

Within the vapor transport subroutine, a specific module deals with the isotopic aspects of vapor transport. It modifies 320 the isotopic compositions in the two snow grain sub-compartments as a result of water vapor transport and 321 recrystallization of snow crystals. It works with four main steps:

1) an initiation step where the vapor isotopic compositions are computed, using equilibrium fractionation, from the323 ones in the grain surface sub-compartment,

2) a transport step where vapor moves from one layer to another, with a kinetic fractionation associated with diffusion,

3) a balance step where the new vapor in the porosity exchanges with the grain surface compartment by

sublimation/condensation. The flux is determined by the difference between actual vapor mass concentration and

expected vapor mass concentration at saturation,

and 4) a 'recrystallization' step where the grain center and grain surface isotopic compositions are homogenized,

- leading to an evolution of grain center isotopic composition.
- The time step in this module is 1s, the same as the time step of the sub-routine.
- The initial vapor isotope composition  $R_{vap ini}^{i}$  in a given layer is taken at equilibrium with the 'grain surface' isotopic 331 composition R<sup>i</sup><sub>surf ini</sub>. Here i denotes heavy isotope, and thus stands for <sup>18</sup>O, <sup>17</sup>O or D. Equilibrium fractionation is a 332 333 hypothesis that is correct in layers where vapor has reached equilibrium with ice grains, physically and chemically. 334 This process is limited by the water vapor - snow mass transfer whose associated speed is of the order of 0.09 m.s-1 335 (Albert and McGilvary, 1992). In our case, we are dealing with centimetric scale layers thickness and recalculate the 336 isotopic composition every second so that we consider that the speed of the mass transfer is not limiting the equilibrium 337 situation at the water vapor - snow interface. To compute isotopic ratios for water vapor we use the following Eq. (9) 338 and (10):

$$339 \quad \begin{cases} R_{vap \, ini}^{18} = \alpha_{sub}^{18} \times R_{surf \, ini}^{18} \\ R_{vap \, ini}^{17} = \alpha_{sub}^{17} \times R_{surf \, ini}^{17} \\ R_{vap \, ini}^{18} + R_{vap \, ini}^{17} + 1 = 1/c_{vap \, ini}^{16} \end{cases}$$
(9)

$$\begin{cases}
R_{vap ini}^{D} = \alpha_{sub}^{D} \times R_{surf ini}^{D} \\
R_{vap ini}^{D} + 1 = 1/c_{vap ini}^{1H}
\end{cases} \tag{10}$$

- The equilibrium fractionation coefficients ( $\alpha_{sub}^i$ ) are obtained using the temperature-based parameterization from 342 Ellehoj et al. (2013). Note that we make a slight approximation here, by replacing molar concentrations by mass 343 concentrations in our mass balance formulas (see Table 1 for symbol definitions).
- The initial vapor mass concentration in air  $C_v$  has already been computed in the vapor transport subroutine, and the volume of the porosity can be obtained from the snow density  $\rho_{sn}$  and the thickness of the layer dz. By combining both, we obtain Eq. (11) which gives the initial mass of vapor in the layer  $m_{vap ini}$ .

$$m_{vap ini} = C_v \times \left(1 - \frac{\rho_{sn}}{\rho_{ice}}\right) \times dz$$
(11)

This mass of vapor should be subtracted from the initial grain surface mass because vapor mass is not tracked outside
of the sub-routine (Fig. 1). The new grain surface isotope composition, after vapor individualization is given by Eq.
(12):

$$c_{surf new}^{18} = \frac{m_{surf new}^{18}}{m_{surf new}} = \frac{m_{surf ini}^{18} - m_{vap ini} \times c_{vap ini}^{18}}{m_{surf ini} - m_{vap ini}}$$
 (12)

The diffusion of isotopes follows the same scheme as the water vapor diffusion described above in Sect. 3.1.2. and Eq. (5). In Eq. (13), the gradient of vapor mass concentrations is replaced by a gradient of concentration of the studied isotopologue. The kinetic fractionation during the diffusion is realized with the  $D^{i}/D$  term where i stands for <sup>18</sup>O or <sup>17</sup>O or <sup>2</sup>H (Barkan and Luz, 2007).

$$F^{18}(n+1 \to n) = \frac{-2 \times D_{eff}(t,n \to n+1) \left( C_{\nu}(t,n) \times c_{\nu ap \ ini}^{18}(t,n) - C_{\nu}(t,n+1) \times c_{\nu ap \ ini}^{18}(t,n+1) \right)}{dz(t,n) + dz(t,n+1)} \times \frac{D^{18}}{D}$$
(13)

As done for water molecules transport (Sect. 3.1.2.), the flux is set to zero at the top of the first layer and at the bottom of the last layer. When the vapor concentration is the same in two adjacent layers, the total flux of vapor is null. But diffusion along isotopic gradients still occurs if the isotopic gradients are non-zero (Eq. (13)). Once top and bottom fluxes of each layer have been computed, the new masses of the various isotopes in the vapor are deduced, as well as the new ratios. After the exchanges between layers, the isotopic composition in the vapor has changed. However, the vapor isotopic composition is not a prognostic variable outside of the vapor transport subroutine. To record this change, it must be transferred to either the 'grain surface compartment' or to the 'grain center compartment' before leaving the subroutine. First, we consider exchanges of isotopes with the grain surface compartment, which is in direct contact with the vapor. Depending on the net mass balance of the layer, two situations must be considered:

1) If the mass balance is positive, condensation occurs, so that the transfer of isotopes takes place from the vapor toward the grain surface. To evaluate the change in the isotope composition in the grain surface, the mass of vapor condensed  $\Delta m_{vap,exc}$  must be computed. It is the difference between the mass of vapor expected at saturation and the mass of vapor present in the porosity after vapor transport. Note that temperature does not evolve in this sub-routine. Nevertheless, the difference is not exactly equal to the mass of vapor that has entered the layer, because of layer porosity change. The excess mass of vapor is given by Eq. (14):

$$373 \qquad \Delta m_{vap,exc} = \left[ (\rho_{sn\,new} - \rho_{sn\,ini}) + C_v \times \left[ \left( 1 - \frac{\rho_{sn\,ini}}{\rho_{ice}} \right) - \left( 1 - \frac{\rho_{sn\,new}}{\rho_{ice}} \right) \right] \right] \times dz \tag{14}$$

Since the excess of vapor is positive, the next step is the condensation of the excess vapor. The number of excess water 375 molecules is determined through comparison with the expected number in the water vapor phase for equilibrium state 376 between surface snow and water vapor. Here the condensation of excess vapor occurs without additional fractionation 377 because (1) there is a permanent isotopic equilibrium between surface snow and interstitial vapor restored at each first 378 step of the sub-routine and (2) kinetic fractionation associated with diffusion is taken into account during diffusion of 379 the different isotopic species along the isotopic gradients.

2) If the mass balance is negative, the transfer of isotopes takes place from the grain surface toward the vapor 381 without fractionation. Ice from the grain surface sub-compartment is sublimated without fractionation to reach the 382 expected vapor concentration at saturation. Note that the absence of fractionation at sublimation is a frequent 383 hypothesis because water molecules move very slowly in ice lattice (Friedman et al., 1991; Neumann et al., 2005; 384 Ramseier, 1967). Consequently, the sublimation removes all the water molecules present at the surface of grains, 385 including the heaviest ones before accessing inner levels. In reality, there are evidences for fractionation at sublimation. 386 It occurs through kinetic effects associated with sublimation / simultaneous condensation, or during equilibrium 387 fractionation at the boundary, especially when invoking the existence of a thin liquid layer at the snow - air interface 388 (Neumann et al., 2008 and references therein; Sokratov and Golubev, 2009; Stichler et al., 2001; Ritter et al., 2016).
The new composition in the vapor results from a mixing between the vapor present and the new vapor recently
produced. The composition in the 'grain surface' ice compartment does not change.

The limit between the surface compartment and the grain center compartment is defined by the mass ratio of the grain surface compartment to the total grain mass i.e.  $\tau = m_{surf}/(m_{center} + m_{surf})$ , (Fig. 1). This mass ratio can be used to 392 393 determine the thickness of the 'grain surface layer' as a fraction of grain radius, for spherical grains. The surface 394 compartment must be thin, to be able to react to very small changes in mass when vapor is sublimated or condensed. 395 Our model has a numerical precision of 6 decimals and is run at a 1 s temporal resolution. Consequently, the isotopic 396 composition of the surface compartment can change in response to surface fluxes only if its mass is smaller than  $10^6$ times the mass of the water vapor present in the porosity. This constrains the maximum value for  $\tau$ :  $m_{surf} < 10^6 m_{vap}$ , or 397  $m_{surf}/(m_{center} + m_{surf}) < 10^6 \frac{\Phi \cdot \rho_{\nu} \cdot V_{tot}}{\rho_{sn} \cdot V_{tot}}$ , i.e.  $\tau < \frac{\rho_{\nu} \cdot \Phi}{\rho_{sn}} \cdot 10^6$ . Considering typical temperatures, snow densities and layer 398 399 thicknesses (Table 3) we obtain a maximum value of  $3.3 \cdot 10^{-2}$ . On the other hand, this compartment must be thick 400 enough to transmit the change in isotopic compositions caused by vapor transport and condensation/sublimation to the 401 grain center. Again, numerical precision imposes that its mass should be no less than  $10^{-6}$  times the mass of the grain 402 center compartment, and thus we get an additional constraint:  $\tau > 10^{-6}$ . Here we use a ratio  $\tau = 5 \cdot 10^{-4}$  for the mass of the 403 grain surface relative to the total mass of the layer (Fig. 1). We have run sensitivity tests with smaller and larger ratios 404 (Sect. 4.3).

Two types of mixing between grain surface and grain center are implemented in the model. The first one is associated 406 with crystal growth or shrinkage, because of vapor transfer. Mixing is performed at the end of the vapor transfer 407 subroutine, after sublimation/condensation has occurred. During the exchange of water between vapor and grain 408 surface, the excess or default of mass in the water vapor caused by vapor transport has been entirely transferred to the 409 grain surface sub-compartment. Thus, the mass ratio between the grain surface compartment and the grain center compartment deviates from the original one. To bring the ratio  $\tau$  back to normal value of 5.10<sup>4</sup>, mass is transferred 410 411 either from the grain surface to the grain center or from the grain center to the grain surface. This happens without 412 fractionation, i.e. if the transfer occurs from the center to the surface, the composition of the center remains constant. 413 The second type of mixing implemented is the grain center translation (Pinzer et al., 2012) which favors mixing 414 between grain center and grain surface in the case of sustained temperature gradient. Pinzer et al. (2012) used the apparent grain displacement to compute vapor fluxes. Here, we reverse this method and use the vapor fluxes computed from Fick's law to estimate the grain center renewal. We could transfer a small proportion of the surface compartment to the grain center every second. Instead, we choose to totally mix the snow grain every few days. The interval  $\Delta t_{surf/center}$ between two successive mixings is derived from the vapor flux F(n+1→n) within the layer using Eq. (15).

$$\Delta t_{surf/center} = \frac{m_{sn} \times \tau}{F(n+1 \to n)}$$
(15)

The average temperature gradient of 3 °C m<sup>-1</sup> corresponds to a flux  $F(n+1 \rightarrow n)$  of 1.3 10<sup>-9</sup> kg·m<sup>-2·</sup>s<sup>-1</sup>. The typical mass for the layer m<sub>sn</sub> is 3.3 kg. Based on these values, the dilution of the grain surface compartment into the grain center should occur every 15 days. Of course, this is only an average, since layers have varying masses, and since the temperature gradient can be larger or smaller. We will however apply this time constant for all the layers and any temperature gradient (see sensitivity tests Sect. 4.3), to ensure that the mixing between compartments occurs at the same time in all layers.

In terms of magnitude, this process is probably much more efficient for mixing the solid grain than grain growth or 427 solid diffusion. It is thus crucial for the modification of the bulk isotopic composition of the snow layer. It makes the 428 link between microscopic processes and macroscopic results.

#### 429 3.1.4 Model initialization

For model initialization, an initial snowpack is defined, with a fixed number of snow layers, and for each snow layer 431 an initial value of thickness, density, temperature and  $\delta^{18}$ O. Typically, processes of oriented vapor transport such as 432 thermally induced diffusion and ventilation occur mainly in the first meters of snow. Therefore, the model starts with 433 an initial snowpack of about 12 m.

The choice of the layer thicknesses depends on the annual accumulation. Because the accumulation is much higher at GRIP than at Dome C (Sect. 3.2., Table 2), the second site is used to define the layer thicknesses. About 10 cm of fresh snow are deposited every year (Genthon et al., 2016; Landais et al., 2017). This implies that to keep seasonal information, at least one point every 4 cm is required in the first meter. For the initial profile, we impose maximal thickness of 2 cm for the layers between 0 and 70 cm depth and 4 cm for the layers between 70 cm and 2 meters depth. As the simulation runs, merging is allowed but restricted in the first meter to a maximum thickness of 2.5 cm. Below 2 meters, the thicknesses are set to 40 cm or even 80 cm. Thus, the diffusion process can only be studied in the first 2 m of the model snowpack. In the very first centimeters of the snowpack, thin millimetric layers are used to accommodate low precipitation amounts and surface energy balance. The initial density profiles are defined for each site specifically (see Sect. 3.2). The initial temperature and  $\delta^{18}$ O profiles in the snowpack depend on the simulation considered (see Sect. 3.3).

#### 445 3.1.5 Model output

A data file containing the spatio-temporal evolution of prognostic variables such as temperature, density, SSA or  $\delta^{18}$ O 447 is produced for each simulation. Here, we present the results for each variable as two-dimensional graphs, with time 448 on the horizontal axis and snow height on the vertical axis. The variations of the considered variable are displayed as 449 color levels. The white color corresponds to an absence of change of the variable. As indicated above, only the first 12 450 m of the polar snowpack are included in the model. The bottom of this initial snowpack constitutes the vertical 451 reference or 'zero' to measure vertical heights h. The height of the top of the snowpack varies with time due to snow 452 accumulation and to snow compaction. In the text, we sometimes refer to the layer depth z instead of its height h. The 453 depth can be computed at any time by subtracting the current height of the considered layer from the current height of 454 the top of the snowpack.

#### 455 **3.2 Studied sites: meteorology and snowpack description**

In this study we run the model under conditions encountered at Dome C, Antarctica and GRIP, Greenland. We chose these two sites because they have been well-studied in the recent years through field campaigns and numerical experiments. In particular for Dome C, a large amount of meteorological and isotopic data is available (Casado et al., 2016a; Stenni et al., 2016; Touzeau et al., 2016). Typical values of the main climatic parameters for the two studied sites, GRIP and Dome C, are given in Table 2, as well as typical  $\delta^{18}$ O range. Dome C has lower accumulation rates of 2.7 cm ice equivalent per year (i.e. yr.<sup>-1</sup>) compared to GRIP rates of 23 cm i.e. yr.<sup>-1</sup> (Table 2), making it more susceptible to be affected by post-deposition processes.

In this study, we also compare the results obtained for GRIP to results from two other Greenland sites, namely NGRIP

- and NEEM. GRIP is located at the ice-sheet summit, whereas the two other sites are located further north, in lower
- elevation areas with higher accumulation rates. In detail, NGRIP is located 316 km to the NNW of GRIP ice-drill site

(Dahl-Jensen et al., 1997). GRIP and NGRIP have similar temperatures of -31.6 °C and -31.5 °C but different
accumulation rates of 23 cm i.e. yr.<sup>-1</sup> and 19.5 cm i.e. yr.<sup>-1</sup> respectively. NEEM ice-core site is located some 365 km to
the NNW of NGRIP on the same ice-ridge. It has an average temperature of -22 °C and an accumulation rate of 22 cm
i.e. yr.<sup>-1</sup>.

The  $\delta^{18}$ O value in the precipitation at a given site reflects the entire history of the air mass, including evaporation, transport, distillation, and possible changes in trajectory and sources. However, assuming that these processes are more or less repeatable from one year to the next, it is possible to empirically relate the  $\delta^{18}$ O to the local temperature, using measurements from collected samples. Here, using data from one-year snowfall sampling at Dome C (Stenni et al., 2016; Touzeau et al., 2016), we use the following Eq. (16) to link  $\delta^{18}$ O in the snowfall to the local temperature  $T_{air}$ , in K:

$$\delta^{18}O_{sf} = 0.45 \times (T_{air} - 273.15) - 31.5$$
 (16)

We do not provide an equivalent expression for GRIP, Greenland, because the simulations run here (see Sect. 3.1.1)do not include precipitation.

The initial density profile in the snowpack is obtained from fitting density measurements from Greenland and
Antarctica (Bréant et al., 2017). Over the first 12 m of snow, we obtain the following evolution (Eq. (17) and Eq. (18))
for GRIP and Dome C respectively:

$$\rho_{sn}(t,n) = 17.2 \cdot z(t=0,n) + 310.3 \quad (N=22; R^2=0.95)$$
 (17)

$$\rho_{sn}(t,n) = 12.41 \times z(t=0,n) + 311.28 \text{ (N=293; R^2=0.50)}$$
 (18)

#### 484 **3.3 List of simulations**

Table 4 presents the model configuration for the 6 simulations considered here.

#### 486 3.3.1 Greenland simulations

The first simulation, listed as number 1 in Table 4, is dedicated to the study of diffusion along isotopic gradients. It is

realized on a Greenland snowpack with an initial sinusoidal profile of  $\delta^{18}$ O (see Eq. 19) and with a uniform and constant

- vertical temperature profile at 241 K. In addition to comparison to  $\delta^{18}$ O profiles for GRIP and other Greenland sites,
- the aim of the first simulation is to compare results from Crocus model to the models of Johnsen et al. (2000) and
- Bolzan and Pohjola (2000) run at this site with only diffusion along isotopic profiles. To compare our results to theirs,

we consider an isothermal snowpack, without meteorological forcing, and we deactivate modules of surface exchanges

and heat transfer. The initial seasonal sinusoidal profile at GRIP is set using Eq. (19):

$$\delta^{18}O(t,n) = -35.5 - 8 \times \sin\left(\frac{2\pi \times z(t,n)}{a \times \rho_{ice}/\rho_{sn}(t,n)}\right)$$
 (19)

where z is the depth of the layer n,  $\rho_{sn}$  is its density,  $\rho_{ice}$  is the density of ice and has a value of 917 kg m<sup>-3</sup>, *a* is the average accumulation at GRIP and is equal to 0.23 m i.e./yr (Dahl-Jensen et al., 1993). The peak to peak amplitude value of 16 ‰ is close to the back-diffused amplitude at Summit (Sjolte et al., 2011).

The second simulation is run with evolving temperature in the snowpack. The snow temperature is computed by the model, using meteorological forcing from ERA-Interim (see Table 4). In that case, the transport of isotopes in the vapor phase results both from diffusion along isotopic gradients and from vapor concentration gradients. The initial snowpack is the same as in the previous simulation.

In the two GRIP simulations, the modules of wind compaction and weight compaction are inactive. Indeed, as weight compaction is taken to compensate yearly accumulation (Eq. (3) and (4)), applying this compaction without precipitation would lead to an unrealistic drop in snow level. The wind compaction was absent from the model of Johnsen et al. (2000) and using this module would make comparisons more difficult.

#### 506 **3.3.2 Dome C simulations**

In simulations 3 to 6, we take advantage of the high documentation of the Dome C site to disentangle the different 508 effects on the variations of water isotopic composition. All the simulations at Dome C were performed with an evolving 509 temperature profile. Temperatures in the snow layers were computed using a modified meteorological forcing from ERA-Interim (Dee et al., 2011; Libois et al., 2014; see details in Table 4), as well as the modules of energy exchange 510 and transfer. In this series of simulations, the  $\delta^{18}$ O values thus evolve as a result of diverging and/or alternating vapor 511 512 fluxes. The simulations are ordered by increasing complexity. First, in Simulation 3, the modules of homogeneous compaction and wind drift are deactivated, as well as the module of snowfall. Thus, the impact of vapor transport 513 514 forced by temperature gradients on the snow isotopic compositions is clearly visible. Then, in Simulation 4, the module 515 of compaction and the module of wind drift are activated, to see their impact on the isotopes. We use an accumulation rate dm<sub>sn</sub>/dt for Dome C of 0.001 kg m<sup>-2</sup> per 15 min (see Eq. (3)). Next, in Simulation 5, snowfall is added, to assess 516 how new layers affect snow  $\delta^{18}$ O values. Lastly, in Simulation 6, the model is run over 10 years at Dome C, to build 517

up a snowpack with realistic 'sinusoidal' variation in  $\delta^{18}$ O values.

processes are active.

**4 Results** 

4.1 Greenland

#### 521 4.1.1 Results of the Crocus simulations (Simulations 1 and 2) 522 Figure 2 shows the result of Simulation 1, where only diffusion along isotopic gradients is active, as in Johnsen et al., 523 2000. As expected the peak to peak amplitude of $\delta^{18}$ O cycles is reduced because of diffusion. Over 10 years, from 2000 524 to 2009, the amplitude decreases by 1.2 ‰ which corresponds to a 7.3 % variation. 525 Figure 3 shows the result of Simulation 2, i.e. with varying temperature in the snowpack. The attenuation is stronger 526 than the one observed in the previous simulation. The minima at 11.46 m increases by 1.03 ‰ over ten years, and the 527 maxima at 11.15 m decreases by 0.84 \. Thus, the total attenuation is ~1.9 \. or 11.7 % for this height range. Below, 528 the attenuation is smaller, with a total attenuation of only 6 % for heights between 10.54 and 10.85 m. If we compare 529 attenuation for heights 11.46 and 11.56 m in the 1<sup>st</sup> and 2<sup>nd</sup> simulation, we note that including temperature gradients 530 leads to an increased attenuation by 50 %. Between 11.46 m and 11.56 m, the $\delta^{18}O_{gcenter}$ values increase over ten years by 1 to 4 ‰. This increase is not caused 531 only by attenuation of the original sinusoidal signal. Indeed, at h=11.60 m, the values get higher than the initial maxima 532 533 which was -36 ‰ at 11.64 m. There is therefore a local accumulation of heavy isotopes in this layer as a result of vapor transport. This maximum corresponds to a local maximum in temperature and is coherent with departure of <sup>18</sup>O-534 535 depleted water vapor from this layer. Thus, thermally induced vapor transport does not only result into signal attenuation, but can also shift the $\delta^{18}$ O value, regardless of the initial sinusoidal variations. 536 537 Lastly, in the first 2-3 cm of the snowpack, strong depletion is observed over the period, with a decrease by 2 to 3 ‰ 538 instead of 0.5 ‰ when the temperature gradients were absent (Simulation 1). This depletion probably results from 539 arrival of <sup>18</sup>O-depleted water vapor from warmer layers below. This shows again the influence of temperature gradients 540 which were absent from the previous simulation. However, note that in this simulation we neglect precipitation and 541 exchange of vapor with the atmosphere. Thus, the depletion observed here may not occur in natural settings when these

543 In conclusion, at GRIP, the diffusion of vapor as a result of temperature gradients has a double impact on isotopic

compositions. It increases the attenuation in the first 60 cm of snow, because of higher vapor fluxes. And it also creates 545 local isotopic maxima and minima, in a pattern corresponding to temperature gradients in the snowpack but 546 disconnected from the original  $\delta^{18}$ O sinusoidal signal.

#### 547 4.1.2 Comparison with core data

Here, we evaluate the attenuation of the initial seasonal signal in  $\delta^{18}$ O over 10 years at 2 Greenland ice-core sites, 548 549 NEEM and GRIP. For the first site, we use 4 shallow cores (NEEM2010S2, NEEM2008S3, NEEM2007S3, 550 NEEM2008S2) published in Steen-Larsen et al., 2011 and in Masson-Delmotte et al., 2015. For the second site, we 551 use one shallow core (1989-S1), published in White et al., 1997. For the GRIP core, only the first 80 meters are 552 considered. Therefore, the data presented corresponds to deposition and densification conditions like the modern ones. 553 For NEEM the values of the four cores are taken together. For NEEM and GRIP, the semi-amplitude is computed along 554 the core. In the first 10 meters, the maximum value every 30 cm is retained, and deeper in the firm, because of 555 compaction, the maximum value every 20 cm is retained (see also Supp. Material; Fig. 4). For this study, we have 556 chosen to estimate attenuation on years with a clearly marked seasonal cycle, a strategy that can be debated but at least 557 documented. Consequently, from this first series of maxima, a second series of maxima is computed, with a larger 558 window of 1 meter. The 'attenuated amplitudes' at each level is then defined as the ratio between these 1-meter maxima 559 and the initial 1-meter maxima. Maximum semi-amplitudes every 5 m are also computed and displayed on Figure 4. 560 The 2.5 m attenuation is slightly higher at GRIP, leading to a remaining amplitude of 86 %, than at NEEM where the 561 remaining amplitude is 90 % (Fig. 4). The amplitude decreases with depth in parallel for the two cores, with the 562 amplitude at NEEM staying always higher than at GRIP. For comparison with our model, we estimate attenuation after 563 10 years, i.e. at a depth of ~5.8 m for NEEM and ~5.65 m for GRIP. The remaining amplitude is 80 % and 72 % at 564 GRIP and NEEM respectively. Our Simulation 1 produced 7 % of attenuation only on the same duration, showing that 565 our model, run on an isothermal snowpack, underestimates the attenuation observed in the data.

#### 566 4.1.3 Comparison with other models

At 2.5 m at NGRIP, Johnsen et al. (2000) simulate remaining amplitude of 77 % (Fig. 4). For a depth of 5.43 m,
corresponding to an age of 10 years, the simulated remaining amplitude is 57 %. For Bolzan and Pohjola (2000), at
GRIP after 10 years, 70 % of the initial amplitude is still preserved. The slower attenuation for Bolzan and Pohjola

(2000) compared to Johnsen et al. (2000) may be due more to the different sites considered than to the different models.
Indeed, GRIP has higher accumulation rates that should limit diffusion. Nevertheless, the attenuation of 30 % simulated
by Bolzan and Pohjola at GRIP is stronger than the attenuation of 7 % simulated in our model. Town et al. (2008, Sect.
[31]) found attenuations of a few tenth of per mil after several years when implementing only diffusion, a result
consistent with ours since we get a decrease by 1.2 ‰ after 10 years.

We explore below the reasons for discrepancies between models. The equation for effective diffusivity of vapor in firm used in our study is different from the ones used by Johnsen et al. (2000) or by Bolzan and Pohjola (2000). Indeed, we do not consider the tortuosity factor *l*, nor the adjustable scale factor *s* of Bolzan and Pohjola. However, using the values given by the previous authors for *l* and *s* lead to  $D_{eff}$  values ranging from 6.7 10<sup>-6</sup> to 9.9 10<sup>-6</sup> m<sup>2</sup> s<sup>-1</sup> for a density of 350 kg m<sup>-3</sup> and a temperature of 241 K which is coherent with our value of 8.7 10<sup>-6</sup> m<sup>2</sup> s<sup>-1</sup>. As indicated by Bolzan and Pohjola (2000), the choice of one equation or another has little impact here.

The most probable difference lie in the way diffusion is taken into account. Johnsen et al. (2000) and Bolzan and 582 Pohjola (2000) use a single equation of diffusion to predict the evolution of the isotopic composition of the layer. In 583 our case, we specifically compute the fluxes in the vapor each second and at each depth level and deduce the evolution of  $\delta^{18}$ O in the grain center, after sublimation/condensation and recrystallization. Denux (1996) and van der Wel et al. 584 (2015) indicate that the model developed by Johnsen (1977) and used in Johnsen et al. (2000) overestimates the 585 attenuation compared to observed values. For Denux (1996), the model of Johnsen (1977) should consider the presence 586 587 of ice crusts, and maybe also the temperature gradients in the surface snow, to get closer to the real attenuation at 588 remote Antarctic sites. Van der Wel et al. (2015) have compared the model results to a spike-layer experiment realized 589 at Summit. Because an artificial snow layer cannot be representative of natural diffusion, they took care to evaluate 590 diffusion based only on the natural layers present above and below the artificial layer. van der Wel et al. (2015) propose 591 three causes to the discrepancy between Johnsen et al.'s model prediction and actual measured attenuation at GRIP. 592 They blame either ice crusts, or a bad knowledge and parametrization of the tortuosity in the first meters of snow, 593 and/or a bad description of the isotopic heterogeneity within the ice grain. In our model, the grain heterogeneity is 594 included. Even if the parameters defining the mixing between the two compartments are not very well constrained (see 595 Sect. 4.3), the attenuation is indeed smaller compared to the Johnsen's model.

#### 596 4.2 Dome C (Antarctica)

The aim of the Simulations 3 to 6, run at Dome C, is to isolate diffusion from other effects affecting water isotopic598 composition, i.e. wind-drift and compaction.

#### 599 4.2.1 Simulation 3: without precipitation, without wind drift, and without homogeneous compaction

Figure 5 presents the results of temperature evolution (a and b) and  $\delta^{18}O$  evolution (c and d) for Simulation 3. The main changes of  $\delta^{18}O_{gsurf}$  and  $\delta^{18}O_{gcenter}$  occur in summer (Fig. 5c and 5d). On the one hand, the first 20 cm of snow tend to become <sup>18</sup>O-enriched by +0.2 ‰ for the grain center compartment. On the other hand, the first centimeter becomes depleted by 1.0 ‰ for grain center. This pattern is coherent with the temperature profiles for the summer period (Fig. 5a). Indeed, vapor moves out of the warmest layers and toward colder layers where it condensates. This causes an increase in  $\delta^{18}O$  in warm layers and a decrease in colder layers. This pattern is also confirmed by snow density changes (see Fig. S2).

During winter, the temperature generally decreases toward the surface (Fig. 5a). Vapor transport is thus reversed in the 608 first 20 cm, but this only slightly reduces the dispersion of  $\delta^{18}O_{gcenter}$  values. On the first of August, the temperature at 609 the surface temporarily increases to 235 K. This warm event strongly modifies the temperature profile in the snowpack, 610 and therefore the pattern of vapor transport. It is associated with an increase of  $\delta^{18}O$  values at the surface, which is 611 particularly visible for the  $\delta^{18}O_{gsurf}$  values (Fig. 5c).

Thus, vapor transport can modify  $\delta^{18}$ O values in surface snow, even in the absence of precipitation or condensation 613 from the atmosphere. This mechanism could explain the parallel evolution of surface snow isotopic composition and 614 temperature described by Steen-Larsen et al. (2014) and Touzeau et al. (2016) between precipitation events.

#### 615 4.2.2 Simulation 4: without precipitation, with wind drift, and with homogeneous compaction

Compaction and wind drift are not supposed to modify directly the  $\delta^{18}$ O values. However, the change in densities and 617 layer thicknesses modifies slightly the temperature profile and the diffusivities. These processes thus could have an 618 indirect impact on  $\delta^{18}$ O values. Figure 6 shows  $\delta^{18}O_{gcenter}$  changes that are reduced compared to the simulation without 619 wind drift and compaction. This is coherent with a decrease in the density changes associated with vapor transport in 620 the case with compaction (see Fig. S5).

#### 621 4.2.3 Simulation 5: with precipitation, with wind drift, with homogeneous compaction

In Simulation 5, we add precipitation to wind and weight compaction effects. Both snowfall and wind compaction are responsible for irregular changes, respectively positive and negative, of the height of the snowpack (Fig. 7). In the new deposited layers, the  $\delta^{18}O_{gcenter}$  values reflect the  $\delta^{18}O$  values in the precipitation. They vary as expected from -40 % on the 31<sup>th</sup> of December to -59 ‰ in July (Fig. 7, Fig. S8). The effect of vapor transport is visible only in 'old' layers which were originally homogeneous in terms of  $\delta^{18}O$ . These old layers, which were reaching the surface in January, have been buried below the new layers and are found from 11 cm depth downward in December.

#### 628 4.2.4 Simulations 6: Ten-years simulation at Dome C

Simulations 6 corresponds to a simulation run over 10 years at Dome C, with variable  $\delta^{18}$ O in the precipitation. Over 629 these 10 years, about 1 m of snow is deposited. At the end of the simulation, the vertical profile of  $\delta^{18}$ O in the new 630 631 layers has an average value of -49.7 ‰, and a semi-amplitude of 4.5 ‰ (Fig. 8). Here we take into account all the 632 maxima and minima at a vertical resolution of 9 cm of fresh snow. Based on the atmospheric temperature variations 633 only, the isotopic composition in the precipitation should vary around an average value of -53.2 ‰, with a semiamplitude of 8.6 ‰. The main reason for this difference is the precipitation amounts: large precipitation events in 634 winter are associated with relatively high  $\delta^{18}$ O values. The vertical resolution chosen for the model of 2.5 cm may also 635 636 contribute to the decrease of the semi-amplitude. Indeed, light snowfall events do not result in the production of a new 637 surface layer but are integrated into the old surface layer. As expected, the peak to peak amplitude of  $\delta^{18}$ O variations 638 is then further reduced as a result of the two vapor diffusion processes and of associated vapor/solid exchanges. The 639 effect of vapor transport is relatively small. To help its visualization, we selected four layers and displayed the evolution 640 of  $\delta^{18}$ O in these layers over the years (Fig. 8d). The selected layers were deposited during winter 2000, and during 641 summer seasons 2002, 2004, and 2006.

For the layer deposited during winter 2000, there is an increase in  $\delta^{18}$ O values of about +0.8 ‰ over ten years. The slope is irregular, with the strongest increases occurring during summers, between November and February, when vapor transport is maximal. The slope is also stronger when the layer is still close to the surface, probably because of the stronger temperature gradients in the first centimeters of snow (Fig. 8a). For the layers deposited during the summers, the evolution of  $\delta^{18}$ O values is symmetric to the one observed for winter 2000. Over 10 years, i.e. between

2000 and 2009, the  $\delta^{18}$ O amplitude thus decreases by about 1.6 %. This corresponds to a decrease of 18 % relative to 647 the initial amplitude in the snow layers. This is higher than the 7 % attenuation modelled in Greenland for constant 648 temperature, and to the 11.7 % attenuation observed when including diffusion caused by temperature gradients (Sect. 649 650 4.1). However, the comparison between the two sites is not straightforward, because of differences in temperature and 651 accumulation counteracting each other. On the one hand, at GRIP, the diffusion is forced by low vertical gradients of  $\delta^{18}$ O of the order of 0.24 ‰ cm<sup>-1</sup>. These are much smaller than the typical  $\delta^{18}$ O gradients at Dome C which are close 652 653 to 1.10 ‰ cm<sup>-1</sup>. On the other hand, the temperature of 241 K at GRIP is higher than the 220 K measured at Dome C, 654 thus favoring diffusion.

#### 655 4.3 Sensitivity tests for duration of recrystallization

We have shown above that attenuation of the isotopic signal seems too small at least for the GRIP site. In parallel, the parameters  $\tau$  and  $\Delta t_{gsurf/center}$  of the model associated to grain renewal could only loosely be estimated leading to uncertainty in the attenuation modeling. In this section, we perform some sensitivity tests to quantify how  $\delta^{18}O$ attenuation can be increased by exploring the uncertainty range on the renewal of the snow grain. Indeed, the assumed values for the ratio between grain surface and the total mass of the grain  $\tau$  may have been under or over-estimated. The same is true for the periodicity of mixing between these two compartments  $\Delta t_{surf/center}$ .

The sensitivity tests are first designed for Greenland sites, run for 6 months, with initial amplitude of the sinusoidal  $\delta^{18}$ O signal of 16 ‰, and a fixed temperature of 241 K in all the layers (Fig. 9). First, we use a periodicity of mixing  $\Delta t_{surf/center}$  of 15 days and vary the value for the mass ratio  $\tau$ : 1·10<sup>-6</sup>, 5·10<sup>-4</sup>, 3.3·10<sup>-2</sup>. In practice, for  $\Delta t_{surf/center}$ =15 days, we realize mixing on the second and 16<sup>th</sup> of each month. Second, we use the usual value of 5 10<sup>-4</sup> for  $\tau$  and change the periodicity of the mixing to 2 days.

- In the first case, where  $\tau = 1 \ 10^{-6}$ , and the mixing occurs every 15 days, the grain surface compartment is very 668 small. Its original sinusoidal  $\delta^{18}$ O profile disappears in less than one day due to exchanges with vapor (not shown). 669 The impact on grain center is then very small with an increase of the first minimum by ~1.0  $10^{-4}$  ‰ over 6 months 670 (Fig. 9a). In this case, the attenuation due to diffusion is even reduced compared to the results displayed above.
- In the second case, where  $\tau = 5 \ 10^4$ , and the mixing occurs every 15 days, the grain surface compartment is 672 larger, and the attenuation is slower. Thus, in the grain surface compartment, half of the original amplitude still remains

at the end of the simulation (not shown). The impact on the grain center compartment is clearly visible with an increase of the first minimum by of  $2.2 \ 10^{-2}$  % after 6 months (Fig. 9b).

- In the third case, with  $\tau$ = 3.3 10<sup>-2</sup>, and mixing every 15 days, the attenuation of the sinusoidal signal in the 676 grain surface compartment is only of 1 % because the grain surface compartment is very large. On opposite, attenuation 677 in the grain center is quite large, i.e. the first minimum increases by 4.0 10<sup>-2</sup> ‰ after 6 months (Fig. 9c).

- In the fourth case, with  $\tau = 5 \ 10^{-4}$  and mixing every 2 days, the first minimum increases by 4.1  $10^{-2}$  ‰ after 6 679 months for the grain center compartment (Fig. 9d). It is similar to the attenuation observed in the third case.

The results of these sensitivity tests suggest that the impact of vapor transfer on the grain center isotopic compositions is maximized when the grain surface compartment is large and/or refreshed often. They also show clearly that using a small grain surface compartment such as  $\tau = 1 \ 10^{-6}$  drastically reduces the impact on the grain center isotopic values. However, our best estimates for  $\tau$  and  $\Delta t_{surf/center}$  were not chosen randomly (see Sect. 3.1.3). Moreover, the use of  $\tau = 3.3$  $10^{-2}$  or  $\Delta t_{surf/center} = 2$  days leads to a near doubling of the  $\delta^{18}$ O attenuation (see above). This is not yet sufficient to explain the gap between our model output for isothermal simulation and the data. However, if this doubling is applicable to the case with temperature gradients, the attenuation obtained might reach the one observed in the data at GRIP.

At Dome C, sensitivity tests show that we can increase the attenuation by a factor of 3 by reducing the mixing time from 15 to 2 days (Figure 10b-c). Similarly, if the ratio  $\tau$  is put at 3.3 10<sup>-2</sup> instead of 5 10<sup>-4</sup>, attenuation is more than doubled over 3 years (Figure 10d-e). Thus, at Dome C, the values of  $\tau$  and  $\Delta t_{surf/center}$  seem to affect more strongly the attenuation obtained compared to GRIP. This greater sensitivity at Dome C could result from the influence of temperature gradients, as well as from steeper  $\delta^{18}$ O gradients caused by the low accumulation. Indeed, the average layer thickness of 2 cm in the first meter corresponds to ~4 points per year at Dome C, but 35 points per year at GRIP.

#### 693 4.4 Additional missing processes

In the previous sections, we have seen that model outputs for GRIP generally lead to smaller attenuation than the one observed in ice-cores. To improve the model compatibility with data, two kinds of approaches are possible. On the one hand, it would be useful to realize simulations adapted to on-site experiments such as the one by van der Wel et al. (2015). This would allow verifying how diffusion can be improved in the model. For instance, previous studies have suggested that water vapor diffusivity within the snow porosity may be underestimated by a factor of 5 (Colbeck, 699 1983), but this is debated (Calonne et al., 2014). On the other hand, we also believe that other processes should probably 700 be considered to explain the remaining attenuation. Ventilation is an additional process that has already been implemented in the snow water isotopic model of Town et al. (2008) and Neumann (2003). Because of strong porosity 701 702 and sensitivity to surface wind and relief, ventilation is probably as important as diffusion in the top of the firn, even 703 if diffusion is expected to be more effective at greater depths. Indeed, for the Dome C simulation (Fig. 8), the slope 704  $d(\delta^{18}O)/dz$  decreases slowly, indicating that diffusion remains almost as active at 60 centimeters than at 10 centimeters 705 depth. Neumann (2003) indicates that at Taylor Mouth the diffusion becomes the only process of vapor transport below 2 meters depth. For Dome C, for a temperature gradient of 3 °C m<sup>-1</sup>, we compute an average speed due to diffusion of 706 3 10<sup>-6</sup> m s<sup>-1</sup>. This is comparable to air speed due to wind pumping of about 3 10<sup>-6</sup> m s<sup>-1</sup> within the top meters of snow 707 at WAIS (Buizert and Severinghaus, 2016). We conclude that, in as much as these results can be applied to Dome C, 708 709 the two processes would have comparable impact at this site in the first meters of snow. The next step for Crocus-iso 710 development is thus to implement ventilation. Finally, we are also aware that in Antarctic central regions, the wind 711 reworking of the snow has a strong effect in shaping the isotopic signal. A combination of stratigraphic noise and 712 diffusion could indeed be responsible for creating isotopic cycles of non-climatic origin in the firn (Laepple et al., 713 2017). Wind reworking may also contribute to attenuation, by mixing together several layers deposited during different 714 seasons.

#### 715 5. Conclusions and perspectives

Water vapor transport and water isotopes have been implemented in the Crocus snow model enabling depicting the 717 temporal  $\delta^{18}$ O variations in the top 50 cm of the snow in response to new precipitation, evolution of temperature 718 gradient in the snow and densification. The main process implemented here to explain post-deposition isotopic 719 variations is diffusion. We have implemented two types of diffusion in vapor phase: 1) water vapor diffusion along 720 isotopic gradients, and 2) thermally induced vapor diffusion. The vapor diffusion between layers was realized at the 721 centimetric scale. The consequences of the two vapor diffusion processes on isotopes in the solid phase were 722 investigated. The solid phase was modelled as snow grains divided in two sub-compartments: (1) a grain surface subcompartment in equilibrium with interstitial water vapor and (2) an inner grain only exchanging slowly with the surface 723 724 compartment. We parameterized the speed of diffusion through the renewal time of a snow grain and proportion of the two snow grain compartments.

Our approach based on a detailed snow model makes it possible to investigate at fine scale various processes explaining 727 the variations of density and  $\delta^{18}$ O in the firn. We look specifically at the effect of evolution of the temperature gradient, new snow accumulation and compaction event linked to wind drift. Over the first 30 cm, the snow density variations 728 729 are mainly driven by compaction events linked to wind drift. Vapor transport and long-term compaction have secondary 730 effects. Below 30 cm, wind drift driven compaction is no more visible. Because of strong temperature gradient and 731 low density, water vapor transport will have a significant effect down to 60 cm.  $\delta^{18}$ O is primary driven by variations in  $\delta^{18}O$  of precipitation as expected. The seasonal variations are then attenuated by water vapor transport and diffusion 732 733 along isotopic gradients, with an increase of these effects at higher temperatures i.e. during summer periods.

From 10 years simulations of the Crocus-iso model both at GRIP Greenland and Dome C Antarctica, we have estimated the post-deposition attenuation of the annual  $\delta^{18}$ O signal in the snow to about 7-18 % through diffusion. This attenuation is smaller than the one obtained from isotopic data on shallow cores in Greenland suggesting missing processes in the Crocus model when implementing water vapor. It is also significantly smaller than the diffusion implemented by Johnsen et al. (2000) but some studies have suggested that the Johnsen isotopic diffusivity is too strong (Denux, 1996; Van der Wel et al., 2015).

We see our study as a first step toward a complete post-deposition modelling of water isotopes variations. Indeed, 741 several other developments are foreseen in this model. First, wind pumping is currently not implemented in the Crocus 742 model. This effect, implemented in the approach of Neumann (2003) and Town et al. (2008) is expected to have a 743 contribution as large as the effect of diffusion for the post-deposition isotopic variations. Second, in low accumulation 744 sites like Dome C, wind scouring has probably an important effect on the evolution of the  $\delta^{18}$ O signal in depth through 745 a reworking of the top snow layers (Libois et al., 2014). This effect has not been considered here but could be 746 implemented in the model in the next years. It could also play a role in the preservation of anomalously strong  $\delta^{18}$ O 747 peaks at Dome C (Denux, 1996).

Other short-term developments concern the implementation of the exchange of water vapor with the atmosphere 749 through hoar deposition. This is particularly timely since many recent studies have studied the parallel evolution of 750 isotopic composition of water vapor and surface snow during summer both in Greenland and Antarctica (Steen-Larsen et al., 2014; Ritter et al., 2016; Casado et al., 2016a; 2016b). Similarly, implementation of ventilation of the snowpack
in the model since this effect is expected to significantly participate to signal attenuation.

Another aspect is to look at the post-deposition d-excess and <sup>17</sup>O-excess variations in snow pits. Indeed, recent studies have shown that the relationship between <sup>17</sup>O-excess and  $\delta^{18}$ O is not the same when looking at precipitation samples and snow pits samples in East Antarctica (Touzeau et al, 2016). This observation questions the influence of diffusion within the snowpack on second order parameters such as <sup>17</sup>O-excess. Indeed, <sup>17</sup>O-excess is strongly influenced by kinetic diffusion driven fractionation which may be quantified by the implementation of <sup>17</sup>O-excess in our Crocus-iso model.

#### 760 Code availability

The code used in the manuscript is a development of the open source code for SURFEX/ISBA-Crocus model based on version V8.0, hosted on an open git repository at CNRM (<u>https://opensource.umr-cnrm.fr/projects/surfex\_git2</u>). Before downloading the code, you must register as a user at <u>https://opensource.umr-cnrm.fr/</u>. You can then obtain the code used in the present study by downloading the revision tagged 'Touzeau\_jan2018' of the branch touzeau\_dev (last access: January 2018). The meteorological forcing required to perform the runs is available as a supplement.

#### 767 Author contribution

S. Morin wrote the new module of vapor diffusion. A. Touzeau inserted isotopes and isotope transport into the numerical code with help of S. Morin for numerical issues and physics and help from A. Landais for concepts and hypotheses of the theory of isotopes. G. Picard and L. Arnaud provided information and references on snow microstructure and microphysics as well as direct field experience on site meteorology and accumulation conditions. A. Touzeau run the simulations and interpreted the results. A. Touzeau and A. Landais wrote the manuscript. All the authors corrected the manuscript.

#### 775 Competing interests

The authors declare that they have no conflict of interest.

#### 778 Acknowledgements

The research leading to these results has received funding from the European Research Council under the European Union's Seventh Framework Programme (FP7/2007-30 2013)/ERC grant agreement no. 306045. We want to 780 781 acknowledge A. Orsi and M. Casado from LSCE, who contributed to this work through fertile discussions of the model 782 contents and its applications. We are grateful to M. Casado and C. Bréant for providing snow pit temperature and 783 density profiles as well as firn density profiles for model initialization. We would like to thank D. Roche for his advice 784 during model development, as well as on how to make the code available. We are also indebted to J.-Y. Peterschmidt, 785 who provided continual help and useful tutorials on Fortran and Python languages. We thank M. Lafaysse and V. 786 Vionnet at CNRM/CEN for help with the model and meteorological driving data. CNRM/CEN and IGE are part of 787 LabEX OSUG@2020. We thank three anonymous reviewers for their questions and comments, which helped to 788 improve the present article.

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

### **Table 1.** Definition of the symbols used.

| Symbol                     | Description                                                                                                                    |
|----------------------------|--------------------------------------------------------------------------------------------------------------------------------|
|                            |                                                                                                                                |
| Constants                  |                                                                                                                                |
| $T_0$                      | Temperature of the triple point of water (K)                                                                                   |
| $R_v$                      | Vapor constant for water (J·kg <sup>-1</sup> K <sup>-1</sup> )                                                                 |
| $L_{sub}$                  | Latent heat of sublimation of water (J·m <sup>-3</sup> )                                                                       |
| $C_{v0}$                   | Vapor mass concentration at 273.16 K (kg·m <sup>-3</sup> of air)                                                               |
| Dice                       | Diffusivity of water molecules in solid ice $(m^2 \cdot s^{-1})$                                                               |
| $D_v$                      | Diffusivity of vapor in air at 263 K (m <sup>2</sup> ·s <sup>-1</sup> ) (temperature dependency neglected)                     |
| $\rho_{ice}$               | Density of ice (kg·m <sup>-3</sup> )                                                                                           |
| Ă                          | Accumulation (m i.e. per year)                                                                                                 |
| R <sub>moy</sub>           | Average snow grain radius (m)                                                                                                  |
| 2                          |                                                                                                                                |
| $\Delta t_{sol}$           | Characteristic time for solid diffusion (s)                                                                                    |
| $\Delta t_{surf/center}$   | Periodicity of the mixing between grain center and grain surface, because of grain center translation                          |
|                            | (s)                                                                                                                            |
|                            |                                                                                                                                |
| 1D-variables               |                                                                                                                                |
| Т                          | Time (s)                                                                                                                       |
| Ν                          | Layer number from top of the snowpack                                                                                          |
| $\delta^{18}O_{sf}(t)$     | Isotopic composition of oxygen in the snowfall (%)                                                                             |
| $T_{air}(t)$               | Temperature of the air at 2 m (K)                                                                                              |
| ()                         |                                                                                                                                |
| 2D-variables               |                                                                                                                                |
| h(t.n)                     | Height of the center of the snow layer relative to the bottom of the snowpack (m)                                              |
| z (t, n)                   | Depth of the center of the snow layer (m from surface)                                                                         |
| dz(t,n)                    | Thickness of the snow layer (m)                                                                                                |
| T (t, n)                   | Temperature of the snow layer (K)                                                                                              |
| $\rho_{sn}(t, n)$          | Density of the snow layer $(kg \cdot m^{-3})$                                                                                  |
|                            | <i>y y</i> ( <i>b )</i>                                                                                                        |
| $m_{sn}(t, n)$             | Mass of the snow layer (kg)                                                                                                    |
| $C_{v}(t.n)$               | Vapor mass concentration at saturation in the porosity of the snow layer (kg·m <sup>-3</sup> of air)                           |
| $D_{eff}(t,n)$             | Effective diffusivity of vapor in the layer $(m^{2} s^{-1})$                                                                   |
| $\delta^{18}O(t, n)$       | Isotopic composition of oxygen in the snow layer (%)                                                                           |
|                            |                                                                                                                                |
| $F^{18}(n+1\rightarrow n)$ | Flux of the heavy water molecules ( $^{18}$ O) from layer n+1 to layer n (kg· m- $^{2}$ ·s <sup>-1</sup> )                     |
| $F(n+1 \rightarrow n)$     | Vapor flux from layer $n+1$ to layer n (kg· $m-2\cdot s^{-1}$ )                                                                |
| $D_{eff}(t, n)$            | Effective interfacial diffusivity between layers n and $n+1$ (m <sup>2</sup> ·s <sup>-1</sup> )                                |
| $\rightarrow n + 1$ )      |                                                                                                                                |
|                            |                                                                                                                                |
| $R_{van ini}^{i}$          | Isotopic ratio in the initial vapor (i is either <sup>18</sup> O, <sup>17</sup> O or D)                                        |
| $R^{i}_{i}$                | Isotopic ratio in the grain surface sub-compartment before vapor individualization                                             |
| $c^{x}$                    | Ratio between the mass of a given isotopologue in the initial vanor (x is ${}^{18}O$ ${}^{17}O$ ${}^{16}O$ ${}^{1}H$ or D) and |
| vap ini                    | the total mass of vanor (no unit). The mass balance is made senarately and independently for H and                             |
|                            | $O(i a : c^{18} + c^{17} + c^{16} - 1 \text{ and } c^{1H} + c^{D} - 1)$                                                        |
|                            | $O(i.e., c_{vap ini} + c_{vap ini} + c_{vap ini} - 1 and c_{vap ini} + c_{vap ini} - 1).$                                      |
| a i                        | Exactionation coefficients at equilibrium during sublimation (i.i.e. sither $\frac{180}{170}$ or D)                            |
| $\alpha_{sub}$             | Fractionation coefficients at equilibrium during sublimation (1 is entire $^{10}$ , $^{10}$ or D)                              |
|                            | rractionation coefficients during condensation (1 is either <sup>10</sup> U, <sup>17</sup> U or D)                             |
| $\alpha^i_{cond}$          | No fractionation                                                                                                               |
| $\alpha_{cond, aff}^{i}$   | Effective (total) fractionation                                                                                                |
| conu cj j                  |                                                                                                                                |

| $\begin{array}{c} \alpha_{cond\ kin}^{i}\\ \alpha_{cond\ eq}^{i}\\ m_{vap}\ ini\\ m_{surf\ ini}\\ m_{surf\ new}\\ T\\ m_{surf}\\ m_{center}\\ m_{vap} \end{array}$ | Kinetic fractionation only<br>Equilibrium fractionation only<br>Initial mass of vapor in the porosity (kg)<br>Mass of water in the grain surface sub-compartment before vapor individualization (kg)<br>Mass of water in the grain surface sub-compartment after vapor individualization (kg)<br>Ratio of between the mass of the grain surface compartment and the mass of total grain<br>Mass of grain surface compartment<br>Mass of grain center compartment<br>Mass of vapor in the porosity |
|--------------------------------------------------------------------------------------------------------------------------------------------------------------------|---------------------------------------------------------------------------------------------------------------------------------------------------------------------------------------------------------------------------------------------------------------------------------------------------------------------------------------------------------------------------------------------------------------------------------------------------------------------------------------------------|
| $\begin{array}{c} V_{tot} \\ \Phi \end{array}$                                                                                                                     | Total volume of the considered layer<br>Porosity of the layer                                                                                                                                                                                                                                                                                                                                                                                                                                     |
| $m_{surf\ ini}^{18}$<br>$m_{surf\ new}^{18}$<br>$D^{18}/D$<br>$\Delta m_{vap,exc}$                                                                                 | Mass of heavy water molecules ( <sup>18</sup> O) in the grain surface before vapor individualization (kg)<br>Mass of heavy water molecules ( <sup>18</sup> O) in the grain surface after vapor individualization (kg)<br>Ratio of diffusivities between heavy isotope and light isotope<br>Mass of vapor in excess in the porosity after vapor transport (kg)                                                                                                                                     |
| ρ <sub>sn ini</sub><br>ρ <sub>sn new</sub>                                                                                                                         | Density of the snow layer before vapor transport<br>Density of the snow layer after vapor transport                                                                                                                                                                                                                                                                                                                                                                                               |
| T <sub>ini</sub> , T <sub>new</sub>                                                                                                                                | Temperature of the snow layer before and after vapor transport                                                                                                                                                                                                                                                                                                                                                                                                                                    |

| GRIP                       |                               |                                             |
|----------------------------|-------------------------------|---------------------------------------------|
| Accumulation               | 23 cm i.e. yr. <sup>-1</sup>  | Dahl-Jensen et al., 1993                    |
| Annual temperature         | 241 K                         | Masson-Delmotte et al., 2005                |
| Winter temperature         | 232 K                         | (Feb.) Shuman et al., 2001                  |
| Summer temperature         | 261 K                         | (Aug.) Shuman et al., 2001                  |
| Mean $\delta^{18}$ O       | -35.2‰                        | Masson-Delmotte et al., 2005                |
| $\delta^{18}O$ min         | -43 ‰                         | (2m snow pit) Shuman et al., 1995           |
| $\delta^{18}O max$         | -27 ‰                         | (2m snow pit) Shuman et al., 1995           |
| $\delta^{18}O/T$ slope     | 0.46 ‰/°C                     | (2m snow pit) Shuman et al., 1995           |
| DOME C                     |                               |                                             |
| Accumulation               | 2.7 cm i.e. yr. <sup>-1</sup> | Frezzotti et al., 2005; Urbini et al., 2008 |
| Annual temperature         | 221 K                         | Stenni et al., 2016                         |
| Min winter T               | 199 K                         | Stenni et al., 2016                         |
| Max summer T               | 248 K                         | Stenni et al., 2016                         |
| Mean $\delta^{18}$ O       | -56.4 ‰                       | Stenni et al., 2016                         |
| $\delta^{18}$ O min winter | -71.8 ‰                       | Stenni et al., 2016                         |
| $\delta^{18}O$ max summer  | -40.2 ‰                       | Stenni et al., 2016                         |
| $\delta^{18}$ O/T slope    | 0.49 ‰/°C                     | Stenni et al., 2016                         |

 Table 2. Climate and isotope variability at GRIP (Greenland) and Dome C (Antarctica).

| Variable                                            | Equation         |                                                  | Average  | Range    |                     |
|-----------------------------------------------------|------------------|--------------------------------------------------|----------|----------|---------------------|
| Thickness (m)                                       | dz               |                                                  | 1.2.10-1 | 5.10-4   | 8·10 <sup>-1</sup>  |
| Density (kg m <sup>-3</sup> )                       | $\rho_{sn}$      |                                                  | 340      | 300      | 460                 |
| Temperature (K)                                     | Т                |                                                  | 225      | 205      | 255                 |
| Mass (kg)                                           | m <sub>sn</sub>  | $= dz \cdot \rho_{sn}$                           | 42       | 0.15     | 368                 |
| Vapor mass con-<br>centration (kg·m <sup>-3</sup> ) | $C_{v}$          | Eq. (8)                                          | 1.8.10-5 | 1.2.10-6 | $4.4 \cdot 10^{-4}$ |
| Porosity                                            | Φ                | =1- ( $\rho_{sn}$ / $\rho_{ice}$ )               | 0.63     | 0.5      | 0.67                |
| Vapor mass (kg)                                     | m <sub>vap</sub> | Eq. (11)                                         | 1.3.10-6 | 3.10-10  | 2.4.10-4            |
| Minimum ratio                                       | τmin             | $= 1/10^{6}$                                     | 1.10-6   | 1.10-6   | 1.10-6              |
| Maximum ratio                                       | τmax             | $=\frac{c_{v}\cdot \Phi}{\rho_{sn}}\cdot 10^{6}$ | 3.3.10-2 | 1.3.10-3 | 1                   |

<sup>1053</sup> 

**Table 3.** Typical thickness, density, temperature and other parameters of the snow layers in the simulations. The ratio  $\tau$  is the mass ratio between the grain surface compartment and the grain center compartment. It must be chosen within the interval  $[10^{-6}; 10^{6} (C_v \Phi/\rho_{sn})]$  to allow exchanges between grain surface compartment and grain center compartment, on the one hand; and between grain surface compartment and vapor compartment on the other hand (see text for details).

|                              | GRI                      | P simulation                                     | Dome C simulations                               |                                                       |                                                  |                       |  |  |
|------------------------------|--------------------------|--------------------------------------------------|--------------------------------------------------|-------------------------------------------------------|--------------------------------------------------|-----------------------|--|--|
| N°                           | 1 2                      |                                                  | 3                                                | 4                                                     | 5                                                | 6                     |  |  |
| Section                      | 4.1.1.                   | 4.1.2.                                           | 4.2.1.                                           | 4.2.2.                                                | 4.2.3.                                           | 4.2.4.                |  |  |
| Figures                      | Figure 2                 | Figure 3                                         | Figure 5                                         | Figure 6                                              | Figure 7                                         | Figure 8              |  |  |
| Duration                     | 10 years                 | 10 years                                         | 1 year                                           | 1 year                                                | 1 year                                           | 10 years              |  |  |
| Period Jan 2000-<br>Dec 2010 |                          | Jan 2001-<br>Dec 2011                            | Jan-<br>Dec 2001                                 | Jan-         Jan-           Dec 2001         Dec 2001 |                                                  | Jan 2000-<br>Dec 2010 |  |  |
| Atmospheric forcing applied  |                          |                                                  |                                                  |                                                       |                                                  |                       |  |  |
| Air T                        | -                        | ERA-Interim (GR)                                 | ERA-Interim                                      | ERA-Interim                                           | ERA-Interim                                      | ERA-Interim           |  |  |
| Specific humidity            | -                        | ERA-Interim (GR)                                 | ERA-Interim                                      | ERA-Interim                                           | ERA-Interim                                      | ERA-Interim           |  |  |
| Air pressure                 | -                        | ERA-Interim (GR)                                 | ERA-Interim                                      | ERA-Interim                                           | ERA-Interim                                      | ERA-Interim           |  |  |
| Wind velocity                | -                        | ERA-Interim (GR)                                 | ERA-Interim                                      | ERA-Interim                                           | ERA-Interim                                      | ERA-Interim           |  |  |
| Snowfall                     | NO                       | NO                                               | NO                                               | NO                                                    | YES                                              | YES                   |  |  |
| $\delta^{18}O_{sf}$          | -                        | -                                                | -                                                | -                                                     | Function (T)*                                    | Function (T)*         |  |  |
| Model configuration          |                          |                                                  |                                                  |                                                       |                                                  |                       |  |  |
| Initial snow T               | Flat profile<br>(241 K)  | One-year run<br>initialization<br>(Jan-Dec 2000) | One-year run<br>initialization<br>(Jan-Dec 2000) | One-year run<br>initialization<br>(Jan-Dec 2000)      | One-year run<br>initialization<br>(Jan-Dec 2000) | Exponential profile** |  |  |
| Evolution of snow T          | Constant                 | Computed                                         | Computed                                         | Computed                                              | Computed                                         | Computed              |  |  |
| Initial snow d18O            | Sinusoidal<br>profile*** | Sinusoidal<br>profile***                         | -40 ‰                                            | -40 ‰                                                 | -40 ‰                                            | -40 ‰                 |  |  |
| Wind drift                   | NO                       | NO                                               | NO                                               | YES                                                   | YES                                              | NO                    |  |  |
| Homogeneous compaction       | NO                       | NO                                               | NO                                               | YES                                                   | YES                                              | NO                    |  |  |

Table 4. List of simulations described in the article with the corresponding paragraph number. The external atmospheric forcing used for Dome C is ERA-Interim

reanalysis (2000-2013). However, the precipitation amounts from ERA-Interim reanalysis are increased by 1.5 times to account for the dry bias in the reanalysis

(as in Libois et al., 2014). For the second simulation at GRIP, Greenland meteorological conditions are derived from the atmospheric forcing of Dome C, but the

- temperature is modified ( $T_{GRIP}=T_{DC}+15$ ) as well as the longwave down ( $LW_{GRIP}=0.85 LW_{DC}+60$ ).
- \* Using data from one-year snowfall sampling at Dome C (Stenni et al., 2016; Touzeau et al., 2016), we obtained the following Eq. (16) linking  $\delta^{18}$ O in the snowfall
- to the local temperature:  $\delta^{18}O_{sf} = 0.45 \times (T 273.15) 31.5$ .
- **1066** \*\*The exponential profile of temperature used in simulation 6 is defined using Eq. (20):
- $T(z) = T(10m) + \Delta T \times \exp(-z/z0) + 0.1 \times z$

(20)

- with T(10m) = 218 K,  $\Delta T = 28$  K, and z0 = 1.516 m.
- It fits well with temperature measurements of midday in January (Casado et al., 2016b).
- \*\*\* The Greenland snowpack has an initial sinusoidal profile of  $\delta^{18}$ O defined using Eq. (19):  $\delta^{18}O = -35.5 8 \times \sin\left(\frac{2\pi \times z}{a \times \rho_{ice}/\rho_{sn}}\right)$

|                                                 | GRIP sensitivity tests     |                                 |                                   |                                 | Dome C sensitivity tests  |                           |                                 |                                                        |                                                        |  |
|-------------------------------------------------|----------------------------|---------------------------------|-----------------------------------|---------------------------------|---------------------------|---------------------------|---------------------------------|--------------------------------------------------------|--------------------------------------------------------|--|
| N°                                              | 1                          | 2                               | 3                                 | 4                               | 1                         | 2                         | 3                               | 4                                                      | 5                                                      |  |
| Section                                         | 4.3.                       | 4.3.                            | 4.3.                              | 4.3.                            | 4.3.                      | 4.3.                      | 4.3.                            | 4.3.                                                   | 4.3.                                                   |  |
| Figures                                         | Figure 9                   | Figure 9                        | Figure 9                          | Figure 9                        | Figure 10                 | Figure 10                 | Figure 10                       | Figure 10                                              | Figure 10                                              |  |
| Duration                                        | 6 months                   | 6 months                        | 6 months                          | 6 months                        | 3 years                   | 3 years                   | 3 years                         | 3 years                                                | 3 years                                                |  |
| Period                                          | Jan-<br>Jun 2000           | Jan-<br>Jun 2000                | Jan-<br>Jun 2000                  | Jan-<br>Jun 2000                | Jan 2000-<br>Dec 2002     | Jan 2000-<br>Dec 2002     | Jan 2000-<br>Dec 2002           | Jan 2001-<br>Dec 2003                                  | Jan 2001-<br>Dec 2003                                  |  |
| Atmospheric forcing applied                     |                            |                                 |                                   |                                 |                           |                           |                                 |                                                        |                                                        |  |
| Air T                                           | -                          | -                               | -                                 | -                               | -                         | -                         | -                               | ERA-Interim                                            | ERA-Interim                                            |  |
| Specific humidity                               | -                          | -                               | -                                 | -                               | -                         | -                         | -                               | ERA-Interim                                            | ERA-Interim                                            |  |
| Air pressure                                    | -                          | -                               | -                                 | -                               | -                         | -                         | -                               | ERA-Interim                                            | ERA-Interim                                            |  |
| Wind velocity                                   | -                          | -                               | -                                 | -                               | -                         | -                         | -                               | ERA-Interim                                            | ERA-Interim                                            |  |
| Snowfall                                        | NO                         | NO                              | NO                                | NO                              | NO                        | NO                        | NO                              | NO                                                     | NO                                                     |  |
| $\delta^{18}O_{sf}$                             | -                          | -                               | -                                 | -                               | -                         | -                         | -                               | -                                                      | -                                                      |  |
| Model configuration                             |                            |                                 |                                   |                                 |                           |                           |                                 |                                                        |                                                        |  |
| Initial snow T                                  | Flat<br>profile<br>(241 K) | Flat<br>profile<br>(241 K)      | Flat<br>profile<br>(241 K)        | Flat<br>profile<br>(241 K)      | Flat profile<br>(241 K)   | Flat profile<br>(220 K)   | Flat profile<br>(220 K)         | One-year<br>run<br>initialization<br>(Jan-Dec<br>2000) | One-year<br>run<br>initialization<br>(Jan-Dec<br>2000) |  |
| Evolution of snow T                             | Constant                   | Constant                        | Constant                          | Constant                        | Constant                  | Constant                  | Constant                        | Computed                                               | Computed                                               |  |
| Initial snow d180                               | Sinusoidal<br>profile***   | Sinusoidal<br>profile***        | Sinusoidal<br>profile***          | Sinusoidal<br>profile***        | Sinusoidal<br>profile**** | Sinusoidal<br>profile**** | Sinusoidal<br>profile****       | Sinusoidal<br>profile****                              | Sinusoidal<br>profile****                              |  |
| Wind drift                                      | NO                         | NO                              | NO                                | NO                              | NO                        | NO                        | NO                              | NO                                                     | NO                                                     |  |
| Homogeneous compaction                          | NO                         | NO                              | NO                                | NO                              | NO                        | NO                        | NO                              | NO                                                     | NO                                                     |  |
| Mass ratio $\boldsymbol{\tau}$ within the grain | 1.10-6                     | 5 <sup>.</sup> 10 <sup>-4</sup> | 3.3 <sup>.</sup> 10 <sup>-2</sup> | 5 <sup>.</sup> 10 <sup>-4</sup> | 5.10-4                    | 5.10-4                    | 5 <sup>.</sup> 10 <sup>-4</sup> | 5 <sup>.</sup> 10 <sup>-4</sup>                        | 3.3·10 <sup>-2</sup>                                   |  |

|      | Period for recrystallization<br>Δtsurf/center | 15 days | 15 days | 15 days | 2 days | 15 days | 2 days | 15 days | 15 days | 15 days |
|------|-----------------------------------------------|---------|---------|---------|--------|---------|--------|---------|---------|---------|
| 1074 |                                               |         |         |         |        |         |        |         |         |         |

- Table 5. List of the sensitivity tests performed at GRIP and at Dome C. The external atmospheric forcing used for Dome C is ERA-Interim reanalysis (see Table 4).
- \*\*\*The Greenland snowpack has an initial sinusoidal profile of  $\delta^{18}$ O defined using Eq. (19):

$\delta^{18}O = -35.5 - 8 \times \sin\left(\frac{2\pi \times z}{a \times \rho_{ice}/\rho_{sn}}\right)$

**1078** \*\*\*\*The Dome C snowpack has an initial sinusoidal profile of  $\delta^{18}$ O defined using Eq. (21):

1079 
$$\delta^{18}O = -48.5 - 6.5 \times \sin\left(\frac{2\pi \times z}{a \times \rho_{ice}/\rho_{sn}}\right)$$
 (21)

- Figure 1. Splitting of the snow layer into two compartments, grain center and grain surface, with a constant mass ratio
- between them. The vapor compartment is a sub-compartment inside the grain surface compartment and is only defined at specific steps of the model.