# Peer review of "Numerical experiments on vapor diffusion in polar snow and firn and its impact on isotopes using the multi-layer energy balance"

_Geoscientific Model Development, 2017_

## Short Comment (SC1) · 7 Nov 2017

Dear authors,

In my role as Executive editor of GMD, I would like to bring to your attention our Editorial version 1.1:

http://www.geosci-model-dev.net/8/3487/2015/gmd-8-3487-2015.html

This highlights some requirements of papers published in GMD, which is also available on the GMD website in the 'Manuscript Types' section:

http://www.geoscientific-model-development.net/submission/manuscript_types.html

[Figure]

In particular, please note that for your paper, the following requirements have not been met in the Discussions paper:

- "The main paper must give the model name and version number (or other unique identifier) in the title."

- "If the model development relates to a single model then the model name and the version number must be included in the title of the paper. If the main intention of an article is to make a general (i.e. model independent) statement about the usefulness of a new development, but the usefulness is shown with the help of one specific model, the model name and version number must be stated in the title. The title could have a form such as, "Title outlining amazing generic advance: a case study with Model XXX (version Y)"."

- "All papers must include a section, at the end of the paper, entitled 'Code availability'. Here, either instructions for obtaining the code, or the reasons why the code is not available should be clearly stated. It is preferred for the code to be uploaded as a supplement or to be made available at a data repository with an associated DOI (digital object identifier) for the exact model version described in the paper. Alternatively, for established models, there may be an existing means of accessing the code through a particular system. In this case, there must exist a means of permanently accessing the precise model version described in the paper. In some cases, authors may prefer to put models on their own website, or to act as a point of contact for obtaining the code. Given the impermanence of websites and email addresses, this is not encouraged, and authors should consider improving the availability with a more permanent arrangement. After the paper is accepted the model archive should be updated to include a link to the GMD paper."

Thus please add the models name (SURFEX/Crocus ?) and the version number to the

title of your article. Additionally, it would be good if the explicit version described in this article would be archived in a permanent archive providing a DOI (e.g. Zenodo).

Yours,

Astrid Kerkweg
* * *

---

## Referee Comment (RC1) · Anonymous Referee #1 · 30 Nov 2017

Touzeau et al. presents a detailed study on implementing isotopes into a semi-complex one-dimensional snow pack model. Unfortunately it is my opinion that the authors still need a little bit more work to allow this publication to become a significant contribution to the community. I am though positive that the manuscript will be publishable after my major comments have been taken into account.

Major comments: (The following list of comments are not ordered in accordance with importance as they are more or less equally important)

- The use of parentheses throughout the manuscript is not in accordance with good practice. It makes reading the manuscript difficult. Please rewrite relevant sentences.

[Figure]

- The term 'oriented vapor transport' seems to complicate the reading. The model has already been defined as 1D and hence no need to include the word 'oriented'. Please remove throughout paper.

- 'Vapor density gradients'. Please change to 'vapor pressure gradients' throughout the paper. The use of vapor pressure is the normal term used i.e Merlivat and Jouzel 1979 and Jouzel and Merlivat 1984 etc.

- If a sentence is longer than 2 lines, it is most likely too long. Please refrain from using extremely long sentence that complicates the understanding of the manuscript. This is seen at several instances through out the manuscript, but my favorite example is section 2.1 L106-109 where I really have no idea what is being described.

- Rephrase 'mean local pluriannual value' or describe what you mean.

- Rephrase 'oriented processes' or describe what you mean

- In L113 you write "Indeed, higher temperatures correspond to higher vapor densities, and also higher diffusivities in the vapor and the solid phase". This is correct, but then you line 260 define the vapor diffusivity in air to be a constant despite that it is depending on both temperature and pressure. This needs to be corrected. You need to allow for a temperature and pressure dependence on the diffusivity.

- I have a problem with your first sentence in the introduction "Ice is a key archive for past climate reconstruction, which preserves . . . indications relevant to the temperature of formation of the snow precipitation. . . variations of the isotopic ratio of oxygen and deuterium". This sentence is problematic because you have co-authors who have published papers documenting in both Greenland and Antarctica how the isotopic composition of the deposited precipitation is changed through exchange with the atmospheric water vapor isotopes. You cite 8 publications to document your statement, but they are between 10 and 30 years old. You thereby disregard published research for the last five years. Please update.

- In L 17: Why not study the influence of temperature and not only temperature gradients? What is the difference between "compaction" and "Wind compaction".? Do you study the effect of amount of precipitation or the isotopes of the precipitation?

- L 52: Use another word than "Mechanical shuffling"

- L119: You write that the annual cycles generally disappear at sites with accumulation lower than 200 kg/mˆ2/year – but does that not depend on time scales – please be more precise.

- L120: You write that the diffusion is more intense in the upper layers – but don't the diffusion depend on the isotopic gradient and would you not expect that to be larger further down in the snow? Please be precise! Also the word 'intense' might not be the best to use in this case

- Section 3.1.2: Describe why the new vapor transport subroutine is inserted after module 5 but before module 6? What are the thoughts behind this?

- L251: "…is the effective diffusivity of water vapor in the snow at the interface". Do you mean effective diffusivity of water vapor in the air between the snow grains?

- Equation 6: I am not sure, but isn't a layer thickness missing from this formula as you might not have the same layer thickness in layer n and n+1?

- Equation 7: Why do you use an analytical approximation of Clausius-Clapeyron around zero and not a more precise empirical formula?

- L 313 : "Long time" – what do you mean – please be precise

- L334: What vapor are you referencing to? $H_2O$ in general or $H_2^{16}O$?.

- L335: I believe you meant to write "we will still have diffusion of heavy water isotopes during conditions where the water isotopic gradient is non-zero.

- L335-336: The sentence is very convoluted. I believe you could also have zero flux

of H216O but a flux of H218O in one direction and HD16O in another direction.

- L353: "Here the condensation of excess vapor occurs without additional fractionation". Why do you make this assumption". Whenever you have a phase change due to condensation you will have isotopic fractionation. I think this is something that needs to be updated in your code.

- L356: "The transfer of isotopes takes place from the grain surface toward the vapor without fractionation" If you assume this then the interstitial vapor will not be in isotopic equilibrium with the snow surface. This would then correct itself. Hence I think that your code needs to be set-up such that the interstitial vapor is in isotopic equilibrium with the snow surface at all time.

- Please note that you throughout the paper are mixing up GRIP and Summit. They are two different geographical places in Greenland albeit being close to each other.

- I am surprised to read that there are no density measurements for neither GRIP nor Summit and that you therefore use NGRIP. Please double-check this.

- You do not give a relationship for the isotope-temperature relationship for GRIP. Please correct.

- Figure 2: You should include a comparison with the model of Johnsen et al. 2000

- Figure 3: You write in the manuscript that the temperature is varying but on the figure you only show temperatures for the summer. Does this mean that you only use summer temperatures? I would expect you would use varying temperatures through the whole year.

- I am surprised to find that your model do not show an influence of temperature gradients at GRIP as you would normally assume that temperature gradients would force vapor to be transported between layers due to the vapor pressure gradient?

- L503: Is the attenuation at GRIP significant larger than NEEM? 86% and 90% seems

very similar.

- L511: Why don't you calculate the attenuation using Johnsen at GRIP such that you can compare with Bolzan and Pohjola?

- L526: It is unclear how Denux in 1996 can indicate that a study by Johnsen et al. in 2000 overestimates the attenuation. Time travel hasn't really been possible yet. You might write that "A study by Denux (1996)..."

- L528: You write that Johnsen et al. should take into considerations temperature gradients in order to not overestimate the attenuation. But would you not expect that temperature gradients would increase the attenuation due the vapor transport driven by vapor pressure gradients?

- I strongly suggest that you set up an experiment with Crocus that allow you compare as closely as possible the simulated attenuation with the calculated attenuation using the model of Johnsen et al. 2000.

- Section 4.2.1: I suggest to remove the detailed description of simulation of density at Dome C to a supplementary material as it influences the flow of the manuscript which should be focusing on the evolution of isotopes in the snow pack.

- L 604: You suggest that the higher diffusion at GRIP compared to Dome C could be explained by higher temperatures – but in line 260 you assume that the diffusivity is constant and not influenced by temperature.

- In general for all the figures you need to adjust the values for the color bar such that you don't have too many digits. For example in Figure 2 the color bar should go from -0.6 to 0.6 and in figure 3 it should be -1.9 to 0.8.

- Figure S1: Why not combine panel b, c, and d

Minor comments

L14 "The isotopes ... resolution" should not be in abstract

L16 "condensation is realized" – what does this mean

L21: "model underestimates" -> modeled attenuation due to diffusion is underestimated, or that other processes, such as ventilation influences attenuation

L24-25: should be moved to conclusion

L42: Randomness in the core stratigraphy -> stratigraphic noise

L45: series of snow pits -> series of records from snowpits

L53: ice microstructure at solid state ->snow grains due to solid diffusion

L58-61: Cite Ebner et al. 2016 and 2017

L87 Missing parenthesis after Brun et al. 2011

L99: Quick survey-> brief overview

L118: Wavelength of what?

L178: What do you mean by "Permanent cycles"

L184: to get an -> to obtain an

L185: Remove the content of the parenthesis.

L224: What does this mean: "and taken to compensate yearly accumulation

L240: What about the influence of absorption of radiation energy in layers below the surface layer?

L254: "Interface": Please be more precise on defining what interface you are referring to

L258: "interpenetrate": What do you mean?

L296: "that are" -> being

L304: Have you defined kinetic fractionation previously?

EQ 12: typo in D_eff_n&n

L486: "Amplitude decrease by -1.3 o/oo" – do you mean amplitude increase by 1.3 o/oo

---

## Referee Comment (RC2) · Anonymous Referee #2 · 1 Dec 2017

The posr-depositional modification is an important but poorly understood part of the "isotopic paleo-thermometer". After the solid precipitation is deposited on the top of the polar ice sheet snow surface, its isotopic content is changed drastically due to the water and mass exchange with the atmospheric water vapor and due to molecular diffusion in snow. These processes disturb or even completely erase the initial climatic signal recorded in the isotopic content of the precipitation. To solve this problem, different approaches are applied including modeling of the snow pack evolution during snow metamorphism.

This manuscript is an attempt to simulate the snow isotopic content of the polar snow in

the course of the post-depositional processes. For the first time the snow-pack Crocus model is applied for this purpose.

The authors clearly understand that this work is a small step towards the full description of the isotopic post-depositional modifications. A lot of efforts still has to be done. However, this attempt deserves to be published as a separate paper in "Geoscientific Model Developement" journal. The manuscript is nicely structured and provides a good review of literature on the formation of the climatic signal in the snow isotopic composition. The authors make a nice attempt to describe in simple way a rather complicated process of the isotopic modifications in the snow thickness. I do not have major corrections, only a few minor comments or questions:

In your model you do not take into account the mechanical snow mixing by wind. This mixing erases the initial climatic signal (shorter than few years) in central Antarctica, and makes the vertical isotopic profiles in the upper part of snow thickness similar to white noise. Recent study by Thomas Laepple (https://www.the-cryosphere-discuss.net/tc-2017-199/) showed that the filtering of this noise by isotopic diffusion can create false cycles in the isotopic profiles. So, I suggest that in the further versions of you model you introduce random component of the initial isotopic composition of the precipitation (or of the upper snow layer if you wish) in parallel to the regular component given by precipitation events. You might mention this in section 4.4. and conclusion.

Other minor comments:

line 33 - better to write "1950s"

lines 352-353 - why condensation is without additional fractionation?

lines 502-503: the values (86% and 90%) are the remaining amplitudes, right?

section 4.2.3: how much snowfall have you added to the snow thickness in this simulation?

lines 582-583: this gives 10 cm / year, but above you said that the accumulation rate at

DC is 8 cm / year (snow equiv.).

Figure 2d is a bit misleading. From the first glance a reader may think that the seasonal amplitude is increasing with time. Then, it becomes clear that it is actually d18O change that is increasing with time. It would be nicer to show here the d18O values themselves (instead of d18O changes), so that the colors would nicely illustrate the fading isotopic variability.

The same comment is for Figures 3 and 12.

line 1055: December 2001.

---

## Referee Comment (RC3) · Anonymous Referee #3 · 4 Dec 2017

**1   Overview**

Firn isotope diffusion is a process that affects the $\delta^{18}$O signal of polar snow from the time of deposition until pore close–off. Taking place in the vapor phase within the porous medium of the firn and driven by the apparent seasonal, annual and multiannual isotopic gradients it results in an attenuation of the $\delta^{18}$O signal, often obliterating its annual component. Assuming a good estimate of the diffusive rates in firn is obtained, a "reverse calculation" of diffusion can be possible that allows the (almost) complete reconstruction of the initial signal. Additionally, knowledge of the diffusive rates offers valuable information on past firn temperatures and as a result can be used as a pale-

othermometry tool if ice core data of sufficient resolution and precision are available.

Previous studies have looked into the desciption and characterisation of these effects and part of these studies suggests that post depositional processes different to purely fickian diffusion of water isotopes can also be at play acting supplementary to the signal attenuation affects or even introducing biases (Town et al., 2008). These processes are mostly of advective nature caused by the bulk movement of air and vapor in the snow, driven by pressure and temperature variations.

In this work titled "Numerical Experiments on isotopic diffusion in polar snow and firn using a multi-layer balance model", Touzeau et al attempt to build and test a water isotope module on top of the Crocus snowpack model. In particular, the authors focus on trying to simulate post-depositional effects that cause changes of the initial $\delta^{18}$O signal in polar snow and firn. Processes related to snow/firn isotope diffusion as well as diffusive vapor transport due to temperature gradients in the firn are modelled assuming various scenarios. The study focuses on two different regimes that are representative of conditions typical for deep ice coring sites on Greenland and East Antarctica. Ice core data sets are also used in order to evaluate the performance of the model and the results are also compared to existing firn isotope diffusion modelling approaches.

This is a very welcome condtribution and it most certainly points to the correct direction with respect to future modelling efforts. The study also fits very well the description and scope of the GMD journal and the overall quality of the research conducted is of high level. Thus I would recommend it for publication in GMD after the following points are carefully considered by the authors.

**2 General comments**

1. Unfortunately the language of the manuscript requires a signifficant revision. In particular there are examples in the text where technical/physical terms are used

wrongly and many definitions appear to be loose. This is particularly problematic for a manuscript of this type, where modelling approaches and physical processes are described.

The most notable axample is the description of the transport mechanisms in snow in sections 2.1.1 and 2.1.2. Diffusion is a very well defined process and unfortuanately the term is used falsely several times in sections 2.1.1 and 2.1.2 (and elsewhere in the manuscript). After reading these two sections I feel confused about the meaning of many of the terms used here and as a result about the kind of methods followed and the assumptions made in this study.

What is an "oriented process" for example? In page7line73 the sentence "We focus on the impact of oriented vapor transport caused by vapor density gradients in the snow..." is very untechnical and unfortunately creates a lot of confusion about what the authors have done. If the term "vapor density" indeed refers to "vapor (molar?) concentration" as I assume then the process described here is a vapor diffusion process.

After having read the text several times and tried to infer what the authors try to describe in sections 2.1.1 and 2.1.2 I conclude that they split the processes under consideration in two kinds. The first, what they call "signal attenuation on a vertical profile", is a the combination of two processes, (a) solid isotope diffusion and (b) firn isotope diffusion in the vapor phase. The first is extremely slow and can easily be neglected in this study. I find it important that the authors point out in the text that solid diffusion affects all isotopes equally. The second is a diffusive process taking place in the porous medium pf the firn driven by the isotopic gradients. Both processes introduced here follow the same physical principle ie transport of mass due to concentration gradients of a substance. The transport occur along (or down) the concentration gradients and not "against" as often described in the text.

The second category of processes outlined in section 2.1.2 and termed as "oriented processes". My interpretation of the text is that this type processes are "bulk motion" processes either due to pressure or temperature gradients. The first case is a typical example of advective transport. The second is a bit more complicated however the term diffusion used by the authors is incomplete. Temperature gradients in the snow will eventually cause vapor concentration gradients. The latter, will drive a diffusion process for the vapor as a whole. However this cannot be seen as an isotope diffusion process due to the fact that the diffusive transport of vapor has nothing to do with isotopic gradients. Eventually of course the diffusive transport of water vapor will very likely bring vapor molecules in layers of the snow with different isotopic composition where subsequently an isotope diffusion process will occur localy.

This was only an example of how the loose use of technical terms and faulty language creates unnecessary confusion to the reader already from the introduction, leading possibly to confusion and misunderstandings of the methods and principles used in this study. I find it essential that the authors look into the manuscript carefully and revise the text accordingly. In the "Specific Comments" section I include more of these examples as they appear in the text.

2. There is an unclear situation regarding the vapor diffusivity parametrisation and value used in this study. It is not exactly clear if there is a temperature dependency of the effective diffusivity $D_{eff}$ to temperature. Based on equation 5 in the manuscript and the comment on the value of $D_v$ I conclude that the value of $D_{eff}$ is taken constant and reflects a temperature of 263 K. If this is indeed the case I would be inclined to question the validity of many of the statements found in the manuscript that concern the comparison of this model with other models of diffusion or results from ice core data. The diffusivity coefficient is heavily dependant on temperature and thus a constant value is an oversimplification for such a study.

I would strongly prefer a version of the manuscript where the diffusivity is allowed

to depend on temperature. However, if the authors indeed choose to follow the approach of constant diffusivity they will need to stress out very clearly in the manuscript that the comparisons presented here are essentially between different things. This should be even more prominent for the case of the Dome C modelling experiments due to the very large difference between the site temperature and the temperature used for the diffusivity coefficient value (almost 40K).

3. Despite my belief that the work performed by the authors is of high quality I need to point out that several elements of the manuscript feel opaque not allowing the reader to judge for herself on the quality of the work and the significance of the results. I find this a fundamental weakness of the manuscript that needs to be addressed. In particular:

- The authors claim that the model is evaluated for the top 10 m of snow. However only the top 50-60 cm are presented.

- The authors do not provide any information about neither the ice core data used nor the method used to calculate peak-amplitudes. The latter is not a straight forward procedure and can have a significant influence on the result of such model-data comparisons for diffusion. Information about the depth interval the data originate from, the temperature, accumulation and pressure conditions of the sites as well as the resolution of the data are pieces of information a thorough reader needs to have access to. Present the ice core data.

- The initial $\delta^{18}$O profile as well as snapshots of some layers should be plotted. The difference plots with the plethora of colors do not add anything neither for the case of density nor for the case of $\delta^{18}$O. The colormaps of these plots are unfortunately very ambiguous to read and despite having the max and min values it looks to me that some of these colormaps are non linear. In combination with the very small difference values for both the density and

the $\delta^{18}$O these color plots leave me guessing. There is very little valuable information I can extract from them.

- The study considers all three isotopologues of water ($\delta^{18}$O, $\delta^{17}$O and $\delta$D) however the authors choose to present only the results for $\delta^{18}$O. Based on (Johnsen et al., 2000) the diffusive attenuation is expected to be stronger for $\delta^{18}$O compared to $\delta$D. Can the model produce this differential signal? This is a very simple test.

4. The discussion about the comparison with GRIP data feels incomplete and not thorough. The actual data set is never shown in the manuscript while there is very little information about how diffusion is estimated for this data set. Measuring peak-to-peak amplitudes on ice core $\delta^{18}$O data can be very misleading as the initial $\delta^{18}$O value is unknown and most likely it has been variable through the time. One technically correct way to estimate diffusion on data is to look into the spectral domain and estimate diffusion length values. Either way the reader har practically no access to information about how diffusion is estimated from the GRIP data. Additionally, it should be noted that the GRIP data set, originating from a certain depth interval in the ice core (that is not given in the manuscript) it may have experienced a combination of temperature and acccumulation different from the modern one. Does the comparison presented here take this into account? In particular if the CROCUS model only uses a fixed diffusivity value for 263 K, there is no doubt that there will be a discrepancy with the data deduced diffusion. These are very important elements of such a study and are notaby absent in the manuscript.

5. Plots and captions need to be reworked. There are several stylistic inconsistencies that should not be allowed for a publication of this quality. A mixture of different font types, missing measurement units from axes, different approaches in presenting measurement units (using either parentheses or a / sign) and a ‰sign presented in two ways. I think that many of the captions are too long while

in the same time they miss one important piece of information that is the number of the experiment and maybe the ice core site under consideration. I do not think it is the job of a reviewer to go through every single detail and problem with the plots thus I will trust that the authors are certainly able to carefully go through the presented plots and make the necessary changes.

6. Regarding the references given, I think that for the introduction section there is probably an overwhelming number of works cited and a small clean-up is possible. More importantly though, some of the works cited are not peer reviewed belonging to the "Discussions" versions of some of the Copenicus publications journals. I believe that the authors should consider these cases and preferably either omit them or update their references list in case some of the papers in question have reached a post peer-review status.

**3  Specific comments**

Here some more specific comments for the authors.

P2L45

"and then only stacking...". As one looks in higher depths in a core this is less of an issue.

P3L57

Make sure the reader understands this is vapor-solid exchange in the porous medium of the firn.

**GMDD**

P3L67

Diffusion length mentioned here but no definition given.

P3L74

"...and of diffusion against isotopic gradients" Is this vapor firn diffusion or solid?

P5L117

For an informative plot on the matter see Gkinis et al 2014. Higher accumulation rates also result in increased densification rates and therefore reduced diffusivities.

P5L120

Diffusion indeed takes place in the ice column but with rates orders of magnitude lower than that of firn. You want to be more specific about it in the text as you often mix the terms vapor and solid diffusion without being specific about the process taking place in the porous of the firn or in the solid ice.

P6L143

It would be helpful to add even one sentence where you explain why and how the spherical ice elements approach is too simplistic (is it?).

[Figure]

"...the transfer of molecules from the grain boundary towards the center of the grain is very slow" Solid diffusion at the temperatures we are talking about is indeed slow. However this sentence gives a false impression that there is a 1-way motion from boundary to center. This is wrong for two reasons. Firstly, any diffusion process would not result in a 1-way motion of molecules. Secondly and more important, solid diffusion in ice seems to be a self-diffusion process following a vacancy mechanism. This means that there is no isotope effect and diffusion affects all isotopologues equally or in other words molecular transport does not take place along and due to the isotopic gradients in ice (therefore there is no index denoting isotopic species in Eq. 1 - ice diffusivity concerns water molecules in the solid phase regardless of their isotopic composition). As a result the model used here of an isotopically heterogenuous material with internal and external layers does not cause any isotope diffusion in the solid phase due to the radial gradients. In a perfectly homogeneous material you should be expecting the same magnitude of diffusive mixing in the solid phase as in the heterogenuous meterial assumed in the text. It would be good to correct these errors in section 2.2.2 and clarify the precence of the self-diffusion mechanism.

The calculations of characteristic times in this section look correct and are relevant though. Just make sure that you clearly explain that this characteristic time concerns not only movement of the different isotopologues along a specific path (surface to center of grain) but of ALL water molecules towards all directions in the grain and across the grains.

section 3.1.1

I was wondering if it would be possible to outline the components of the Crocus model in a summarising table and shorten this section significantly?

P10L221

There does not seem to be any dependence of the densification rate to temperature or accumulation rate. Neither is there a two or three stage densification process as done usually in some other densification models. Can you elaborate on this? Would this model be suitable for modelling the full firn profile from surface to firn–ice transition?

Eq.4

I think the right term for the quantity $\rho_v$ should be *mass concentration* instead of *vapor density* (this term is wrongly used in more places in the manuscript). Density refers to the ratio of mass to volume of the same substance whereas what you use here is the mass of vapor devided by the volume of air in the open porosity of the layer under consideration. Accordingly I think you should change the symbol from $\rho_v$ to $C_v$ or similar. I may be missing something but if I use Fick's first law and a forward difference differenciation scheme I do not get the factor of 2 as in Eq. 4. Can you elaborate please?

P11L260

$D_v$ is a function of temperature and pressure. How significant is the fact that you are using a fixed value?

Eq.5

The fact that the diffusivity used here is independent of temperature and site pressure seems problematic to me. Can you comment on this and add a line in the manuscript about the effect of this approach?

Eq. 7

Again strictly speaking the quantity you need here is a concentration and not a density. Change the symbols as well.

P14L312

Consider using the term *rare isotope* instead of *heavy isotope*. Also using an index $i$ is more appropriate than a * sign as later on in Eq. 8 and 9 you use "17", "18" and "D" in the position of the * sign.

Eq. 8 and 9

The term $C_{\mathsf{vap}}^i$ needs to be clarified both here and in Table 1. What I understand is that $C_{\mathsf{vap}}^i$ refers to isotope concentration as

$$C_{\mathsf{vap}}^{16} = \frac{[\mathrm{H}_2^{16}\mathrm{O}]}{[\mathrm{H}_2^{18}\mathrm{O}] + [\mathrm{H}_2^{17}\mathrm{O}] + [\mathrm{H}_2^{16}\mathrm{O}]} = \frac{{}^{16}R}{{}^{18}R + {}^{17}R + {}^{16}R} = \frac{1}{{}^{18}R + {}^{17}R + 1} \tag{1}$$

(again (Mook , 2000) is a good source for definitions). However later in Eq. 11 you seem to be using the same quantity for something slightly different, this time the masses ratio and not the abundancies ratio. Can you comment on that and make sure the definitions are clear to the reader? If needed add a definition equation in Table 1.

P14L319

I would be very interested to know why you have used the fractionation factors from (Ellehoj et al., 2013)

P14l329

This note concerns the use of the term *kinetic fractionation* throughout the whole manuscript. Kinetic effects refer to anything that is non-equilibrium. And indeed fractionation due to the different diffusivity coefficients for the different isotopologues is a type of kinetic fractionation. Though it is an overstatement to claim that you have included all possible kinetic fractionation processes by only using the ratio of the diffusivities. Fractionation effects related to different binding energies of the molecules for example can also be affected by a non-equilibrium/kinetic regime and this is something that is not addressed by the $D^*/D$ term. I would suggest that you go through the manuscript and clarify this (term kinetic is used in pages 1, 13, 14, 16, 30, 42 and 43). I would also refer the authors to the sections 3.1 to 3.5 in vol. 1 of (Mook , 2000). Even though some of this definitions sound trivial I think the manuscript can benefit greatly by getting these small details right, thus avoiding misconceptions.

Eq. 12

See previous comment on Eq. 4

P15l353

"Here the condensation of excess vapor occurs without additional fractionation". Is this not unphysical. Can you comment?

P18l407

Rephrase the sentence. The term "oriented processes" (also used in 2.1.2) is not a technical term. From what I understand your use of the term "oriented processes"

refers to advection–based processes that bias the isotopic signal. Diffusion is not such a process, it attenuates the isotopic signal and is driven by isotopic composition gradients as apposed to for example ventilation that is driven by a bulk motion of air in the open porosity. Additionally diffusion takes place for much longer than 12 m (depending on close–off depth) whereas the extend to which ventilation is apparent in polar firn can be debaded.

P18l414

"Thus the diffusion process can only be studied in the first 2 m of the model snowpack" Can you elaborate on this? Is it a computation time issue that does not allow for thinner layers below the top 2 m. How do the calculations look like below this depth?

P19l432

Stick to one name for GRIP/Summit throughout the manuscript.

P19l445

Citing a published work (Bréant et al., 2017) dealing with the density studies at DomeC and GRIP is of course acceptable though the density profile here is of great importance for the diffusion calculations, therefore giving some more information and possibly figures would be appreciated. Additionally you give the density as a function of $n$ and $t$ where $t$ (the model time) is an independent variable to $z$. Can you explain this a little bit better? How do Eq. 16 and 17 give you an evolution of the column density and the densification rates? Please also update the reference to the one past the review process, published in the Climate of the Past.

P19l450

Earlier in the manuscript you mentioned that all diffusivity values are for a temperature of 263K. Does the isothermal profile at 241 K affect this and if yes how?

P20l457

In Table 2 you refer to a different work for the value of accumulation at GRIP. Be consistent and use only one reference.

P20l457

You can be a bit more specific and call it "peak to peak amplitude".

P20l461

Please refer to general comment nr. 1 with respect to the difference between "isotope firn diffusion" due to isotopic gradients in the snow/firn and signal attenuation/alteration because of air or vapor "bulk motion" driven by pressure or temperature gradients in the snow.

P20l471

What does the term "densities" refer to here? Vapor densities (use term water vapor concentration instead) or firn densities. If it refers to firn densities can you be more specific about how your densification rates depend on temperature?

Section 3.3.3

In my view this section is unnecessary and its sole sentence can be included in the previous section.

P21l486

Is this peak to peak amplitude?? Also writing that maxima and minima are reduced sounds inaccurate. Attenuation would result in reduced maxima and increased minima, or in the difference between the two being lower. Lack of visual examples makes this type of language errors quite critical as they can be very confusing for the reader.

P21l490

The description of the model in the previous sections suggests that the diffusivity coefficient is independent of temperature. It is not clear though if there is some dependancy of the diffusivity to temperature for your model experiments. One possible cause of the increased depletion for the upper few cm could also be that the firn appears to be quite warmer, something that would result in enhanced diffusion rates for these few cm of the firn column thus attenuating this part more compared to the layers below. I also miss some info on the density profile here and specifically the surface density.

Section 4.1.2

This section lacks a proper description of the methods used in order to estimate the amplitude of the isotopic signal for the cores presented. In Johnsen et al. the amplitude of the annual signal is computed using a rather sophisticated modificaton of the Maximum Entropy Method where the annual signal spectral peak is integrated to

give a value in permile. This of course is an estimate dependent on the initial iso-topic signature (some years have a greater amplitude thatn others) and for this reason 5m intervals are considered in Johnsen et al. How is this analysis performed here? Can a 20cm interval produce satisfactory results when the layer thickness for these depths at NEEM is in the order of 50-60 cm? Also the term half–amplitude should be peak–amplitude or semi–amplitude.

P22l513

GRIP is also slightly colder.

P22l520

This is a very good point. Temperature has a strong impact on the diffusivity coefficient. It is certainly relevant to consider various other processes that can be the cause to these discrepancies though a very simple test you culd do here is to apply the Johnsen et al diffusivity parametrisation in Crocus and compare the results. I am puzzled by the values that are given here ofr the firn diffusivity. These are much closer to air diffu values. Firn diffusivity values for $\rho = 350\mathrm{kgm}^{-3}$ around the temperature of 241 K are orders of magnitude lower. I attach a plot of the Johnsen et al diffusivity for a range of temperatures. What is the reason for such a large difference?

**Fig. 1.** Diffusivity in firn for O18 at $\rho = 350\mathrm{kgm}^{-3}$

P23l530

It should be mentioned here that the van der Wel (Van der Wel et al., 2015) study is made by spraying a layer of isotopically spiked artificial snow on top of the natureal Summit snow. Such experiments are extremely challening and the approach of using artificial snow can pottentially introduce artifacts with respect to the diffusion processes.

P23l548

Perhaps you can slightly rephrase as "..are required to observe significant change in densities due to vapor transport at the seasonal cycle".

P24-25

The numbers of the experiments should be stated for clarity in the subsection titles or very soon after in the main body of each subsection.

P25l585

It is a little bit unclear here why and how the precipitation intermittency results in a biasing of the isotopic signal (from -53.2 ‰ to -49.8 ‰). I can see how the winter precipitation events are biased towards warmer temperatures and more enriched $\delta^{18}O$ values but cannot understand how this creates an additinaly bias in the isotopic composition of the snow.

**GMDD**

P25l590

"As expected the maxima and the minima of $\delta^{18}O$ are further reduced as a result..."
A more precise and careful writing would be very much appreciated. What does this
sentence mean? Is this a decrease of the whole $\delta^{18}O$ signal, a decrease in the peak
to peak amplitude of the signal or a decrease in only the minimum and the maximum
of the sinal? Additionaly (see also general comments) you can technically not have
isotopic diffusion because of temperature gradients. The latter can indeed create water
vapor concentration gradients that will result in diffusive transport of all water vapor
molecules. This is though not the same process as isotope diffusion.

P26l600

It is very difficult for the reader to follow the discussion of this paragraph when no ac-
cess is given to the $\delta^{18}O$ profiles pre and after diffusion. The approach of using contour
plots or tracking single layers does not give a good picture of the initial conditions and
the evolution of the simulation experiments. Even when those plots are presented they
only cover the top 40-50 cm of the studied snow-firn column. As a result, referring to
gradients of for example 24 ‰/m feels as an irrelevant piece of information.

P26l605

Indeed lower temperatures will slow down diffusive fluxes. This though can only be
modelled if the diffusivity coefficient is temperature dependent something that is not
the case for this study. Can you comment on this?

[Figure]

P26l607

Which other parameters are loosely estimated? When the term "large uncertainty" is used it is only logical for the reader to ask how large is the uncertainty.

P28l644

Replace badly with poorly.

P28l645

Being able to implement more processes in a model sounds in principle as a step forward. However I think that a discussion on improving on the knowledge, assumptions and parameters used in the more dominating processes of diffusion is missing here. Integration of more processes that are poorly implemented can be misleading and give the false impression of an improved approach for the description of the problem. With this in mind I think that a comment on proposed improvements, measurements and proper tests with real data would be most welcome in this manuscript especially if it focuses on the more dominating processes of the problem.

P28l660

The top 10 m of snow may have been modelled in this study but results only the top 0.5 m are presented here. Thus I think this sentence should be rephrased in order to reflect the actual results presented in the study.
P29l675

Refer to my general comments on the GRIP case.

**4   Comments on figures**

Figures of experiments results

The experiment number should be included in the captions and titles of all relevant figures.

Color maps of figures

The color maps of the density and $\delta^{18}$O change plots can become more readable if there is also some informationabout where the zero value is. I assume it is the white but cannot tell with certainty.

Density and O18 change plots

I find these plots confusing and not intuitive. The meaning of the term "density change" and "$\delta^{18}$O change" appears only in the caption of fig. 7 and 8. It is very hard for the reader to understand what this change refers to. My impression until I reached figure 7 and 8 was that these were rates ie change per time. Please clarify in the main text and on the legends of the figures.

Figure 11

It is odd that while the slope for the 2000 winter layer is oposite to the other summer layers and you choose to comment on this, the scale of the axis for these data is inverted thus visually "masking" the event. I would really not mind of the lines end up crossing each other if all axis are plotted in the same way.

**References**

C. Bréant, P. Martinerie, A. Orsi, L. Arnaud, and A. Landais. Modelling firn thickness evolution during the last deglaciation: constraints on sensitivity to temperature and impurities. *Clim. Past*, 13(7):833–853, July 2017.

M. D. Ellehoj, H. C. Steen-Larsen, S. J. Johnsen, and M. B. Madsen. Ice-vapor equilibrium fractionation factor of hydrogen and oxygen isotopes: Experimental investigations and implications for stable water isotope studies. *Rapid Commun. Mass Spectrom.*, 27(19):2149–2158, 2013.

S. J. Johnsen, H. B. Clausen, K. M. Cuffey, G. Hoffmann, J. Schwander, and T. Creyts. Diffusion of stable isotopes in polar firn and ice. the isotope effect in firn diffusion. In T. Hondoh, editor, *Physics of Ice Core Records*, pages 121–140, Sapporo, 2000. Hokkaido University Press.

W. Mook. *Environmental Isotopes in the Hydrological Cycle: Principles and Applications, vol. I, IAEA*. Unesco and IAEA, 2000.

M. S. Town, S. G. Warren, V. P. Walden, and E. D. Waddington. Effect of atmospheric water vapor on modification of stable isotopes in near-surface snow on ice sheets. *Journal of Geophysical Research-atmospheres*, 113:D24303, December 2008.

L. G. van der Wel, H. A. Been, R. S. W. van de Wal, C. J. P. P. Smeets, and H. A. J. Meijer. Constraints on the dh diffusion rate in firn from field measurements at summit, greenland. *The Cryosphere*, 9(3):1089–1103, May 2015.

---

## Author Comment (AC1) · 30 Jan 2018

Note that the pdf supplement shows 1) referee comments in violet, 2) our answers in black, and 3) our modifications to the manuscript in blue. The pdf version of this document is therefore easier to read than the plain text. Moreover the pdf file includes figures and equations absent from the plain text document.

Answer to Anonymous Referee 1 (plain text):
Touzeau et al. presents a detailed study in implementing isotopes into a semi-complex one-dimensional snow pack model. Unfortunately it is my opinion that the authors still need a little bit more work to allow this publication to become a significant contribution to the community. I am though positive that the manuscript will be publishable after my major comments have been taken into account.

Major comments:

(The following list of comments are not ordered in accordance with importance as they are more or less equally important)

- The use of parentheses throughout the manuscript is not in accordance with good practice. It makes reading the manuscript difficult. Please rewrite relevant sentences.

Most of the parentheses will be removed in the revised manuscript.

- The term 'oriented vapor transport' seems to complicate the reading. The model has already been defined as 1D and hence no need to include the word 'oriented'. Please remove throughout paper.

We used the term 'oriented vapor transport' to stress that vapor diffusion was not only driven by isotope gradients but also by temperature gradients. Diffusion induced by temperature gradient do not lead to homogeneous repartition of isotopes in contrast to the diffusion along isotopic gradient and this was the reason why we chose the term "oriented". We agree that this may not be obvious, and we have thus replaced 'oriented vapor transport' by 'thermally induced vapor transport' in the manuscript.

- 'Vapor density gradients'. Please change to 'vapor pressure gradients' throughout the paper. The use of vapor pressure is the normal term used i.e Merlivat and Jouzel 1979 and Jouzel and Merlivat 1984 etc.

"Density" was indeed probably not the best term (see also reviewer 3 comments). Be-

cause the unit of this term is kg.m-3, we have chosen to use the term "concentration" as suggested by reviewer 3.

- If a sentence is longer than 2 lines, it is most likely too long. Please refrain from using extremely long sentence that complicates the understanding of the manuscript. This is seen at several instances through out the manuscript, but my favorite example is section 2.1 L106-109 where I really have no idea what is being described.

We have rephrased the introduction of Section 2.1. using shorter sentences. l. 110-114: 'Here we describe first processes leading only to attenuation of the original amplitude (Sect. 2.1.1.). Then we describe processes which lead to other types of signal modifications (Sect. 2.1.2.). These modifications include transporting and accumulating isotopes in some layers without consideration of the original isotopic signal. They also comprise processes taking isotopes away from the snow, and therefore shifting the mean $\delta$18O value of the snow deposited.'

- Rephrase 'mean local pluriannual value' or describe what you mean.

Here we define 'mean local pluriannual value' as the average isotopic composition in the precipitation taken over several years ($\sim$10 years). This value averages seasonal variations and synoptic variations in the precipitation. It may be different from the average value in the snow layers that corresponds to the same period of time due to post-deposition processes.

- Rephrase 'oriented processes' or describe what you mean

Here we mean dynamical processes of vapor transport that are forced by atmospheric pressure or temperature variations. We used the term 'oriented' in opposition to 'random', in the sense 'forced' or 'pushed' or 'driven'. Maybe we should have said instead 'orienting' processes, as it is the vapor molecules which get 'oriented', not the processes themselves.

We propose to keep the term 'oriented' for the water molecules themselves, and to

replace 'oriented processes' by 'processes leading to oriented vapor transport'. We also add a line in the text to stress that 'oriented' is used in opposition to 'random agitation', and not in the sense of 'unidimensional' or 'vertical'. l. 137-139: 'We use the term 'oriented' here to describe an overall movement of water molecules that is different from their molecular agitation, and externally forced.'

- In L113 you write "Indeed, higher temperatures correspond to higher vapor densities, and also higher diffusivities in the vapor and the solid phase". This is correct, but then you line 260 define the vapor diffusivity in air to be a constant despite that it is depending on both temperature and pressure. This needs to be corrected. You need to allow for a temperature and pressure dependence on the diffusivity.

The reviewer is perfectly right. We have run the two main simulations again with varying Dv0 (function of atmospheric air pressure and snow temperature using the formula of Johnsen et al., 2000), and found some differences in the attenuation compared to the initial simulations. For the 10 years simulation at Dome C, the attenuation increases by 2-5%, and for the 10 years simulation at GRIP (with fixed temperature) it increases by 9-16%. Therefore we will replace the corresponding figures in the manuscript by the new ones, and modify the values of attenuation given in the text.

New Figure 2. Simulation 1: 10 years at GRIP with fixed temperature (240 K), with Dv0 function of the temperature. (Figure 1.png)

New Figure 11: Simulation 6: 10 years at Dome C with precipitation with varying $\delta$18O; with temperature evolution throughout the year; with Dv0 function of temperature. (Figure 2.png)

- I have a problem with your first sentence in the introduction "Ice is a key archive for past climate reconstruction, which preserves . . . indications relevant to the temperature of formation of the snow precipitation. . . variations of the isotopic ratio of oxygen and deuterium". This sentence is problematic because you have co-authors who have published papers documenting in both Greenland and Antarctica how the

isotopic composition of the deposited precipitation is changed through exchange with the atmospheric water vapor isotopes. You cite 8 publications to document your statement, but they are between 10 and 30 years old. You thereby disregard published research for the last five years. Please update.

We do not see a contradiction here, as a climatic signal may persist even after post-deposition processes have occurred. Therefore information regarding temperature may still be present, even if exchange with vapor isotope has taken place. Nevertheless, we will update the bibliography and soften these statements.

l. 25: 'The isotopic ratios of oxygen or deuterium measured in ice cores have been used for a long time to reconstruct the evolution of temperature over the Quaternary (EPICA comm. members, 2004; Johnsen et al., 1995; Jouzel et al., 2007; Kawamura et al., 2007; Uemura et al., 2012; Lorius et al., 1985; Petit et al., 1999; Schneider et al., 2006; Stenni et al., 2004; WAIS-Divide members, 2013; Stenni et al., 2011). They are however subject to alteration during post-deposition through various processes. As a consequence, even if the link between temperature and isotopic composition of the precipitations is quantitatively determined from measurements and modelling studies (Stenni et al., 2016; Goursaud et al., TCD, 2017), it cannot faithfully be applied to reconstruction of past temperature.'

- In L 17: Why not study the influence of temperature and not only temperature gradients? What is the difference between "compaction" and "Wind compaction".? Do you study the effect of amount of precipitation or the isotopes of the precipitation?

Physically, higher temperatures lead to increased diffusion through increased molecular agitation and also through increased vapor content in the air. In the first case, the control is a power function, while in the second case the control is exponential. Thus, we considered in a first approximation that molecular agitation was of second order and could be neglected. Still, in this new version, we will also consider the direct influence of temperature since the dependency of diffusivity on temperature is added.

There are two possible types of compaction implemented in the model (see Vionnet et al., 2012, for more details): Compaction caused by the weight of overlying layers ("compaction"), Compaction caused by wind reworking of the snow, which leads to increased density in the top layers (Âń wind compaction Âż).

We did not study specifically the effect of precipitation amount, as we used only one set of precipitation data coming from ERA-Interim. We did not vary this parameter to see how diffusion would be modified but it would be easy for future users to make such a study with the available code. Still, over the course of 10 years, variability of the precipitation amounts did occur. We followed 48 layers which were maintained for one year at least, and up to 10 years. For these layers, the thickness was ranging from 3 mm to 2.5 cm, and the slope was ranging from -0.137 to +0.133 ‰10 years. Based on these layers, the slope does not seem to be related to the layer thickness. However, it appears that the slope is related to the original $\delta$18O value in the layer.

Figure 3 and 4 .png

Regarding the isotopic composition in the precipitation we have run a zero-simulation with constant $\delta$18O in the precipitation. We wanted to see how vapor transport could possibly generate $\delta$18O variations, based on temperature gradients, in the absence of initial signal (Figure 6 and Figure 8). For the first layer, the $\delta$18O changes by about 1‰ in one year, whereas for the deeper layers, the change is about 0.1‰ during the same period. We then used the air temperature to compute $\delta$18O variations in the precipitation, to evaluate attenuation based on a realistic $\delta$18O signal (Figure 10 and Figure 11).

- L 52: Use another word than "Mechanical shuffling"

We replaced this term by "mechanical reworking".

- L119: You write that the annual cycles generally disappear at sites with accumulation lower than 200 kg/mȨ̈2/year – but does that not depend on time scales – please be

more precise.

It is true that thinning will also have an effect on the disappearance of annual cycles at deep depths. We will thus modify the statement saying that annual cycles disappeared at shallower depths (100 m deep) for sites with accumulation lower than 200 kg/m2/year.

l. 128: 'In Greenland, Johnsen et al. (1977) indicate that annual cycles generally disappear at depths shallower than 100 m for sites with accumulation lower than 200 kg m-2 yr-1.'

- L120: You write that the diffusion is more intense in the upper layers – but don't the diffusion depend on the isotopic gradient and would you not expect that to be larger further down in the snow? Please be precise! Also the word 'intense' might not be the best to use in this case

Indeed, theoretically, if diffusion was initially very low, and no other processes were active, the effect of compaction could increase $\delta$18O gradients downward by reducing layer thicknesses. In that case, the diffusion based on isotopic gradients would indeed increase downward.

Our model is indeed able to study such effect. It may be the purpose of a future application through a much longer run of the model than those presented here. Our aim here was to take the diffusion effect from the beginning, i.e. from the upper layer where porosity is large and temperature gradient huge hence enabling a strong diffusion. This will be clearly written in the revised version since it was not clear enough here.

- Section 3.1.2: Describe why the new vapor transport subroutine is inserted after module 5 but before module 6? What are the thoughts behind this?

The steps of the model first describe changes in the snow structure and microstructure (new layers, densification, metamorphism, wind drift) and later the energy exchanges. Because vapor diffusion is closely associated with metamorphism, and lead to changes

in the layer density, it seems natural to put it within this first series of modules that describe snow structure. Furthermore, its effect on the temperature profile is probably limited.

- L251: ". . .is the effective diffusivity of water vapor in the snow at the interface". Do you mean effective diffusivity of water vapor in the air between the snow grains?

There is a first step where we indeed compute "effective diffusivity" for each layer (from diffusivity in air and taking into account the size of the porosity, Equation (5)). (D_eff (t,n))/D_v =3/2 (1-(_sn (t,n))/_ice )-1/2 (5)

Then, what we name "interfacial diffusivity" (Deff(t,n→n+1)) is the average of two "effective diffusivities" from two adjacent layers (Deff(t,n) and Deff(t,n+1), Equation (6)). The "interface" here is the limit between the two layers.

D_eff (t,n→n+1)= 1/(1/(D_eff (t,n) )+1/(D_eff (t,n+1) )) (6)

This explanation is already present in the text.

l. 247: "flux of vapor at the interface between two layers"

l. 254: "The effective diffusivity at the interface is obtained in two steps: first the effective diffusivities (Deff(t,n) and Deff(t,n+1)) in each layer are calculated (Eq. (5)), second, the interfacial diffusivity is computed as their harmonic mean (Eq. (6)). "

To facilitate reading, we will add an indication line 252: "and Deff(t, n->n+1) (m2 s-1) is the effective diffusivity of water vapor in the snow at the interface between layers (see below)."

- Equation 6: I am not sure, but isn't a layer thickness missing from this formula as you might not have the same layer thickness in layer n and n+1?

Assessing interfacial effective transport properties in the case where layer thicknesses are different is a classical, yet, critical issue (e. g. D'Amboise et al., 2017 GMD), especially if the contrast in layer thickness is too large. Here we ensure that the contrast

in layer thickness remains as small as possible to limit the impact of this effect, and under such a situation we make the simplifying assumption that the interfacial diffusivity depends equally on the values of the two layers concerned.

- Equation 7: Why do you use an analytical approximation of Clausius-Clapeyron around zero and not a more precise empirical formula?

We are not aware that this formulation would provide worse results that empirical formulae.

- L 313 : "Long time" – what do you mean – please be precise

Original text: 'Equilibrium fractionation is a hypothesis that is correct in layers where the air has been standing still for a long time in the porosity and where vapor has reached equilibrium with ice grains, physically and chemically.'

We implied here that the equilibrium fractionation hypothesis was a reasonable hypothesis in our case. Indeed, the equilibrium situation is limited by the water vapor - snow mass transfer whose associated speed is of the order of 0.09 m.s-1 (Albert and McGilvary, 1992). In our case, we are dealing with centimetric scale layers thickness and recalculate the isotopic composition every second so that we consider that the speed of the mass transfer is not limiting the equilibrium situation at the water vapor - snow interface.

We have thus reformulated the text accordingly:

l. 329: 'Equilibrium fractionation is a hypothesis that is correct in layers where vapor has reached equilibrium with ice grains, physically and chemically. This process is limited by the water vapor - snow mass transfer whose associated speed is of the order of 0.09 m.s-1 (Albert and McGilvary, 1992). In our case, we are dealing with centimetric scale layers thickness and recalculate the isotopic composition every second so that we consider that the speed of the mass transfer is not limiting the equilibrium situation at the water vapor - snow interface.'

- L334: What vapor are you referencing to? H2O in general or H216O?.

Here we refer to H2O. We propose to add this precision in the text:

l. 353: 'When the vapor concentration is the same in two adjacent layers, the total flux of vapor is null. But we still have isotopic diffusion because of the isotopic concentration gradients (Eq. (13)), as long as they are non-zero.'

- L335: I believe you meant to write "we will still have diffusion of heavy water isotopes during conditions where the water isotopic gradient is non-zero.

This is very close to our meaning yes. We forgot to mention that in that case, diffusion is driven by isotopic gradients, only if they themselves are non-zero. Thanks for this precision. However, both heavy and light isotopes will diffuse. Therefore, we propose this correction:

l. 356: "But we still have isotopic diffusion because of the isotopic concentration gradients (Eq. (13)), as long as they are non-zero."

- L335-336: The sentence is very convoluted. I believe you could also have zero flux of H216O but a flux of H218O in one direction and HD16O in another direction.

We will remove this sentence, to simplify the reading.

- L353: "Here the condensation of excess vapor occurs without additional fractionation". Why do you make this assumption? Whenever you have a phase change due to condensation you will have isotopic fractionation. I think this is something that needs to be updated in your code.

We take this fractionation into account earlier in the model. We define our interstitial vapor as being at equilibrium with the solid phase (all the time) due to permanent sublimation/condensation in the porosity. This is why we write "without any additional fractionation". We do not want to apply this fractionation twice.

Kinetic fractionation due to supersaturation is also taken into account during the diffusion of the different isotopes, each with their associated diffusion coefficient.

Still, we understand that this aspect was not very clear in the initial manuscript and propose the following revision:

l.370: "Here the condensation of excess vapor occurs without additional fractionation because (1) there is a permanent isotopic equilibrium between surface snow and interstitial vapor (each first step of the sub-routine) and (2) kinetic fractionation associated with diffusion is taken into account during diffusion of the different isotopic species along the isotopic gradients"

- L356: "The transfer of isotopes takes place from the grain surface toward the vapor without fractionation" If you assume this then the interstitial vapor will not be in isotopic equilibrium with the snow surface. This would then correct itself. Hence I think that your code needs to be set-up such that the interstitial vapor is in isotopic equilibrium with the snow surface at all time.

Yes, temporarily, after this sublimation the vapor is no longer at equilibrium with the solid phase. But this is corrected immediately, as both are merged again before the next step (each step has a duration of one second). At the beginning of the next step, vapor isotopic composition is defined again at equilibrium with snow surface.

It is mathematically difficult to predict the composition of the sublimated vapor needed to have equilibrium in the end, and much easier to merge the two compartments and recreate later an equilibrium.

- Please note that you throughout the paper are mixing up GRIP and Summit. They are two different geographical places in Greenland albeit being close to each other.

We are sorry for this mixing, this will be corrected. Still, the climatic characteristics of these neighbour two sites are very similar so that this does not affect the results presented here.

- I am surprised to read that there are no density measurements for neither GRIP nor

Summit and that you therefore use NGRIP. Please double-check this.

Indeed, GRIP density measurements are available as listed in Bréant et al. (2017) and reference therein ( http://gcmd.nasa.gov/r/d/LSSU_PSU_Firn_data and Schwander et al., 1997; Iizuka et al., 2008). The density profile is close to the NGRIP profile. We ran the model with the correct density profile and found that the new profile did not change the results. Still, the new version will include the correct data.

- You do not give a relationship for the isotope-temperature relationship for GRIP. Please correct.

This is because the simulations at GRIP do not include precipitation, so the isotopic composition in the precipitation (and its relationship to temperature) is not useful here. We have added a sentence in the text to clarify this point:

L. 466: "following Eq. (15) to link $\delta$18O in the snowfall to the local temperature (Tair, in K): $\delta$ãĂŰ(_ ^18)OãĂŮ_sf=0.45$\times$(T_air-273.15)-31.5 (15)

We do not provide an equivalent expression for GRIP, Greenland, because the simulations run here (see Sect. 3.1.1) do not include precipitation."

- Figure 2: You should include a comparison with the model of Johnsen et al. 2000

New figure 4 with the model of Johnsen et al. (2000). (Figure 5.png)

We have added a curve (GRIP-J2000 model) on this figure corresponding to the model of Johnsen et al. (2000) for GRIP. We have used their equation 4 (amplitude as a function of diffusion length $\sigma$ and wavelength $\lambda$) as well as Figure 2 for the evolution of diffusion length with depth. We then obtained the wavelength evolution with depth on the Eurocore data by detection of maxima and minima.

- Figure 3: You write in the manuscript that the temperature is varying but on the figure you only show temperatures for the summer. Does this mean that you only use summer temperatures? I would expect you would use varying temperatures through the whole

[Figure]

year.

The temperature indeed varies the whole year but we have chosen to show only one temperature profile per year, to limit the number of curves on the graph. We chose January, because we considered that this month was one of the warmest, and likely to produce strong temperature gradients and strong vapor diffusion.

We will include a figure showing weekly temperature evolution in the Supplement. We will also add a note in the Figure 3 caption to clarify this point.

Figure 6.png

l. 1102: "(a) Vertical temperature profile for each summer; (b) $\delta 18Ogcenter$ profile for each summer; (c) Deviation of the $\delta 18O$ relative to the original profile, for each summer; (d) Evolution of the deviation to the original profile of $\delta 18Ogcenter$. Note that temperature varies during the whole year (see Figure TT in the Supplement)."

- I am surprised to find that your model does not show an influence of temperature gradients at GRIP as you would normally assume that temperature gradients would force vapor to be transported between layers due to the vapor pressure gradient?

There is indeed a small effect of temperature gradient at GRIP. This can be seen on the two figures 2 and 3. When temperature gradients are active, attenuation is stronger in upper layers, while under constant temperature, the attenuation is the same at 15 cm depth and at 70 cm depth. Quantitatively there is also an increase of attenuation in Figure 3 (from 5.021 10-1 to 7.567 10-1). Thus temperature gradients enhance diffusion at this site. However, this increase is small, and does not bridge the gap with the data. We state:

l. 495: "In conclusion, at GRIP, the diffusion of vapor as a result of temperature gradients has only a limited impact on isotopic compositions, and most of the simulated attenuation can be attributed to diffusion against isotopic gradients."

- L503: Is the attenuation at GRIP significant larger than NEEM? 86% and 90% seems

very similar. The reviewer is right, we will replace "greater" by "slightly higher" in the text.

- L511: Why don't you calculate the attenuation using Johnsen at GRIP such that you can compare with Bolzan and Pohjola?

A comparison with the Johnsen model will be included in the revised version (cf. comment above)

- L526: It is unclear how Denux in 1996 can indicate that a study by Johnsen et al. in 2000 overestimates the attenuation. Time travel hasn't really been possible yet. You might write that "A study by Denux (1996). . ."

Sorry for that, of course he was referring to the study published by Johnsen in 1977, and dealing with the same model. We have corrected the error:

l. 561: 'Denux (1996) and van der Wel et al. (2015) indicate that the model developed by Johnsen (1977) and used in Johnsen et al. (2000) overestimates the attenuation compared to observed values. For Denux (1996), the model of Johnsen (1977) should take into account the presence of ice crusts and the temperature gradients in the surface snow to get. . .'

- L528: You write that Johnsen et al. should take into considerations temperature gradients in order to not overestimate the attenuation. But would you not expect that temperature gradients would increase the attenuation due the vapor transport driven by vapor pressure gradients?

It is not clear yet if including temperature gradients would indeed increase the attenuation of the isotopic signal. This process might move the signal downward or upward without altering it much. It could also produce local isotope accumulation originally not present in the signal (see Figure 6). By creating these local isotope maxima the original signal could in the end 'gain' variability, instead of being smoothed. However, the presence of ice crusts proposed in Denux (1996) is a more straightforward explanation,

and should be tested first.

It is also possible that the discrepancy come from the 'isotopic diffusivity' used by Johnsen et al. (2000), which oversimplify a series of processes into one single equation. Introducing temperature gradients would necessarily imply a rewriting of this equation which might be the occasion to make the model more detailed and accurate.

We will slightly modify our sentence to enlighten which explanation is the most likely:

l. 564: 'For Denux (1996), the model of Johnsen (1977) should take into account the presence of ice crusts, and maybe also the temperature gradients in the surface snow, to get closer to the real attenuation at remote Antarctic sites.'

- I strongly suggest that you set up an experiment with Crocus that allow you compare as closely as possible the simulated attenuation with the calculated attenuation using the model of Johnsen et al. 2000.

This was exactly the aim of section 3.3.1 where indeed, temperature gradient were removed. We have added the comparison of the attenuation from Johnsen model in the figure (see above).

- Section 4.2.1: I suggest to remove the detailed description of simulation of density at Dome C to a supplementary material as it influences the flow of the manuscript which should be focusing on the evolution of isotopes in the snow pack.

OK, this will be moved.

- L 604: You suggest that the higher diffusion at GRIP compared to Dome C could be explained by higher temperatures – but in line 260 you assume that the diffusivity is constant and not influenced by temperature.

This will be corrected in the revised version (see comments above on the dependency of the diffusivity on temperature and pressure).

- In general for all the figures you need to adjust the values for the color bar such that

you don't have too many digits. For example in Figure 2 the color bar should go from -0.6 to 0.6 and in figure 3 it should be -1.9 to 0.8.

The limits are computed automatically as the maximum and minimum values of the variable over the first 60 layers. These values are then used in the text as a point of comparison between the different simulations. If we choose/ascribe the limits, this comparison will not be possible anymore.

- Figure S1: Why not combine panel b, c, and d

We are not sure what the reviewer expects here. We can of course remove the blank spaces. However, if the reviewer was meaning to use only one window, then we prefer not to make the modification. With just one window, we will not be able to show all the information, because of the differences in horizontal scales. Especially, the very small shift caused by compaction on panel (c) would not be visible anymore.

Minor comments L14 "The isotopes . . . resolution" should not be in abstract OK

L16 "condensation is realized" – what does this mean

This sentence means that the vapor density is brought back to its initial value by condensing excess vapor or sublimating snow. This step thus corresponds to solid/vapor exchanges, after vapor transport. We propose the following correction:

"2) kinetic fractionation is applied during transport, and 3) vapor is condensed or snow is sublimated to compensate deviation to vapor pressure at saturation."

L21: "model underestimates" -> modeled attenuation due to diffusion is underestimated, or that other processes, such as ventilation influences attenuation

We have modified the text according to the reviewer suggestion.

L24-25: should be moved to conclusion

OK

L42: Randomness in the core stratigraphy -> stratigraphic noise

We have modified the text according to the reviewer suggestion.

L45: series of snow pits -> series of records from snowpits OK

L53: ice microstructure at solid state ->snow grains due to solid diffusion OK

L58-61: Cite Ebner et al. 2016 and 2017 OK

L87 Missing parenthesis after Brun et al. 2011 OK

L99: Quick survey-> brief overview OK

L118: Wavelength of what?

It was the wavelength of the seasonally periodic isotopic signal. However, the text has been modified and wavelength no longer appear.

L178: What do you mean by "Permanent cycles"

We mean that the snow grain is never fully stable, and always undergoes sublimation and condensation at its borders. Depending on the balance of these two processes, its size may increase or decrease. When the two effects are balanced its size is constant. However, even in that case, its isotopic composition is still subject to evolution as sublimation and condensation are both active.

The term "cycle" does not convey our meaning correctly, as both processes are active at the same time. We propose the following correction:

L.193: "Indeed each grain experiences continuous recycling through sublimation/condensation"

L184: to get an -> to obtain an OK

L185: Remove the content of the parenthesis. OK

L224: What does this mean: "and taken to compensate yearly accumulation

Sorry for this complicated formulation. When we apply compaction we decrease the height of the firn column, while keeping its mass constant. Its total density is thus increased. We do this to make space for the deposition of a new snow layer at the top while keeping the surface level constant.

Using an accumulation at Dome C of 0.001 kg m-2 per 15 min, and considering that total snow column (over 12 meters) weights about 4461 kg, the compaction rate is: 2.2 10-7 per 15 min. For a layer of 330 kg m-3, the density increase is: +7.4 10-5 kg m-3 per 15 min. Per year, the total accumulation would be 35 kg m-2 and the density change, for the selected layer would be +2.59 kg m-3.

L240: What about the influence of absorption of radiation energy in layers below the surface layer?

It increases the heat of the layer, and therefore its temperature.

L254: "Interface": Please be more precise on defining what interface you are referring to

We have added a sentence to define the interface between two layers.

l. 262: 'In this section, the term 'interface' is used for the horizontal surface of exchange between two consecutive layers. The flux of vapor at the interface between two layers is obtained using the Fick's law of diffusion (Eq. (4)):'

L258: "interpenetrate": What do you mean?

When two grains are strongly pressed one against the other, the boundary between them becomes flat, and the two grains are merged together to make only one grain. 'Interpenetration' is the step when their limits cross each other during the merging. If the pressure is not strong enough, the shape of the grain is not modified; they slide one upon another without merging.

Figure 7.png

L296: "that are" -> being OK

L304: Have you defined kinetic fractionation previously?

No. We have added a sentence to define kinetic fractionation in the Introduction. l. 151: 'It becomes the main process of vapor transport when air is stagnant in the porosity. During diffusion, lighter molecules move more quickly in the porosity, leading to kinetic fractionation of the various isotopologues (Barkan and Luz, 2007).'

EQ 12: typo in D_eff_n&n

Thanks, we have replaced the notation Deff,n&n+1 by the symbol used before D_eff (t,n→n+1), in order to keep homogeneous notations.

L486: "Amplitude decrease by -1.3 o/oo" – do you mean amplitude increase by 1.3 o/oo

No, we mean decrease (the amplitude is reduced because of attenuation).

We have corrected the text: l. 515: 'Over 10 years (2000-2009), the amplitude decreases by 1.3 ‰ which corresponds to a 8 % variation.'

Please also note the supplement to this comment:
https://www.geosci-model-dev-discuss.net/gmd-2017-217/gmd-2017-217-AC1-supplement.pdf

————————————————

[Figure]

Fig. 1.

[Figure]

**Fig. 2.**

**Slope vs. initial δ¹⁸O value**

*Slope (‰/10 years)*

*δ¹⁸O value (‰)*

**Fig. 3.**

**Slope vs. layer thickness**

*(scatter plot: Slope (‰/10 years) versus Layer thickness (m))*

**Fig. 4.**

[Figure]

**Fig. 5.**

**Fig. 6.**

[Figure]

[Figure]

[Figure]

**Fig. 7.**

---

## Author Comment (AC3) · 2 Feb 2018

A. Kerkweg kerkweg@uni-bonn.de

Dear authors, In my role as Executive editor of GMD, I would like to bring to your attention our Editorial version 1.1: http://www.geosci-model-dev.net/8/3487/2015/gmd-8-3487-2015.html This highlights some requirements of papers published in GMD, which is also available on the GMD website in the 'Manuscript Types' section: http://www.geoscientific-model-development.net/submission/ manuscript_types.html.

In particular, please note that for your paper, the following requirements have not been met in the Discussions paper:

• "The main paper must give the model name and version number (or other unique identifier) in the title."

• "If the model development relates to a single model then the model name and the version number must be included in the title of the paper. If the main intention of an article is to make a general (i.e. model independent) statement about the usefulness of a new development, but the usefulness is shown with the help of one specific model, the model name and version number must be stated in the title. The title could have a form such as, "Title outlining amazing generic advance: a case study with Model XXX (version Y)".

• "All papers must include a section, at the end of the paper, entitled 'Code availability'. Here, either instructions for obtaining the code, or the reasons why the code is not available should be clearly stated. It is preferred for the code to be uploaded as a supplement or to be made available at a data repository with an associated DOI (digital object identifier) for the exact model version described in the paper. Alternatively, for established models, there may be an existing means of accessing the code through a particular system. In this case, there must exist a means of permanently accessing the precise model version described in the paper. In some cases, authors may prefer to put models on their own website, or to act as a point of contact for obtaining the code. Given the impermanence of websites and email addresses, this is not encouraged, and authors should consider improving the availability with a more permanent arrangement. After the paper is accepted the model archive should be updated to include a link to the GMD paper."

Thus please add the models name (SURFEX/Crocus ?) and the version number to the title of your article. Additionally, it would be good if the explicit version described in this article would be archived in a permanent archive providing a DOI (e.g. Zenodo). Yours,

Astrid Kerkweg

///////////////////////////////////////////

We apologize for not including the model references in the article title. We will add the relevant information to the title:

l. 1: 'Numerical experiments on vapor diffusion in polar snow and firn and its impact on isotopes using the multi-layer energy balance model Crocus in SURFEX V8.0'

We will also update our code availability section. The model SURFEX is open-source and available online after free registration through the platform cnrm-game-meteo.fr. Therefore it is not necessary to provide a copy on Zenodo.

We have updated the code availability section:

l. 762: 'The code used in the manuscript is a development of the open source code for SURFEX/ISBA-Crocus model based on version V8.0, hosted on an open git repository at CNRM (https://opensource.umr-cnrm.fr/projects/surfex_git2). Before downloading the code, you must register as a user at https://opensource.umr-cnrm.fr/. You can then obtain the code used in the present study by downloading the revision tagged 'Touzeau_jan2018' of the branch touzeau_dev (last access: January 2018). The meteorological forcing required to perform the runs is available as a supplement.'

************ Interactive comment on Geosci. Model Dev. Discuss., https://doi.org/10.5194/gmd-2017-217, 2017.

 

Please also note the supplement to this comment:
https://www.geosci-model-dev-discuss.net/gmd-2017-217/gmd-2017-217-AC3-supplement.pdf
* * *
[Figure]

2017.

---

## Author Response (AR2)

Dear authors, In my role as Executive editor of GMD, I would like to bring to your attention our Editorial version 1.1: http://www.geosci-model-dev.net/8/3487/2015/gmd-8-3487-2015.html This highlights some requirements of papers published in GMD, which is also available on the GMD website in the 'Manuscript Types' section: http://www.geoscientific-model-development.net/submission/manuscript_types.html.

In particular, please note that for your paper, the following requirements have not been met in the Discussions paper:

• "The main paper must give the model name and version number (or other unique identifier) in the title."

• "If the model development relates to a single model then the model name and the version number must be included in the title of the paper. If the main intention of an article is to make a general (i.e. model independent) statement about the usefulness of a new development, but the usefulness is shown with the help of one specific model, the model name and version number must be stated in the title. The title could have a form such as, "Title outlining amazing generic advance: a case study with Model XXX (version Y)".

• "All papers must include a section, at the end of the paper, entitled 'Code availability'. Here, either instructions for obtaining the code, or the reasons why the code is not available should be clearly stated. It is preferred for the code to be uploaded as a supplement or to be made available at a data repository with an associated DOI (digital object identifier) for the exact model version described in the paper. Alternatively, for established models, there may be an existing means of accessing the code through a particular system. In this case, there must exist a means of permanently accessing the precise model version described in the paper. In some cases, authors may prefer to put models on their own website, or to act as a point of contact for obtaining the code. Given the impermanence of websites and email addresses, this is not encouraged, and authors should consider improving the availability with a more permanent arrangement. After the paper is accepted the model archive should be updated to include a link to the GMD paper."

Thus please add the models name (SURFEX/Crocus ?) and the version number to the title of your article. Additionally, it would be good if the explicit version described in this article would be archived in a permanent archive providing a DOI (e.g. Zenodo). Yours, Astrid Kerkweg

We apologize for not including the model references in the article title. We will add the relevant information to the title:

l. 1: 'Numerical experiments on **vapor diffusion** in polar snow and firn and **its impact on isotopes** using **the** multi-layer energy balance **model Crocus in SURFEX V8.0**'

We will also update our code availability section. The model SURFEX is open-source and available online after free registration through the platform cnrm-game-meteo.fr. Therefore, it is not necessary to provide a copy on Zenodo.

We have updated the code availability section:

l. 768: 'The code used in the manuscript is a **development of the open source code for SURFEX/ISBA-Crocus model based on version V8.0, hosted on an open git repository at CNRM (https://opensource.umr-cnrm.fr/projects/surfex_git2). Before downloading the code, you must register as a user at https://opensource.umr-cnrm.fr/. You can then obtain the code used in the present study by downloading the revision tagged 'Touzeau_jan2018' of the branch touzeau_dev (last access: January 2018).** The **meteorological forcing** required to perform the runs is available as a supplement.'

*********** **Interactive comment on Geosci. Model Dev. Discuss.,**
**https://doi.org/10.5194/gmd-2017-217, 2017.**

**Touzeau et al. presents a detailed study in implementing isotopes into a semi-complex one-dimensional snow pack model. Unfortunately it is my opinion that the authors still need a little bit more work to allow this publication to become a significant contribution to the community. I am though positive that the manuscript will be publishable after my major comments have been taken into account.**

**Major comments:**

**(The following list of comments are not ordered in accordance with importance as they are more or less equally important)**

**- The use of parentheses throughout the manuscript is not in accordance with good practice. It makes reading the manuscript difficult. Please rewrite relevant sentences.**

Most of the parentheses will be removed in the revised manuscript.

**- The term 'oriented vapor transport' seems to complicate the reading. The model has already been defined as 1D and hence no need to include the word 'oriented'. Please remove throughout paper.**

We used the term 'oriented vapor transport' to stress that vapor diffusion was not only driven by isotope gradients but also by temperature gradients. Diffusion induced by temperature gradient do not lead to homogeneous repartition of isotopes in contrast to the diffusion along isotopic gradient and this was the reason why we chose the term "oriented". We agree that this may not be obvious, and we have thus replaced 'oriented vapor transport' by 'thermally induced vapor transport' in the manuscript.

**- 'Vapor density gradients'. Please change to 'vapor pressure gradients' throughout the paper. The use of vapor pressure is the normal term used i.e Merlivat and Jouzel 1979 and Jouzel and Merlivat 1984 etc.**

"Density" was indeed probably not the best term (see also reviewer 3 comments). Because the unit of this term is $kg.m^{-3}$, we have chosen to use the term "concentration" as suggested by reviewer 3.

**- If a sentence is longer than 2 lines, it is most likely too long. Please refrain from using extremely long sentence that complicates the understanding of the manuscript. This is seen at several instances through out the manuscript, but my favorite example is section 2.1 L106-109 where I really have no idea what is being described.**

We have rephrased the introduction of Section 2.1. using shorter sentences.

l. 117-121: **'Here we describe first processes leading only to attenuation of the original amplitude (Sect. 2.1.1.). Then we describe processes which lead to other types of signal modifications (Sect. 2.1.2.). These modifications include transporting and accumulating isotopes in some layers without consideration of the original isotopic signal. They also comprise processes taking isotopes away from the snow, and therefore shifting the mean $\delta^{18}O$ value of the snow deposited.'**

**- Rephrase 'mean local pluriannual value' or describe what you mean.**

Here we define 'mean local pluriannual value' as the average isotopic composition in the precipitation taken over several years (~10 years). This value averages seasonal variations and synoptic variations in the precipitation. It may be different from the average value in the snow layers that corresponds to the same period of time due to post-deposition processes.

**- Rephrase 'oriented processes' or describe what you mean**

Here we mean dynamical processes of vapor transport that are forced by atmospheric pressure or temperature variations. We used the term 'oriented' in opposition to 'random', in the sense 'forced' or 'pushed' or 'driven'. Maybe we should have said instead 'orienting' processes, as it is the vapor molecules which get 'oriented', not the processes themselves.

We propose to keep the term 'oriented' for the water molecules themselves, and to replace 'oriented processes' by 'processes leading to oriented vapor transport'. We also add a line in the text to stress that 'oriented' is used in opposition to 'random agitation', and not in the sense of 'unidimensional' or 'vertical'.

l. 142-143: **'We use the term 'oriented' here to describe an overall movement of water molecules that is different from their molecular agitation, and externally forced.'**

**- In L113 you write "Indeed, higher temperatures correspond to higher vapor densities, and also higher diffusivities in the vapor and the solid phase". This is correct, but then you line 260 define the vapor diffusivity in air to be a constant despite that it is depending on both temperature and pressure. This needs to be corrected. You need to allow for a temperature and pressure dependence on the diffusivity.**

The reviewer is perfectly right. We have run the two main simulations again with varying Dv0 (function of atmospheric air pressure and snow temperature using the formula of Johnsen et al., 2000), and found some differences in the attenuation compared to the initial simulations. For the 10 years simulation at Dome C, the attenuation increases by 2-5%, and for the 10 years simulation at GRIP (with fixed temperature) it increases by 9-16%. Therefore, we will replace the corresponding figures in the manuscript by the new ones and modify the values of attenuation given in the text.

[Figure]

**New Figure 2. Simulation 1: 10 years at GRIP with fixed temperature (240 K), with D$_{v0}$ function of the temperature.**

[Figure]

**New Figure 11: Simulation 6: 10 years at Dome C with precipitation with varying δ$^{18}$O; with temperature evolution throughout the year; with D$_{v0}$ function of temperature.**

- I have a problem with your first sentence in the introduction "Ice is a key archive for past climate reconstruction, which preserves . . . indications relevant to the temperature of formation of the snow precipitation. . . variations of the isotopic ratio of oxygen and deuterium". This sentence is problematic because you have co-authors who have published papers documenting in both Greenland and Antarctica how the isotopic composition of the deposited precipitation is changed through exchange with the atmospheric water vapor isotopes. You cite 8 publications to document your statement, but they are between 10 and 30 years old. You thereby disregard published research for the last five years. Please update.

We do not see a contradiction here, as a climatic signal may persist even after post-deposition processes have occurred. Therefore, information regarding temperature may still be present, even if exchange with vapor isotope has taken place. Nevertheless, we will update the bibliography and soften these statements.

l. 28: '**The isotopic ratios of oxygen or deuterium measured in ice cores have been used for a long time to reconstruct the evolution of temperature over the Quaternary (EPICA comm. members, 2004; Johnsen et al., 1995; Jones et al., 2018; Jouzel et al., 2007; Kawamura et al., 2007; Lorius et al., 1985; Petit et al., 1999; Schneider et al., 2006; Stenni et al., 2004; Stenni et al., 2011; Uemura et al., 2012; WAIS-Divide members, 2013**). **They are however subject to alteration during post-deposition through various processes**. **As a consequence, even if the link between temperature and isotopic composition of the precipitations is quantitatively determined from measurements and modelling studies (Stenni et al., 2016; Goursaud et al., TCD, 2017), it cannot faithfully be applied to reconstruction of past temperature.**'

**- In L 17: Why not study the influence of temperature and not only temperature gradients? What is the difference between "compaction" and "Wind compaction".? Do you study the effect of amount of precipitation or the isotopes of the precipitation?**

• Physically, higher temperatures lead to increased diffusion through increased molecular agitation and also through increased vapor content in the air. In the first case, the control is a power function, while in the second case the control is exponential. Thus, we considered in a first approximation that molecular agitation was of second order and could be neglected. Still, in this new version, we will also consider the direct influence of temperature since the dependency of diffusivity on temperature is added.

• There are two possible types of compaction implemented in the model (see Vionnet et al., 2012, for more details):

1) Compaction caused by the weight of overlying layers ("compaction"),

**2)** Compaction caused by wind reworking of the snow, which leads to increased density in the top layers (« wind compaction »).

• We did not study specifically the effect of precipitation amount, as we used only one set of precipitation data coming from ERA-Interim. We did not vary this parameter to see how diffusion would be modified but it would be easy for future users to make such a study with the available code. Still, over the course of 10 years, variability of the precipitation amounts did occur. We followed 48 layers which were maintained for one year at least, and up to 10 years. For these layers, the thickness was ranging from 3 mm to 2.5 cm, and the slope was ranging from -0.137 to +0.133 ‰/10 years. Based on these layers, the slope does not seem to be related to the layer thickness. However, it appears that the slope is related to the original $\delta^{18}O$ value in the layer.

[Figure]

• Regarding the isotopic composition in the precipitation we have run a zero-simulation with constant $\delta^{18}O$ in the precipitation. We wanted to see how vapor transport could possibly generate $\delta^{18}O$ variations, based on temperature gradients, in the absence of initial signal (Figure 6 and Figure 8). For the first layer, the $\delta^{18}O$ changes by about 1‰ in one year, whereas for the deeper layers, the change is about 0.1‰ during the same period. We then used the air temperature to compute $\delta^{18}O$ variations in the precipitation, to evaluate attenuation based on a realistic $\delta^{18}O$ signal (Figure 10 and Figure 11).

**- L 52: Use another word than "Mechanical shuffling"**
We replaced this term by "mechanical reworking".

**- L119: You write that the annual cycles generally disappear at sites with accumulation lower than 200 kg/m^2/year – but does that not depend on time scales – please be more precise.**
It is true that thinning will also have an effect on the disappearance of annual cycles at deep depths. We will thus modify the statement saying that annual cycles disappeared at shallower depths (100 m deep) for sites with accumulation lower than 200 kg/m$^2$/year.
**l. 135: '**In Greenland, Johnsen et al. (1977) indicate that **annual cycles generally disappear at depths shallower than 100 m for sites with accumulation lower than 200 kg m$^{-2}$ yr$^{-1}$.'**

**- L120: You write that the diffusion is more intense in the upper layers – but don't the diffusion depend on the isotopic gradient and would you not expect that to be larger further down in the snow? Please be precise! Also the word 'intense' might not be the best to use in this case**

Indeed, theoretically, if diffusion was initially very low, and no other processes were active, the effect of compaction could increase $\delta^{18}O$ gradients downward by reducing layer thicknesses. In that case, the diffusion based on isotopic gradients would indeed increase downward. Our model is indeed able to study such effect. It may be the purpose of a future application through a much longer run of the model than those presented here.

Our aim here was to take the diffusion effect from the beginning, i.e. from the upper layer where porosity is large and temperature gradient huge hence enabling a strong diffusion. This will be clearly written in the revised version since it was not clear enough here.

l. 136: 'Diffusion **along** isotopic gradients exists throughout the entire snow/ice column. **It occurs mainly in the vapor phase in the firn, especially in the upper layers with larger porosities. After pore closure, it** takes place mostly in the solid phase, at a much slower rate.'

**- Section 3.1.2: Describe why the new vapor transport subroutine is inserted after module 5 but before module 6? What are the thoughts behind this?**

The steps of the model first describe changes in the snow structure and microstructure (new layers, densification, metamorphism, wind drift) and later the energy exchanges. Because vapor diffusion is closely associated with metamorphism, and lead to changes in the layer density, it seems natural to put it within this first series of modules that describe snow structure. Furthermore, its effect on the temperature profile is probably limited.

**- L251: ". . .is the effective diffusivity of water vapor in the snow at the interface". Do you mean effective diffusivity of water vapor in the air between the snow grains?**

There is a first step where we indeed compute "effective diffusivity" for each layer (from diffusivity in air and taking into account the size of the porosity, Equation (5)).

$$\frac{D_{eff}(t,n)}{D_v} = \frac{3}{2}\left(1 - \frac{\rho_{sn}(t,n)}{\rho_{ice}}\right) - \frac{1}{2} \tag{5}$$

Then, what we name "interfacial diffusivity" (Deff(t,n→n+1)) is the average of two "effective diffusivities" from two adjacent layers (Deff(t,n) and Deff(t,n+1), Equation (6)). The "interface" here is the limit between the two layers.

$$D_{eff}(t, n \to n+1) = \frac{1}{\frac{1}{D_{eff}(t,n)} + \frac{1}{D_{eff}(t,n+1)}} \tag{6}$$

This explanation is already present in the text.

l. 270: "flux of vapor at the interface between two layers"

l. 277: "The effective diffusivity at the interface is obtained in two steps: first the effective diffusivities (*Deff*(t,n) and *Deff*(t,n+1)) in each layer are calculated (Eq. (5)), second, the interfacial diffusivity is computed as their harmonic mean (Eq. (6)). "

To facilitate reading, we will add an indication line 270:

l. 269: '**In this section, the term 'interface' is used for the horizontal surface of exchange between two consecutive layers.**'

**- Equation 6: I am not sure, but isn't a layer thickness missing from this formula as you might not have the same layer thickness in layer n and n+1?**

Assessing interfacial effective transport properties in the case where layer thicknesses are different is a classical, yet, critical issue (e. g. D'Amboise et al., 2017 GMD), especially if the contrast in layer thickness is too large. Here we ensure that the contrast in layer thickness remains as small as possible to limit the impact of this effect, and under such a situation we make the simplifying assumption that the interfacial diffusivity depends equally on the values of the two layers concerned.

**- Equation 7: Why do you use an analytical approximation of Clausius-Clapeyron around zero and not a more precise empirical formula?**

We are not aware that this formulation would provide worse results that empirical formulae.

**- L 313 : "Long time" – what do you mean – please be precise**
Original text:
'Equilibrium fractionation is a hypothesis that is correct in layers where the air has been standing still for a long time in the porosity and where vapor has reached equilibrium with ice grains, physically and chemically.'

We implied here that the equilibrium fractionation hypothesis was a reasonable hypothesis in our case. Indeed, the equilibrium situation is limited by the water vapor - snow mass transfer whose associated speed is of the order of 0.09 m.s$^{-1}$ (Albert and McGilvary, 1992). In our case, we are dealing with centimetric scale layers thickness and recalculate the isotopic composition every second so that we consider that the speed of the mass transfer is not limiting the equilibrium situation at the water vapor - snow interface.

We have thus reformulated the text accordingly:

l. 336: 'Equilibrium fractionation is a hypothesis that is correct in layers where vapor has reached equilibrium with ice grains, physically and chemically. **This process is limited by the water vapor - snow mass transfer whose associated speed is of the order of 0.09 m.s-1 (Albert and McGilvary, 1992). In our case, we are dealing with centimetric scale layers thickness and recalculate the isotopic composition every second so that we consider that the speed of the mass transfer is not limiting the equilibrium situation at the water vapor - snow interface.**'

**- L334: What vapor are you referencing to? H2O in general or H216O?.**
Here we refer to H2O. We propose to add this precision in the text:
l. 3624: **'When the vapor concentration is the same in two adjacent layers, the total flux of vapor is null. But** we still have isotopic diffusion because of the isotopic concentration gradients **(Eq. (13)), as long as they are non-zero**.'

**- L335: I believe you meant to write "we will still have diffusion of heavy water isotopes during conditions where the water isotopic gradient is non-zero.**

This is very close to our meaning yes. We forgot to mention that in that case, diffusion is driven by isotopic gradients, only if they themselves are non-zero. Thanks for this precision. However, both heavy and light isotopes will diffuse. Therefore, we propose this correction:

l. 363: "But we still have isotopic diffusion because of the isotopic concentration gradients **(Eq. (13)), as long as they are non-zero.**"

**- L335-336: The sentence is very convoluted. I believe you could also have zero flux of H216O but a flux of H218O in one direction and HD16O in another direction.**

We will remove this sentence, to simplify the reading.

**- L353: "Here the condensation of excess vapor occurs without additional fractionation". Why do you make this assumption? Whenever you have a phase change due to condensation you will have isotopic fractionation. I think this is something that needs to be updated in your code.**

We take this fractionation into account **earlier** in the model. We define our interstitial vapor as being **at equilibrium** with the solid phase (all the time) due to permanent sublimation/condensation in the porosity. This is why we write "without any additional fractionation". We do not want to apply this fractionation twice.

Kinetic fractionation due to supersaturation is also taken into account during the diffusion of the different isotopes, each with their associated diffusion coefficient.

Still, we understand that this aspect was not very clear in the initial manuscript and propose the following revision:

l.380: "Here the condensation of excess vapor occurs without additional fractionation **because (1) there is a permanent isotopic equilibrium between surface snow and interstitial vapor (each first step of the sub-routine) and (2) kinetic fractionation associated with diffusion is taken into account during diffusion of the different isotopic species along the isotopic gradients**"

**- L356: "The transfer of isotopes takes place from the grain surface toward the vapor without fractionation" If you assume this then the interstitial vapor will not be in isotopic equilibrium with the snow surface. This would then correct itself. Hence I think that your code needs to be set-up such that the interstitial vapor is in isotopic equilibrium with the snow surface at all time.**

Yes, temporarily, after this sublimation the vapor is no longer at equilibrium with the solid phase. But this is corrected immediately, as both are merged again before the next step (each step has a duration of one second). At the beginning of the next step, vapor isotopic composition is defined again at equilibrium with snow surface.

It is mathematically difficult to predict the composition of the sublimated vapor needed to have equilibrium in the end, and much easier to merge the two compartments and recreate later an equilibrium.

**- Please note that you throughout the paper are mixing up GRIP and Summit. They are two different geographical places in Greenland albeit being close to each other.**

We are sorry for this mixing, this will be corrected. Still, the climatic characteristics of these neighbour two sites are very similar so that this does not affect the results presented here.

**- I am surprised to read that there are no density measurements for neither GRIP nor Summit and that you therefore use NGRIP. Please double-check this.**

Indeed, GRIP density measurements are available as listed in Bréant et al. (2017) and reference therein ( http://gcmd.nasa.gov/r/d/LSSU_PSU_Firn_data and Schwander et al., 1997; Iizuka et al., 2008). The density profile is close to the NGRIP profile. We ran the model with the correct density profile and found that the new profile did not change the results. Still, the new version will include the correct data.

**- You do not give a relationship for the isotope-temperature relationship for GRIP. Please correct.**

This is because the simulations at GRIP do not include precipitation, so the isotopic composition in the precipitation (and its relationship to temperature) is not useful here.

We have added a sentence in the text to clarify this point:

L. 481: "following Eq. (15) to link $\delta^{18}$O in the snowfall to the local temperature ($T_{air}$, in K):

$$\delta^{18}O_{sf} = 0.45 \times (T_{air} - 273.15) - 31.5$$

(16)

**We do not provide an equivalent expression for GRIP, Greenland, because the simulations run here (see Sect. 3.1.1) do not include precipitation."**

**- Figure 2: You should include a comparison with the model of Johnsen et al. 2000**

[Figure]

New figure 4 with the model of Johnsen et al. (2000).

We have added a curve (GRIP-J2000 model) on this figure corresponding to the model of Johnsen et al. (2000) for GRIP. We have used their equation 4 (amplitude as a function of *diffusion length* σ and *wavelength* λ) as well as Figure 2 for the evolution of diffusion length with depth. We then obtained the wavelength evolution with depth on the Eurocore data by detection of maxima and minima.

**- Figure 3: You write in the manuscript that the temperature is varying but on the figure you only show temperatures for the summer. Does this mean that you only use summer temperatures? I would expect you would use varying temperatures through the whole year.**

The temperature indeed varies the whole year but we have chosen to show only one temperature profile per year, to limit the number of curves on the graph. We chose January, because we considered that this month was one of the warmest, and likely to produce strong temperature gradients and strong vapor diffusion.

We will include a figure showing weekly temperature evolution in the Supplement. We will also add a note in the Figure 3 caption to clarify this point.

l. 1110: "(a) Vertical temperature profile for each summer; (b) $\delta^{18}O_{gcenter}$ profile for each summer; (c) Deviation of the $\delta^{18}O$ relative to the original profile, for each summer; (d) Evolution of the deviation to the original profile of $\delta^{18}O_{gcenter}$. **Note that temperature varies during the whole year (see Fig. S1)."**

[Figure]

**- I am surprised to find that your model does not show an influence of temperature gradients at GRIP as you would normally assume that temperature gradients would force vapor to be transported between layers due to the vapor pressure gradient?**

There is indeed an effect of temperature gradient at GRIP. This can be seen on the two figures 2 and 3. When temperature gradients are active, attenuation is stronger in upper layers, while under constant temperature, the attenuation is the same at 15 cm depth and at 70 cm depth. Moreover, thermally induced vapor diffusion does not only attenuate original sinusoidal variations, but also seems to accumulate heavy or light isotopes in certain layers. We have modified this section of the manuscript to better describe these two effects.

l. 529: 'Figure 3 shows the result of **Simulation 2**, i.e. with varying temperature in the snowpack. **The attenuation is stronger than the one observed in the previous simulation. The minima at 11.46 m increases by 1.03 ‰ over ten years, and the maxima at 11.15 m decreases by 0.84 ‰. Thus, the total attenuation for the range of heights is ~1.9 ‰. This corresponds to an attenuation by 11.7 %. For the layers below, the attenuation is smaller, with a total attenuation of only 6 % for heights between 10.54 and 10.85 m. If we compare attenuation for heights 11.46 and 11.56 m in the 1st and 2nd simulation, we note that including temperature gradients leads to an attenuation increased by half.**

**Between 11.46 m and 11.56 m, the $\delta^{18}O_{gcenter}$ values increase over ten years by 1 to 4 ‰. This increase is not caused only by attenuation of the original sinusoidal signal. Indeed, at h=11,60 m, the values get higher than the initial maxima which was -36 ‰ at 11.64 m. There is therefore a local accumulation of heavy isotopes in this**

**layer as a result of vapor transport. This maxima corresponds to a local maxima in temperature and is coherent with departure of $^{18}$O-depleted water vapor from this layer. Thus, thermally induced vapor transport does not only result into signal attenuation, but can also shift the δ$^{18}$O value, regardless of the initial sinusoidal variations.**

Lastly, in the first 2-3 cm of the snowpack**, strong** depletion is observed over the period, **with a decrease by 2 to 3 ‰ instead of 0.5 ‰ previously**. This depletion probably results from arrival of $^{18}$O-depleted water vapor from warmer layers below. **This shows again** the influence of temperature gradients which were absent from the previous simulation. However, **note that in this simulation we neglect precipitation** and exchange of vapor with the atmosphere**. Thus,** the depletion observed here may not occur **in natural settings** when these processes **are active.**

**In conclusion, at GRIP, the diffusion of vapor as a result of temperature gradients has a double impact on isotopic compositions. It increases the attenuation in the first 60 cm of snow, because of higher vapor fluxes. And it also creates local isotopic maximas and minimas, in a pattern corresponding to temperature gradients in the snowpack, but disconnected from the original δ$^{18}$O sinusoidal signal.'**

**- L503: Is the attenuation at GRIP significant larger than NEEM? 86% and 90% seems very similar.**

The reviewer is right, we will replace "greater" by "**slightly higher**" in the text.

**- L511: Why don't you calculate the attenuation using Johnsen at GRIP such that you can compare with Bolzan and Pohjola?**

A comparison with the Johnsen model will be included in the revised version (cf. comment above)

**- L526: It is unclear how Denux in 1996 can indicate that a study by Johnsen et al. in 2000 overestimates the attenuation. Time travel hasn't really been possible yet. You might write that "A study by Denux (1996). . ."**

Sorry for that, of course he was referring to the study published by Johnsen in 1977, and dealing with the same model. We have corrected the error:

l. 591: 'Denux (1996) and van der Wel et al. (2015) indicate that the model **developed by Johnsen (1977) and used in Johnsen et al. (2000)** overestimates the attenuation compared to observed values. For Denux (1996), the model of **Johnsen (1977)** should take into account the presence of ice crusts and the temperature gradients in the surface snow to get…'

**- L528: You write that Johnsen et al. should take into considerations temperature gradients in order to not overestimate the attenuation. But would you not expect that temperature gradients would increase the attenuation due the vapor transport driven by vapor pressure gradients?**

It is not clear yet if including temperature gradients would indeed increase the attenuation of the isotopic signal. This process might move the signal downward or upward without altering it much. It could also produce local isotope accumulation originally not present in the signal (see Figure 6). By creating these local isotope maxima the original signal could in the end 'gain' variability, instead of being smoothed.

However, the presence of ice crusts proposed in Denux (1996) is a more straightforward explanation, and should be tested first.

It is also possible that the discrepancy come from the 'isotopic diffusivity' used by Johnsen et al. (2000), which oversimplify a series of processes into one single equation. Introducing temperature gradients would necessarily imply a rewriting of this equation which might be the occasion to make the model more detailed and accurate.

We will slightly modify our sentence to enlighten which explanation is the most likely:

l. 592: 'For Denux (1996), the model of Johnsen (1977) should take into account the presence of ice crusts**, and maybe also** the temperature gradients in the **surface snow,** to get closer to the real attenuation at remote Antarctic sites.'

**- I strongly suggest that you set up an experiment with Crocus that allow you compare as closely as possible the simulated attenuation with the calculated attenuation using the model of Johnsen et al. 2000.**

This was exactly the aim of section 3.3.1 where indeed, temperature gradient were removed. We have added the comparison of the attenuation from Johnsen model in the figure (see above).

**- Section 4.2.1: I suggest to remove the detailed description of simulation of density at Dome C to a supplementary material as it influences the flow of the manuscript which should be focusing on the evolution of isotopes in the snow pack.**

OK, this will be moved.

**- L 604: You suggest that the higher diffusion at GRIP compared to Dome C could be explained by higher temperatures – but in line 260 you assume that the diffusivity is constant and not influenced by temperature.**

This will be corrected in the revised version (see comments above on the dependency of the diffusivity on temperature and pressure).

**- In general for all the figures you need to adjust the values for the color bar such that you don't have too many digits. For example in Figure 2 the color bar should go from -0.6 to 0.6 and in figure 3 it should be -1.9 to 0.8.**

The limits are computed automatically as the maximum and minimum values of the variable over the first 60 layers. These values are then used in the text as a point of comparison between the different simulations. If we choose/ascribe the limits, this comparison will not be possible anymore.

**- Figure S1: Why not combine panel b, c, and d**

We are not sure what the reviewer expects here. We can of course remove the blank spaces. However, if the reviewer was meaning to use only one window, then we prefer not to make the modification. With just one window, we will not be able to show all the information, because of the differences in horizontal scales. Especially, the very small shift caused by compaction on panel (c) would not be visible anymore.

**Minor comments**

**L14 "The isotopes . . . resolution" should not be in abstract**

OK

**L16 "condensation is realized" – what does this mean**

This sentence means that the vapor density is brought back to its initial value by condensing excess vapor or sublimating snow. This step thus corresponds to solid/vapor exchanges, after vapor transport. We propose the following correction:

**l. 17 "**2) a kinetic fractionation is applied during transport, and 3) **vapor is condensed or snow is sublimated to compensate deviation to vapor pressure at saturation."**

**L21: "model underestimates" -> modeled attenuation due to diffusion is underestimated, or that other processes, such as ventilation influences attenuation**

We have modified the text according to the reviewer suggestion.

**L24-25: should be moved to conclusion**

OK

**L42: Randomness in the core stratigraphy -> stratigraphic noise**

We have modified the text according to the reviewer suggestion.

**L45: series of snow pits -> series of records from snowpits**

OK

**L53: ice microstructure at solid state ->snow grains due to solid diffusion**

OK

**L58-61: Cite Ebner et al. 2016 and 2017**

OK

**L87 Missing parenthesis after Brun et al. 2011**

OK

**L99: Quick survey-> brief overview**

OK

**L118: Wavelength of what?**

It was the wavelength of the seasonally periodic isotopic signal. However, the text has been modified and wavelength no longer appear.

**L178: What do you mean by "Permanent cycles"**

We mean that the snow grain is never fully stable, and always undergoes sublimation and condensation at its borders. Depending on the balance of these two processes, its size may increase or decrease. When the two effects are balanced its size is constant. However, even in that case, its isotopic composition is still subject to evolution as sublimation and condensation are both active.

The term "cycle" does not convey our meaning correctly, as both processes are active at the same time. We propose the following correction:

L. 201: "Indeed each grain experiences **continuous recycling through** sublimation/condensation"

**L184: to get an -> to obtain an**
OK

**L185: Remove the content of the parenthesis.**
OK

**L224: What does this mean: "and taken to compensate yearly accumulation**
Sorry for this complicated formulation. When we apply compaction we decrease the height of the firn column, while keeping its mass constant. Its total density is thus increased. We do this to make space for the deposition of a new snow layer at the top while keeping the surface level constant.

Using an accumulation at Dome C of 0.001 kg m$^{-2}$ per 15 min, and considering that total snow column (over 12 meters) weights about 4461 kg, the compaction rate is: 2.2 10$^{-7}$ per 15 min. For a layer of 330 kg m$^{-3}$, the density increase is: +7.4 10-5 kg m$^{-3}$ per 15 min. Per year, the total accumulation would be 35 kg m$^{-2}$ and the density change, for the selected layer would be +2.59 kg m$^{-3}$.

**L240: What about the influence of absorption of radiation energy in layers below the surface layer?**
It increases the heat of the layer, and therefore its temperature.

**L254: "Interface": Please be more precise on defining what interface you are referring to**
We have added a sentence to define the interface between two layers.
**l. 269: 'In this section, the term 'interface' is used for the horizontal surface of exchange between two consecutive layers.** The flux of vapor at the interface between two layers is obtained using the Fick's law of diffusion (Eq. (4)):'

**L258: "interpenetrate": What do you mean?**
When two grains are strongly pressed one against the other, the boundary between them becomes flat, and the two grains are merged together to make only one grain. 'Interpenetration' is the step when their limits cross each other during the merging. If the pressure is not strong enough, the shape of the grain is not modified; they slide one upon another without merging.

[Figure]

[Figure]

We have rephrased the sentence to facilitate understanding:

l. 279: 'Effective diffusivity can be expressed as a function of the snow density using the relationship proposed by Calonne et al. (2014), for layers **with relatively low density. In these circumstances**, the compaction occurs by 'boundary sliding', meaning that the grains slide on each **other, but that their shape is not modified. It is therefore applicable to our study** where density is always below 600 kg m$^{-3}$.

**L296: "that are" -> being**
OK

**L304: Have you defined kinetic fractionation previously?**
No. We have added a sentence to define kinetic fractionation in the Introduction.
**l. 156:** 'It becomes the main process of vapor transport when air is stagnant in the porosity. **During diffusion, lighter molecules move more quickly in the porosity, leading to kinetic fractionation of the various isotopologues.**'

**EQ 12: typo in D_eff_n&n**
Thanks, we have replaced the notation $D_{eff,n\&n+1}$ by the symbol used before $D_{eff}(t, n \rightarrow n + 1)$, in order to keep homogeneous notations.

**L486: "Amplitude decrease by -1.3 o/oo" – do you mean amplitude increase by 1.3 o/oo**
No, we mean decrease (the amplitude is reduced because of attenuation).
We have corrected the text:
l. 527: 'Over 10 years (2000-2009), the amplitude decreases **by 1.2 ‰** which corresponds to a 7.3 % variation.'

**Interactive comment on Geosci. Model Dev. Discuss., https://doi.org/10.5194/gmd-2017-217,**

**2017.**

**The post-depositional modification is an important but poorly understood part of the "isotopic paleo-thermometer". After the solid precipitation is deposited on the top of the polar ice sheet snow surface, its isotopic content is changed drastically due to the water and mass exchange with the atmospheric water vapor and due to molecular diffusion in snow. These processes disturb or even completely erase the initial climatic signal recorded in the isotopic content of the precipitation. To solve this problem, different approaches are applied including modeling of the snow pack evolution during snow metamorphism. This manuscript is an attempt to simulate the snow isotopic content of the polar snow in the course of the post-depositional processes. For the first time the snow-pack Crocus model is applied for this purpose. The authors clearly understand that this work is a small step towards the full description of the isotopic post-depositional modifications. A lot of efforts still has to be done. However, this attempt deserves to be published as a separate paper in "Geoscientific Model Developement" journal.**

**The manuscript is nicely structured and provides a good review of literature on the formation of the climatic signal in the snow isotopic composition. The authors make a nice attempt to describe in simple way a rather complicated process of the isotopic modifications in the snow thickness. I do not have major corrections, only a few minor comments or questions:**

**In your model you do not take into account the mechanical snow mixing by wind. This mixing erases the initial climatic signal (shorter than few years) in central Antarctica, and makes the vertical isotopic profiles in the upper part of snow thickness similar to white noise. Recent study by Thomas Laepple (https://www.the-cryospherediscuss.net/tc-2017-199/) showed that the filtering of this noise by isotopic diffusion can create false cycles in the isotopic profiles. So, I suggest that in the further versions of you model you introduce random component of the initial isotopic composition of the precipitation (or of the upper snow layer if you wish) in parallel to the regular component given by precipitation events. You might mention this in section 4.4. and conclusion.**

We agree with the reviewer that wind mixing should be included in the model somehow as it is an important process in Antarctica.

Libois et al. (2014) already paved the way to do it. They proposed to run the Crocus model in parallel (50 snow columns for the same site) and to exchange snow between these columns. This method is called 'stochastic snow redistribution scheme'. However, because the vapor transport scheme is run at a 1 s time-step, it is slowing down the model. Thus, running 50 simulations in parallel might be very time-consuming in our case.

Other solutions could be proposed, such as taking the first centimeters of snow away (wind ablation), store it temporarily in an atmospheric reservoir and letting it fall again. If several layers are eroded, they could fall down in a different order, or maybe be mixed together while still in the atmosphere. We will also consider the reviewer suggestion, to add a random component to the signal in precipitation. This could be the simplest way to simulate this process.

We have modified the text in two places to stress the importance of this process to be taken into account in future work:

l. 717: 'The next step for Crocus-iso development is thus to implement ventilation. **Finally, we are also aware that in Antarctic central regions, the wind reworking of the snow has a strong effect in shaping the isotopic signal. A combination of stratigraphic noise and diffusion could indeed be responsible for creating isotopic cycles of non-climatic origin in the firn (Laepple et al., 2017). Wind reworking may also contribute to attenuation, by mixing together several layers deposited during different seasons.'**

L. 753: 'Second, in low accumulation sites like Dome C, wind scouring has probably an important effect on the evolution of the $\delta^{18}$O signal in depth through a reworking of the top snow layers (Libois et al., 2014). This effect has not been considered here **but could be implemented in the model in the next years.'**

**Other minor comments:**
**line 33 - better to write "1950s"**
OK

**lines 352-353 - why condensation is without additional fractionation?**
See above (same comment by the first reviewer).

**lines 502-503: the values (86% and 90%) are the remaining amplitudes, right?**
Yes, we made a mistake. After the correction, the new sentence is:
l. 565: 'The 2.5 m attenuation is slightly higher at GRIP, leading to a remaining amplitude of 86 %, than at NEEM where the remaining amplitude is 90 % (Fig. 4).'

**section 4.2.3: how much snowfall have you added to the snow thickness in this simulation?**
The total cumulated precipitation was 37 kg/m$^2$ for year 2001 (11 cm of fresh snow, 4 cm i.e.). In average for the period 2000-2010, the annual total of precipitation was 29 kg/m$^2$/year. 2001 is the year with the highest accumulation.

**lines 582-583: this gives 10 cm / year, but above you said that the accumulation rate at DC is 8 cm / year (snow equiv.).**
Sorry for this discrepancy. The '**8 cm'** value comes from glaciological analysis, and corresponds to long time scales, whereas the '**10 cm'** value is the one measured for recent years (see Landais et al., 2017). Our forcing data has an accumulation of 29 kg/year, corresponding to 9.6 cm of fresh snow, and therefore coherent with measured accumulation for the last 10 years.
We will modify the text to remove this ambiguity.
l. 439: 'About **10 cm of fresh snow** are deposited every year (Genthon et al., 2016; Landais et al.,

2017), which implies that in order to keep seasonal information, at least one point every 4 cm is required in the first meter…'

**Figure 2d is a bit misleading. From the first glance a reader may think that the seasonal amplitude is increasing with time. Then, it becomes clear that it is actually δ$^{18}$O change that is increasing with time. It would be nicer to show here the δ$^{18}$O values themselves (instead of δ$^{18}$O changes), so that the colors would nicely illustrate the fading isotopic variability. The same comment is for Figures 3 and 12.**

We are aware that our figure is not easy to understand at first glance and we apologize for that.

We hesitated between this 'difference' figure and the original one with the true values of δ$^{18}$O (see below). However, the attenuation is not much easier to see in the original figure (shown below).

[Figure]

An attenuation by 0.5‰ over a half-amplitude of 8 ‰ is barely visible for the maxima (dark red becoming lighter) and no at all for the minima (shades of blue difficult to distinguish).

We therefore prefer to keep the 'difference' figure in the manuscript. Moreover, since the caption clearly states that we are plotting the deviation to the original profile, we do not see our figure as misleading.

**line 1055: December 2001.**
OK. Thanks.

**Interactive comment on Geosci. Model Dev. Discuss., https://doi.org/10.5194/gmd-2017-217, 2017.**

Firn isotope diffusion is a process that affects the $\delta^{18}O$ signal of polar snow from the time of deposition until pore close–off. Taking place in the vapor phase within the porous medium of the firn and driven by the apparent seasonal, annual and multiannual isotopic gradients it results in an attenuation of the $\delta^{18}O$ signal, often obliterating its annual component.

Assuming a good estimate of the diffusive rates in firn is obtained, a "reverse calculation" of diffusion can be possible that allows the (almost) complete reconstruction of the initial signal. Additionally, knowledge of the diffusive rates offers valuable information on past firn temperatures and as a result can be used as a paleothermometry tool if ice core data of sufficient resolution and precision are available.

Previous studies have looked into the desciption and characterisation of these effects and part of these studies suggests that post depositional processes different to purely fickian diffusion of water isotopes can also be at play acting supplementary to the signal attenuation affects or even introducing biases (Town et al., 2008). These processes are mostly of advective nature caused by the bulk movement of air and vapor in the snow, driven by pressure and temperature variations.

In this work titled "Numerical Experiments on isotopic diffusion in polar snow and firn using a multi-layer balance model", Touzeau et al attempt to build and test a water isotope module on top of the Crocus snowpack model. In particular, the authors focus on trying to simulate post-depositional effects that cause changes of the initial $\delta^{18}O$ signal in polar snow and firn. Processes related to snow/firn isotope diffusion as well as diffusive vapor transport due to temperature gradients in the firn are modelled assuming various scenarios. The study focuses on two different regimes that are representative of conditions typical for deep ice coring sites on Greenland and East Antarctica. Ice core data sets are also used in order to evaluate the performance of the model and the results are also compared to existing firn isotope diffusion modelling approaches. This is a very welcome contribution and it most certainly points to the correct direction with respect to future modelling efforts. The study also fits very well the description and scope of the GMD journal and the overall quality of the research conducted is of high level. Thus I would recommend it for publication in GMD after the following points are carefully considered by the authors.

**2 General comments**

1. Unfortunately the language of the manuscript requires a significant revision. In particular there are examples in the text where technical/physical terms are used wrongly and many definitions appear to be loose. This is particularly problematic for a manuscript of this type, where

**modelling approaches and physical processes are described. The most notable example is the description of the transport mechanisms in snow in sections 2.1.1 and 2.1.2. Diffusion is a very well defined process and unfortunately the term is used falsely several times in sections 2.1.1 and 2.1.2 (and elsewhere in the manuscript). After reading these two sections I feel confused about the meaning of many of the terms used here and as a result about the kind of methods followed and the assumptions made in this study.**

In this paper, we distinguish 3 types of diffusion:

-solid diffusion (limited effect in the first meters of snow, because it is very slow);

-diffusion in vapor phase caused by isotopic gradients (gradients present originally in the solid phase, but transmitted to the vapor);

-diffusion in vapor phase caused by temperature gradients (which produce vapor pressure gradients in the porosity).

The first two processes are not **oriented** (=externally forced), they result from random movement of molecules in vapor phase or of ions in solid phase. The last process is **oriented** (=externally forced) as it is forced by an external variable which is the atmospheric temperature. While the two first types of diffusion can only attenuate the original signal, the last type can add 'noise' to the original signal. When diffusion due to temperature gradients is active, original information is not only damped, but even replaced. Thus contrary to the first two processes, it will not be possible to 'reverse' the phenomenon in that case.

Thus we class the 'solid diffusion' and 'diffusion caused by isotopic gradients' within the category of 'processes' leading **only to signal smoothing** in an homogeneous way (Section 2.1.1). And we class 'diffusion caused by temperature gradients' as a process leading to **oriented vapor transport** (Section 2.1.2.), and therefore much more difficult to deconvolute. In this Section 2.1.2., we present other processes also leading to **oriented vapor transport** such as convection and ventilation. Note that we have not included convective processes (convection, and advection due to the vertical gradient of air pressure) in the model yet. Only diffusive processes are present.

The processes we describe are the same as described by others, and the only novelty here is in the way we split the processes. The classical way to do it is to separate convective processes (ventilation and convection) from diffusive processes (the three types of diffusion described). Here, we prefer to split them based on their effect on the original isotopic signal. Is it possible to deconvolute the signal stored in ice cores, because only smoothing is active? Or is the isotopic composition modified more strongly, in particular through the accumulation of heavy isotopes in a specific layer?

We are aware that this splitting may surprise people, as it is not based on the physics of the process, but on its effects. However, it has interest for people who are working at the interface between physical description of processes and interpretation of 'noisy' geochemical data. Moreover, we did not invent this splitting. It was first proposed by Ekaykin et al. in 2009.

In order to help the reader to follow our line of thinking, we have modified the introduction to this section 2.1:

l. 117: 'Several studies address the evolution of the isotopic compositions in the snow column after deposition. **Here we describe first processes leading only to attenuation of the original amplitude (Sect. 2.1.1.). Then we describe processes which lead to other types of signal modifications (Sect. 2.1.2.). These modifications include transporting and accumulating isotopes in some layers without**

**consideration of the original isotopic signal. They also comprise processes taking isotopes away from the snow, and therefore shifting the mean δ¹⁸O value of the snow deposited.'**

Also in relation with a comment made by reviewer 1, we have replaced the term 'oriented processes', which was too vague, by the more precise expression '**processes of oriented vapor transport**' all other the section.

And we define these 'processes of oriented vapor transport' at the beginning of Section 2.1.2:

l. 141: 'We consider here the **oriented movement** of water molecules forced by external variables such as temperature or pressure. **We use the term 'oriented' here to describe an overall movement of water molecules that is different from their molecular agitation, and externally forced**."

**What is an "oriented process" for example? In page7line73 the sentence "We focus on the impact of oriented vapor transport caused by vapor density gradients in the snow..." is very untechnical and unfortunately creates a lot of confusion about what the authors have done.**

'Oriented process': See above.

Line 73: The sentence was modified based on this comment as well as on the first reviewer comment.

l. 78: '**We focus on the movement of water isotopes in the vapor phase in the porosity, in the absence of macroscopic air movement. In that situation, the movement of vapor molecules in the porosity is caused** by **vapor pressure gradients, or by** diffusion **along** isotopic **gradients. Note that in the first case, the** vapor transport is **'thermally induced'** i.e. the **vapor pressure gradients** directly result from temperature gradients within the snowpack.'

Here we focus on two out of the three diffusion processes presented above:

a) the diffusion in vapor phase *caused by isotopic gradients*; b) and the diffusion in vapor phase *caused by temperature gradients*.

**If the term "vapor density" indeed refers to "vapor (molar?) concentration" as I assume then the process described here is a vapor diffusion process.**

It is not a molar concentration, but a **massic concentration (its unit is kg/m³)**. And yes, the concentration gradients drive the diffusion in our model (Fick's law).

We agree that this term leads to confusion. The first reviewer suggested to replace this term by 'vapor pressure' which is more commonly used. However, this is not coherent with our unit, and therefore not applicable for when we are describing the symbols in equations. Using the term 'concentration' is probably a better option in these cases.

Therefore, we have replaced 'vapor density gradients' by 'vapor concentration gradients' as much as possible in the paper. For equations, we have also used the term 'mass concentration' which is more accurate.

**After having read the text several times and tried to infer what the authors try to describe in sections 2.1.1 and 2.1.2 I conclude that they split the processes under consideration in two kinds.**

Thanks for trying to understand this section, despite our particular way of splitting processes. We apologize for not being clear enough. We hope that after adjustments, this section of the manuscript will be easier to follow for the reader.

o **The first, what they call "signal attenuation on a vertical profile", is a the combination of two processes, (a) solid isotope diffusion and (b) firn isotope diffusion in the vapor phase.**

**(a)     The first is extremely slow and can easily be neglected in this study. I find it important that the authors point out in the text that solid diffusion affects all isotopes equally.**

We have added a sentence in the text as a reminder:

l. 138: '**Note that in the solid phase, all isotopes have the same diffusion coefficient**.'

**(b)     The second is a diffusive process taking place in the porous medium of the firn driven by the isotopic gradients.**

**Both processes introduced here follow the same physical principle ie transport of mass due to concentration gradients of a substance. The transport occurs along (or down) the concentration gradients and not "against" as often described in the text.**

As suggested, we have replaced 'against' by 'along' in the manuscript.

o **The second category of processes outlined in section 2.1.2 and termed as "oriented processes". My interpretation of the text is that this type processes are "bulk motion" processes either due to pressure or temperature gradients.**

The reviewer is right. The processes described in this second section indeed correspond to an overall movement of vapor molecules resulting from temperature or pressure gradients. Thus, they are not limited to ventilation and convection.

**The first case is a typical example of advective transport.**

NB: The first case (l. 145) was wind-pumping, which we agree to be convective by nature.

**The second is a bit more complicated however the term diffusion used by the authors is incomplete. Temperature gradients in the snow will eventually cause vapor concentration gradients. The latter, will drive a diffusion process for the vapor as a whole. However this cannot be seen as an isotope diffusion process due to the fact that the diffusive transport of vapor has nothing to do with isotopic gradients. Eventually of course the diffusive transport of water vapor will very likely bring vapor molecules in layers of the snow with different isotopic composition where subsequently an isotope diffusion process will occur locally.**

NB2: The second case (l. 147) was thermally induced vapor diffusion.

There has never been any confusion in our mind between diffusion driven by isotopic gradients and diffusion driven by temperature gradients. Indeed, we have separated strictly diffusion in vapor phase caused **by isotopic gradients** in the first section, from **thermal diffusion of vapor** in the second section. We agree with the reviewer that the second one affects isotopes only indirectly.

Although thermal induced diffusion is certainly a diffusion process, it is indeed a shortcut to call it an 'isotope diffusion' process, and a better expression would be 'vapor diffusion process **with**

consequences for isotopes'.

We will modify instances in the text where 'isotope diffusion' or 'isotopic diffusion' were appearing. We will clarify every time if we were talking about *diffusion along isotopic gradients* or about *thermally induced diffusion of vapor... and its consequences on isotopes*.

L. 1: 'Numerical experiments on **vapor diffusion** in polar snow and firn **and its impact on isotopes** using a multi-layer energy balance model'

l. 20: 'We also run complete simulations of **vapor diffusion along isotopic gradients** and **of vapor diffusion driven by temperature gradients** at GRIP, Greenland and at Dome C, Antarctica over'

l. 363: '… the total flux of vapor is null. But **diffusion along isotopic gradients still occurs if the isotopic gradients are non-zero**(Eq. (13)).'

l. 726: 'The main process implemented here to explain post-deposition isotopic variations is diffusion. We have implemented **two types of diffusion in vapor phase: 1) water vapor diffusion along isotopic gradients, and 2) thermally induced vapor diffusion. The vapor diffusion** between layers **was realized** at the centimetric scale. **The consequences of the two vapor diffusion processes on isotopes in the solid phase were investigated. The solid phase was modelled as snow grains divided in two sub-compartments: a grain surface sub-compartment in** equilibrium with interstitial water vapor **and an** inner grain only exchanging slowly with the surface compartment. We parameterized the'

**This was only an example of how the loose use of technical terms and faulty language creates unnecessary confusion to the reader already from the introduction, leading possibly to confusion and misunderstandings of the methods and principles used in this study. I find it essential that the authors look into the manuscript carefully and revise the text accordingly. In the "Specific Comments" section I include more of these examples as they appear in the text.**

We have made our best to modify the terms in order to make the sentences and our meaning clearer. We will check the use of the term 'diffusion' and remove the term 'isotope diffusion' as stated above.

**2. There is an unclear situation regarding the vapor diffusivity parametrization and value used in this study. It is not exactly clear if there is a temperature dependency of the effective diffusivity Def f to temperature. Based on equation 5 in the manuscript and the comment on the value of Dv I conclude that the value of Def f is taken constant and reflects a temperature of 263 K. If this is indeed the case I would be inclined to question the validity of many of the statements found in the manuscript that concern the comparison of this model with other models of diffusion or results from ice core data. The diffusivity coefficient is heavily dependent on temperature and thus a constant value is an oversimplification for such a study. I would strongly prefer a version of the manuscript where the diffusivity is allowed to depend on temperature. However, if the authors indeed choose to follow the approach of constant diffusivity they will need to stress out very clearly in the manuscript that the comparisons presented here are essentially between different things. This should be even more prominent for the case of the Dome C modelling experiments due to the very large difference between the site temperature and the temperature used for the diffusivity coefficient value (almost 40K).**

Indeed, this has to be addressed, see answer to this question in the first review.

**3.** **Despite my belief that the work performed by the authors is of high quality I need to point out that several elements of the manuscript feel opaque not allowing the reader to judge for herself on the quality of the work and the significance of the results. I find this a fundamental weakness of the manuscript that needs to be addressed. In particular:**

**• The authors claim that the model is evaluated for the top 10 m of snow. However only the top 50-60 cm are presented.**

We do not present results downwards simply because the layers are too thick (~20 cm). Therefore no seasonal pattern is visible in these layers and the attenuation by diffusion is impossible to evaluate.  But we can easily print a window of the whole snowpack for those interested.

**• The authors do not provide any information about neither the ice core data used nor the method used to calculate peak-amplitudes. The latter is not a straight forward procedure and can have a significant influence on the result of such model-data comparisons for diffusion. Information about the depth interval the data originate from, the temperature, accumulation and pressure conditions of the sites as well as the resolution of the data are pieces of information a thorough reader needs to have access to. Present the ice core data.**

We are particularly sensitive to this remark, as we encountered exactly the same problem while looking at previous publications where these ice core data were published and used for diffusion study. While the attenuation (%) was given, we were unable to find the original signal nor the methodology used to calculate the attenuation.

We propose to give in the appendix of the revised manuscript the methodology that we followed to compute the attenuation.

1)    We define 'half-amplitude' as:  abs($\delta^{18}$O-mean($\delta^{18}$O)). Thus, we first compute the mean $\delta^{18}$O in the core, and then for each depth compute the absolute difference to the mean. Following a suggestion by Reviewer 3, we will replace 'half-amplitude' by the more common term 'semi-amplitude' in the manuscript.

2)    We then look for maximas in this series of half-amplitudes. In the first version of the manuscript, we used 20-cm windows at this step. However, this is not well adapted for NEEM. In the revised version we will present results obtained with a 30-cm window for the first 10 meters of the core. Indeed, in this shallow part of the firn the density is about 400 kg m$^3$. Using accumulation rates of 0.23 m i.e. at GRIP, and 0.22 m i.e. at NEEM, the expected length for the cycles is 52 and 55 cm respectively. Since we are looking at half-cycles, a window of 30 cm should allow to get all the maximas present in the record. Deeper in the firn we will use **a window of only 20 cm** coherent with higher snow density downward.

This has been clarified in the text:

l. 558: 'For NEEM the values of the four cores are taken together. For NEEM and GRIP, the **semi-amplitude** is computed along the core. **In the first 10 meters, the maximum value every 30 cm is retained, and deeper in the firn, the maximum value every 20 cm is retained (Fig. 4).** Maximum'

3)    Then over this series of maxima, we keep the maximum value for **each meter of the core** (see Figure below). We use this larger window for the fitting, because we prefer to evaluate attenuation based on the larger (well-defined) maxima. We use the value obtained in the first meter as our '**initial**

**half-amplitude'**. All other maximas are expressed relative to this first meter "maxima of maximas", even if the maximas downward happened to be larger.

We add two sentences in the text to clarify this step:

l. 562: '**Consequently, from this first series of maxima, a second series of maxima is computed, with a larger window of 1 meter. The 'remaining amplitude' is then defined as the ratio between any of these 1-meter maxima and the initial 1-meter maxima**. Maximum semi-amplitudes every 5 m are also computed **and displayed on Figure 4**.'

4)       Lastly, we apply an exponential fit to these values:

[Figure]

• **The initial δ18O profile as well as snapshots of some layers should be plotted. The difference plots with the plethora of colors do not add anything neither for the case of density nor for the case of δ ¹⁸O. The colormaps of these plots are unfortunately very ambiguous to read and despite having the max and min values it looks to me that some of these colormaps are non linear. In combination with the very small difference values for both the density and the δ 18O these color plots leave me guessing. There is very little valuable information I can extract from them.**

The omni-present 2D colored graph is the most common way to present Crocus outputs. The script to make this graph is indeed delivered with the model.

However, we are aware of its limitations, which is why we decided to add **on the side of most figures** a series of 1D profiles, that are often more explicit than the traditional 2D Figure. We plot in particular **the original profile of δ¹⁸O on Figure 2, 3 and 11**. We also plot the deviation to the original **as 1D profiles (depth) on Figure 2 and 3** and as **1D (time) profiles on Figure 11** for a selection of layers.

Although these 1D profiles are often more explicit, we also keep the traditional 2D figure in the article, because people might be interested to see the whole picture and not only what we select.

For instance, we hesitated to insert a 2D figure of temperature evolution when submitting this manuscript. We decided against, as we had already a lot of figures to display. This was an error: because we extracted only summer vertical profiles, the first reviewer asked for profiles in winter which were not apparent. So even if they are difficult to read, the 2D plots remain necessary.

Regarding the color bar, the reviewer is right in thinking that it is non-linear. We have two different (linear) scales for positive and negative change. Positive change is in level of red and negative change in levels of blue. Thus, when negative change is ten times smaller than positive change, it is still visible in the graph. Of course, white color corresponds to zero change. This color convention (red-white-blue, two linear scales both sides of zero) is used everywhere in the paper (except for Figure 11, which will be modified).

l. 453: '**The white color corresponds to an absence of change of the variable**.'

We hope our explanation will make the figure easier to understand.

• **The study considers all three isotopologues of water (δ 18O, δ 17O and δD) however the authors choose to present only the results for δ 18O. Based on (Johnsen et al., 2000) the diffusive attenuation is expected to be stronger for δ 18O compared to δD. Can the model produce this differential signal? This is a very simple test.**

This is a good question. We did not look at the attenuation of δD signal simply because this variable never appear (we use $\delta^{18}O$, dex and $^{17}Oex$). However δD can of course be deduced from the other parameters.

For winter 2000, the slope for δD is -5.64‰/10years. Divided by 8, this would correspond to 0.71‰ attenuation per 10 years in $\delta^{18}O$ which is indeed less than the value of 0.82‰/10years obtained for the $\delta^{18}O$ slope. We thus indeed find that attenuation is larger for $\delta^{18}O$ than for δD.

**4. The discussion about the comparison with GRIP data feels incomplete and not thorough. The actual data set is never shown in the manuscript while there is very little information about how diffusion is estimated for this data set. Measuring peak-to-peak amplitudes on ice core δ 18O data can be very misleading as the initial δ 18O value is unknown and most likely it has been variable through the time.**

We have already answered above about our methods. We are acutely aware of the lack of a reliable method to estimate attenuation from an ice core dataset.

**One technically correct way to estimate diffusion on data is to look into the spectral domain and estimate diffusion length values. Either way the reader had practically no access to information about how diffusion is estimated from the GRIP data.**

Looking at the spectral domain was not possible in our case because the longest simulation was run for 10 years and this is much too short for this approach. For such short record, looking at the spectral domain (and at the diffusion length) would be efficient only if the ice record was perfect (no wind, no stratigraphic noise). **Then (in that case only)** we can expect periodicity and amplitude to decrease jointly in a predictable way.

**Additionally, it should be noted that the GRIP data set, originating from a certain depth interval in the ice core (that is not given in the manuscript) it may have experienced a combination of temperature and acccumulation different from the modern one. Does the comparison presented here take this into account?**

We are presenting a figure (3) that goes **down to 80 meters from the surface**. Thus, the core considered comes from **a depth interval** that **is 0 to 80 meters,** hence directly comparable to modern conditions. We will precise it in the text since it appears to be confusing.

We have added two sentences to clarify this point:

l. 556: published in White et al., 1997. **For the GRIP core, only the first 80 meters are considered. Therefore, the data presented corresponds to deposition and densification conditions similar to the modern ones.**

**In particular if the CROCUS model only uses a fixed diffusivity value for 263 K, there is no doubt that there will be a discrepancy with the data deduced diffusion. These are very important elements of such a study and are notably absent in the manuscript.**

We made a mistake in keeping the diffusivity constant with temperature. However, as indicated above, the direct effect of temperature (molecular agitation) is much less than its indirect effect through vapor concentration and vapor concentration gradients.

When using the formula provided by Johnsen et al. for temperature control on diffusivity, we find that the main difference compared to diffusivity in our simulation **comes not** from the temperature component but from the **atmospheric pressure** component. Indeed, we used air diffusivity at 263 K and 1 atm before and therefore just adding the pressure component (1/0.650) almost double the diffusivity because air is less dense and molecules have more space to move. Again compared to this effect, the temperature effect is limited.

**5. Plots and captions need to be reworked. There are several stylistic inconsistencies that should not be allowed for a publication of this quality. A mixture of different font types, missing measurement units from axes, different approaches in presenting measurement units (using either parentheses or a / sign) and a ‰ sign presented in two ways. I think that many of the captions are too long while in the same time they miss one important piece of information that is the number of the experiment and maybe the ice core site under consideration. I do not think it is the job of a reviewer to go through every single detail and problem with the plots thus I will trust that the authors are certainly able to carefully go through the presented plots and make the necessary changes.**

We will take these remarks into consideration for improving the figures in the revised version of the manuscript.

**6.     Regarding the references given, I think that for the introduction section there is probably an overwhelming number of works cited and a small clean-up is possible. More importantly though, some of the works cited are not peer reviewed belonging to the "Discussions" versions of some of the Copenicus publications journals. I believe that the authors should consider these cases and preferably either omit them or update their references list in case some of the papers in question have reached a post peer-review status.**

This will be taken into account in the revised version.

**3 Specific comments**

**Here some more specific comments for the authors.**

**P2L45 "and then only stacking...". As one looks in higher depths in a core this is less of an issue.**

Yes.

**P3L57 Make sure the reader understands this is vapor-solid exchange in the porous medium of the firn.**

We have added a precision.

**l. 56: '**Second, within the porosity, the vapor isotopic composition can change due to: **1)** diffusion **along** isotopic gradients **in gaseous state**, **2) thermally induced** vapor transport caused by **vapor pressure gradients, 3) ventilation in gaseous state, or 4)** exchanges between the gas phase and the solid **phase i.e.** sublimation and condensation. **In the porosity,** the combination of diffusion **along** isotopic gradients in the vapor and of exchange between vapor and the solid phase has been suggested to be the main explanation to the smoothing of the isotopic signal in the solid phase (Johnsen et al., 2000; Gkinis et al., 2014; **Ebner et al., 2016, 2017**).'

**P3L67 Diffusion length mentioned here but no definition given.**

In Johnsen et al. (1977) the diffusion length is defined as the **mean displacement of a water molecule during its time of presence in the porosity**. More precisely:

$$\left(\Delta L_f\right)^2 = 2\,D\,\tau_v\left(\frac{2}{\pi}\right)^2$$

With D the diffusivity of water molecules, $\tau_v$, the residence time of vapor in the porosity and $\Delta L_f$ the diffusion length.

We have added this definition in the text.

L. 71:  were able to simulate and deconvolute the influence of diffusion along isotopic gradients in the vapor at GRIP and NGRIP using a numerical model**. To do this, they define a quantity named 'diffusion length' which is the mean displacement of a water molecule during its residence time in the porosity. Using a thinning model and an equation of diffusivity of the water isotopes in snow, they compute this diffusion length as a function of depth. It is** then used to compute the attenuation ratio (A/Ao), and in the end retrieve the original amplitude (Ao).

**P3L74 "...and of diffusion against isotopic gradients" Is this vapor firn diffusion or solid?**
We are focusing on vapor diffusion, because solid diffusion plays only a minor role.
We propose the following modification to the manuscript:
l. 78: 'We focus **on the movement of water isotopes in the vapor phase in the porosity, in the absence of macroscopic air movement**.  In that situation, the movement of vapor molecules in the porosity is caused by vapor pressure gradients, or by diffusion along isotopic gradients. Note that in the first case, the vapor transport is 'thermally induced' i.e. the vapor pressure gradients directly result from temperature gradients within the snowpack.'

**P5L117 For an informative plot on the matter see Gkinis et al 2014. Higher accumulation rates also result in increased densification rates and therefore reduced diffusivities.**

Thanks for this complement of information. We will add it to the text:

l.132: 'high accumulation rates ensure a greater separation between seasonal $\delta^{18}O$ peaks (Ekaykin et al., 2009; Johnsen et al., 1977) thereby limiting the impact of diffusion.  **They also result in**

**increased densification rates, and therefore reduced diffusivities (Gkinis et al., 2014).** Because sites with high accumulation'

      **P5L120 Diffusion indeed takes place in the ice column but with rates orders of magnitude lower than that of firn. You want to be more specific about it in the text as you often mix the terms vapor and solid diffusion without being specific about the process taking place in the porous of the firn or in the solid ice.**

      Again, our aim here was to separate "smoothing" processes from "building/shifting" processes. Diffusion **along isotopic gradients**, in the vapor phase **AND** in the solid phase will only lead to smoothing. This is why we do not distinguish between the two processes in this early section of the manuscript: they have the same effect.

      But we are aware that they act at different time scales and at different depths. We are also aware that the solid diffusion is much slower (as indicated in section 2.2.1.).

      We will modify the last sentence to clarify our meaning:

      l. 137: 'Diffusion along isotopic gradients exists throughout the entire snow/ice column. **It occurs mainly in the vapor phase in the firn, especially in the upper layers with larger porosities. After pore closure, it takes place mostly in the solid phase, at a much slower rate.'**

      **P6L143 It would be helpful to add even one sentence where you explain why and how the spherical ice elements approach is too simplistic (is it?).**

      Approximating snow microstructure by a monodisperse collection of spherical ice elements has been carried out in several studies in the past (Legagneux and Domine, 2005, Flanner and Zender, 2006). This makes it possible to perform explicit calculations, for a medium featuring the same surface area/volume ratio, without accounting for the complex microstructure of snow. Several limitations arise, related to the requirement to better account for the full distribution of curvature of the ice/air interface, which is critical for snow metamorphism (Flin and Brzoska, 2008). Furthermore, the ice sphere geometry modifies the distribution of ice chord distances, i.e. the mean ice path which is relevant for ice diffusion. Such effects would better be accounted for using a more comprehensive description of the snow microstructure, although the level of complexity would make it untractable using the current generation of multilayer snowpack models.

      **P7L161 "...the transfer of molecules from the grain boundary towards the center of the grain is very slow" Solid diffusion at the temperatures we are talking about is indeed slow. However this sentence gives a false impression that there is a 1-way motion from boundary to center. This is wrong for two reasons. Firstly, any diffusion process would not result in a 1-way motion of molecules. Secondly and more important, solid diffusion in ice seems to be a self-diffusion process following a vacancy mechanism. This means that there is no isotope effect and diffusion affects all isotopologues equally or in other words molecular transport does not take place along and due to the isotopic gradients in ice (therefore there is no index denoting isotopic species in Eq. 1 - ice diffusivity concerns water molecules in the solid phase regardless of their isotopic composition). As a result the model used here of an isotopically heterogenuous material with internal and external layers does not cause any isotope diffusion in the solid phase due to the radial gradients. In a perfectly homogeneous**

**material you should be expecting the same magnitude of diffusive mixing in the solid phase as in the heterogenuous meterial assumed in the text. It would be good to correct these errors in section 2.2.2 and clarify the precence of the self-diffusion mechanism. The calculations of characteristic times in this section look correct and are relevant though. Just make sure that you clearly explain that this characteristic time concerns not only movement of the different isotopologues along a specific path (surface to center of grain) but of ALL water molecules towards all directions in the grain and across the grains.**

We are sorry for this mistake and will correct the text.

l. 182: '**The grain center isotopic composition may change either as a result of crystal growth/sublimation or as a result of solid diffusion within the grain. For solid diffusion, water molecules move in the crystal lattice through a vacancy mechanism, in a process of self-diffusion that has no particular direction, and that is very slow. The diffusivity of water molecules in solid ice…'**

l. 192: 'Therefore the solid diffusion  **within** the grain, at the time'

**section 3.1.1 I was wondering if it would be possible to outline the components of the Crocus model in a summarising table and shorten this section significantly?**

We feel that, while possible, it is not necessary to shorten this section, given that this manuscript is a model description paper submitted to GMD. It is preferable that the manuscript is a little long, at place, rather than cutting apparently unnecessary details which may hamper the comprehension of some readers. We suggest to keep this part as it is, given that it does not include confusing or misleading information, and rather describes how the isotopic modules are incorporated in the overall Crocus structure.

**P10L221 There does not seem to be any dependence of the densification rate to temperature or accumulation rate. Neither is there a two or three stage densification process as done usually in some other densification models. Can you elaborate on this? Would this model be suitable for modelling the full firn profile from surface to firn–ice transition?**

The parametrization that we use for densification is very simplified, and would not work for a deeper snowpack or the entire firn column. Indeed, we 'compensate' the annual accumulation falling on top of the snow by densifying only the first 10 meters in order to keep the level of the surface steady. And we apply homogeneous compaction, not differentiating between layers with small or large crystal grains or made of resistant hoar.

The original scheme present in Crocus is much better by any regard, and should lead to density predictions much closer to reality. However, it will lead to stable surface level only if the entire snowpack is present within the model. Since we decided to study only the first meters of the snowpack, the original scheme was leading to a permanent increase in the surface level (the compaction below ten meters was absent, as were these layers). We therefore decided to modify the scheme to remove this side-effect.

We will add this precision to the text:

l. 240: '**Layer thickness decreases, and layer density increases** under the burden of the overlying layers and resulting from metamorphism. **In the original module,** snow viscosity is parameterized using the layer density and also using information on the presence of hoar or liquid water.  However, this parameterization of the viscosity was designed for alpine snowpack (Vionnet et al., 2012) and may not be adapted to polar snow packs. **Moreover, since we are considering only the first 12 m of the snowpack in the present simulations, the compaction in the considered layers does not compensate the yearly accumulation, leading to rising snow level with time. To maintain a stable surface level in our simulations,** we used a simplified'

**Eq.4 I think the right term for the quantity ρv should be mass concentration instead of vapor density (this term is wrongly used in more places in the manuscript). Density refers to the ratio of mass to volume of the same substance whereas what you use here is the mass of vapor devided by the volume of air in the open porosity of the layer under consideration. Accordingly I think you should change the symbol from ρv to Cv or similar.**

We have replaced 'vapor density' by '**vapor concentration**' or by '**vapor mass concentration**' everywhere in the manuscript.

**I may be missing something but if I use Fick's first law and a forward difference differenciation scheme I do not get the factor of 2 as in Eq. 4. Can you elaborate please?**

We are looking at the diffusion between the middle of the lower layer and the middle of the upper layer. Therefore, the water molecules travel along a total distance that is dz**low**/2 + dz**up**/2. Half the thickness of the lower layer and half the thickness of the upper layer. So this factor comes from the denominator.

**P11L260 Dv is a function of temperature and pressure. How significant is the fact that you are using a fixed value?**

See above.

**Eq.5 The fact that the diffusivity used here is independent of temperature and site pressure seems problematic to me. Can you comment on this and add a line in the manuscript about the effect of this approach?**

See above.

**Eq. 7 Again strictly speaking the quantity you need here is a concentration and not a density. Change the symbols as well.**

We have modified the text according to the reviewer suggestion. Vapor density was replaced by **vapor concentration** and $\rho_v$ by $C_v$, in the text, equations and tables. We also modified the definition of $c^x_{vap\ ini}$, to stress that it is not the same type of concentration as $C_v$. $C_v$ has a unit (kg m$^{-3}$) whereas $c^x_{vap\ ini}$ is a ratio of mass and has no unit.

l. 1060:

$c^x_{vap\ ini}$ **Ratio between the mass of a given isotopologue in the initial vapor (ˣ is $^{18}$O, $^{17}$O, $^{16}$O, $^{1}$H or D) and the total mass of vapor (no unit). The mass balance is made separately**

**and independently for H and O (*i.e.*: $c^{18}_{vap\ ini} + c^{17}_{vap\ ini} + c^{16}_{vap\ ini} = 1$ and $c^{1H}_{vap\ ini} + c^{D}_{vap\ ini} = 1$).**

**P14L312 Consider using the term rare isotope instead of heavy isotope. Also using an index i is more appropriate than a * sign as later on in Eq. 8 and 9 you use "17", "18" and "D" in the position of the * sign.**

For the isotopes considered here, 'heavy' and 'rare' are interchangeable, without harm, but it is not necessarily the case for other isotopes. We will add a note on this matter in the manuscript.

We will also replace * by i in the equations as suggested by the reviewer.

**Eq. 8 and 9**

The term $C^i_{vap}$ needs to be clarified both here and in Table 1. What I understand is that $C^i_{vap}$ refers to isotope concentration as

$$C^{16}_{vap} = \frac{[H_2^{16}O]}{[H_2^{18}O] + [H_2^{17}O] + [H_2^{16}O]} = \frac{^{16}R}{^{18}R + ^{17}R + ^{16}R} = \frac{1}{^{18}R + ^{17}R + 1} \qquad (1)$$

**(again (Mook , 2000) is a good source for definitions).**

**However later in Eq. 11 you seem to be using the same quantity for something slightly different, this time the masses ratio and not the abundances ratio. Can you comment on that and make sure the definitions are clear to the reader? If needed add a definition equation in Table 1.**

It is true that we make an approximation here. In the traditional equation to define the ratio of two isotopologues, molar concentrations are used. Here we approximate this ratio of molar concentrations by a ratio of masses. We thus neglect the molecular mass term (g mol$^{-1}$).

$$R^{18} = \frac{[H_2^{18}O]}{[H_2^{16}O]} \approx \frac{m^{18}_{vap}}{m^{16}_{vap}} = \frac{c^{18}_{vap\ ini}\ m_{vap}}{c^{16}_{vap\ ini}\ m_{vap}}$$

Our $c^{18}_{vap\ ini}$ is therefore also a mass ratio. It is the mass of the studied isotopologue in the porosity relative to the total mass of vapor in the porosity. We will modify the definition in Table 1 to make this clearer.

$c^{x}_{vap\ ini}$      **Ratio between the mass of a given isotopologue in the initial vapor ($^x$ is $^{18}$O, $^{17}$O, $^{16}$O, $^{1}$H or D) and the total mass of vapor (no unit). The mass balance is made separately and independently for H and O (*i.e.*: $c^{18}_{vap\ ini} + c^{17}_{vap\ ini} + c^{16}_{vap\ ini} = 1$ and $c^{1H}_{vap\ ini} + c^{D}_{vap\ ini} = 1$).**

We will also add a remark in the text about this approximation:

l. 346: 'The equilibrium fractionation coefficients (αsubi) are obtained using the temperature-based parameterization from Ellehoj et al. (2013). **Note that we make a slight approximation here, by replacing molar concentrations by massic concentrations in our mass balance formulas (see Table 1 for symbol definitions).**'

**P14L319 I would be very interested to know why you have used the fractionation factors from (Ellehoj et al., 2013)**

We used them because they are recent.

**P14l329 This note concerns the use of the term kinetic fractionation throughout the whole manuscript. Kinetic effects refer to anything that is non-equilibrium. And indeed fractionation due to the different diffusivity coefficients for the different isotopologues is a type of kinetic fractionation. Though it is an overstatement to claim that you have included all possible kinetic fractionation processes by only using the ratio of the diffusivities. Fractionation effects related to different binding energies of the molecules for example can also be affected by a non-equilibrium/kinetic regime and this is something that is not addressed by the D∗/D term. I would suggest that you go through the manuscript and clarify this (term kinetic is used in pages 1, 13, 14, 16, 30, 42 and 43). I would also refer the authors to the sections 3.1 to 3.5 in vol. 1 of (Mook , 2000). Even though some of this definitions sound trivial I think the manuscript can benefit greatly by getting these small details right, thus avoiding misconceptions.**

We will check the manuscript to make sure that this term is used correctly.

When only the ratio of diffusivity is taken into account, we will use the term 'a' before kinetic fractionation, to underline the fact that it is only one aspect of kinetic fractionation. Alternatively we will precise kinetic fractionation 'during diffusion' or 'during transport', to distinguish this from the kinetic fractionation associated for instance to binding.

**Eq. 12 See previous comment on Eq. 4**

The factor 2 comes from the distance between the center of the two considered layers (see above).

**P15l353 "Here the condensation of excess vapor occurs without additional fractionation". Is this not unphysical. Can you comment?**

Please see above.

**P18l407 Rephrase the sentence. The term "oriented processes" (also used in 2.1.2) is not a technical term. From what I understand your use of the term "oriented processes" refers to advection–based processes that bias the isotopic signal. Diffusion is not such a process, it attenuates the isotopic signal and is driven by isotopic composition gradients as opposed to for example ventilation that is driven by a bulk motion of air in the open porosity. Additionnally diffusion takes place for much longer than 12 m (depending on close–off depth) whereas the extent to which ventilation is apparent in polar firn can be debated.**

We apologize for being unclear.

However, we **do** include diffusion driven by temperature gradients in the '**processes of oriented vapor transport'**, which are **not necessarily** driven by advection. Here, diffusion is driven by the gradients of vapor concentration, and ultimately by the temperature gradients.

As already indicated above, 'oriented vapor transport' means that it is **forced by an external variable** (temperature, pressure) and not resulting from random molecular agitation.

Because temperature gradients are particularly strong in the upper part of the snowpack, and because the porosity is also larger et shallow depths, this **thermally induced diffusion** is mostly effective in the top meters of the snow.

We will amend the sentence to precise that we are talking here about thermally induced diffusion, and not diffusion along isotopic gradients.

l. 435: Typically, **processes of oriented vapor transport** such as **thermally induced diffusion** and ventilation occur mainly in the first meters of snow**. Therefore**, the model starts with an initial snowpack of about 12 m.

**P18l414 "Thus the diffusion process can only be studied in the first 2 m of the model snowpack" Can you elaborate on this? Is it a computation time issue that does not allow for thinner layers below the top 2 m. How do the calculations look like below this depth?**

Yes, we have set an upper limit to the number of layers (100) to limit computational time. However, splitting 12 meters of snow into equal pieces would have led to layers of 12 cm. These thick layers would not have been very useful to quantify attenuation of annual cycles which have shorter wavelength.

The various modules are still active in the next 10 meters, and vapor transport occurs, but the density changes and isotopic changes are much reduced because of the low temperature gradients, larger distances, and larger masses of the layers.

**P19l432 Stick to one name for GRIP/Summit throughout the manuscript.**
See above.

**P19l445 Citing a published work (Bréant et al., 2017) dealing with the density studies at Dome C and GRIP is of course acceptable though the density profile here is of great importance for the diffusion calculations, therefore giving some more information and possibly figures would be appreciated.**
See above.

**Additionnally you give the density as a function of n and t where t (the model time) is an independent variable to z. Can you explain this a little bit better? How do Eq. 16 and 17 give you an evolution of the column density and the densification rates? Please also update the reference to the one past the review process, published in the Climate of the Past**

- In the Crocus model, t and z are orthogonal (as observed on the various 2D graphs in the manuscript). 't' corresponds to time evolution forward, over a few months or years. 'z' corresponds to the depth of a given layer (layer number is n). Of course, the depth 'z' of a given layer n will change with time because of compaction and deposition of new snow layers. Therefore, z depends on both n and t.

- The **initial density profile** is defined as a function of depth, but only at a given time t=0. So, in the equations (17) and (18), the density varies only with depth z, and depth varies only with layer number n, as t is fixed. We will amend the equations to make this clearer:

**The initial density** profile in the snowpack is obtained from fitting density measurements from Greenland and Antarctica (Bréant et al., 2016). Over the first 12 m of snow, we obtain the following evolution (**Eq. (17)** and **Eq. (18)**) for GRIP and Dome C respectively:

$$\rho_{sn}(\boldsymbol{t=0}, n) = 17.2 \cdot z(\boldsymbol{t=0}, n) + 310. \text{ (N=22; R}^2\text{=0.95)}$$ **(17)**

$$\rho_{sn}(\boldsymbol{t=0}, n) = 12.41 \times z(\boldsymbol{t=0}, \boldsymbol{n}) + 311.28 \text{ (N=293; R}^2\text{=0.50)}$$

**(18)**

- Equations (17) and (18) are not used elsewhere in the model (only at model initiation). For t>0, densification occurs based on Equations (3) and (4).

- NB: The NGRIP profile will be replaced by the GRIP profile in the revised version of the manuscript.

- We have updated the reference for Bréant et al., 2017.

**P19l450 Earlier in the manuscript you mentioned that all diffusivity values are for a temperature of 263K. Does the isothermal profile at 241 K affect this and if yes how?**

Please see above.

**P20l457 In Table 2 you refer to a different work for the value of accumulation at GRIP. Be consistent and use only one reference.**

We will keep the value of Dahl-Jensen et al. (1993) which was the original reference.

**P20l457 You can be a bit more specific and call it "peak to peak amplitude".**

OK.

**P20l461 Please refer to general comment nr. 1 with respect to the difference between "isotope firn diffusion" due to isotopic gradients in the snow/firn and signal attenuation/alteration because of air or vapor "bulk motion" driven by pressure or temperature gradients in the snow.**

We have rephrased the sentence, and replaced 'isotopic diffusion' by 'transport of isotopes'. Isotopes are indeed transported in both cases, either through diffusion along isotopic gradients or through the overall movement of water vapor forced by temperature gradients.

l. 502: 'The second simulation is run with evolving temperature in the snowpack (computed by the model, using meteorological forcing from ERA-Interim, see Table 4). In that case, the **transport of isotopes** in the vapor phase results both from **diffusion along isotopic gradients** and from **vapor concentration gradients**. The initial snowpack'

**P20l471 What does the term "densities" refer to here? Vapor densities (use term water vapor concentration instead) or firn densities. If it refers to firn densities can you be more specific about how your densification rates depend on temperature?**

We are talking about the snow densities.

As already indicated above the compaction scheme is very simple here and taken to compensate yearly accumulation. It does not depend at all on the temperature.

However, the temperature and temperature gradients control the intensity of water vapor diffusion in the snowpack. Thus, layers with higher temperature will lose water and therefore density, because their thicknesses will not be modified during water vapor transport. Oppositely, colder layers will receive more water vapor and their density will increase. This is really a result of the vapor transport module, and has nothing to do with compaction. It is not exactly a densification process, since layers can gain or lose mass.

**Section 3.3.3 In my view this section is unnecessary, and its sole sentence can be included in the previous section.**
OK. We moved the sentence to line 475, right after the section title.

**P21l486 Is this peak to peak amplitude?? Also writing that maxima and minima are reduced sounds inaccurate. Attenuation would result in reduced maxima and increased minima, or in the difference between the two being lower. Lack of visual examples makes this type of language errors quite critical as they can be very confusing for the reader.**
The reviewer is right, our sentence was imprecise. What we meant to say was that the amplitude was reduced, with the maximas decreasing and the minimas increasing. This attenuation is visible on panel (b) of the Figure 2.
This is the revised sentence:
l. 527: 'As expected **the peak to peak amplitude of δ¹⁸O cycles is reduced** as a result of diffusion.'

**P21l490 The description of the model in the previous sections suggests that the diffusivity coefficient is independent of temperature. It is not clear though if there is some dependancy of the diffusivity to temperature for your model experiments. One possible cause of the increased depletion for the upper few cm could also be that the firn appears to be quite warmer, something that would result in enhanced diffusion rates for these few cm of the firn column thus attenuating this part more compared to the layers below. I also miss some info on the density profile here and specifically the surface density.**
• The uppermost millimeters of the snowpack are indeed warm, but a little colder than the layers around 2 cm depth. The fact that the temperature is high implies large vapor concentration and therefore effective vapor transport and reinforced attenuation. So, we agree with the reviewer that this elevated temperature facilitates vapor transport and attenuation. The reverse temperature gradient that we describe will act on top of the previous phenomenon, by moving vapor preferentially upwards, and therefore bringing also preferentially light isotopes to the very first layer. Both mechanisms could produce the depletion observed, and therefore, we consider the two explanations valid.
• The initial density profile is the same as in the previous simulation. See text:
l. 446: The initial density profiles are defined for each site specifically (see Sect. 3.2.).
l. 483: **The initial density** profile in the snowpack is obtained from fitting density measurements from Greenland and Antarctica (**Bréant et al., 2017**). Over the first 12 m of snow, we obtain the following evolution (**Eq. (17)** and **Eq. (18)**)) for GRIP and Dome C respectively:

$$\rho_{sn}(t=0,n) = 17.2 \cdot z(t=0,n) + 310.3 \quad (N=22; R^2=0.95)$$
**(17)**

$$\rho_{sn}(t=0,n) = 12.41 \times z(t=0,n) + 311.28 \quad (N=293; R^2=0.50)$$
**(18)**

The first layer has therefore initially a density of 310.3 kg m$^{-3}$ (or very close, depending on its thickness).

[Figure]

Supp. Figure: Evolution of snow density with time (Dome C, **Simulation 6**, D$_{v0}$ varies). Initial density profile linear (black line). New layers deposited with a density of 304 kg m$^{-3}$. They gain mass due to vapor transfer as long as they are exposed. After burial, their density decreases again, as they are subject to alternating vapor fluxes (and the overall density is still close to 304 kg m$^{-3}$).

**Section 4.1.2 This section lacks a proper description of the methods used in order to estimate the amplitude of the isotopic signal for the cores presented. In Johnsen et al. the amplitude of the annual signal is computed using a rather sophisticated modification of the Maximum Entropy Method where the annual signal spectral peak is integrated to give a value in permile. This of course is an estimate dependent on the initial isotopic signature (some years have a greater amplitude thatn others) and for this reason 5m intervals are considered in Johnsen et al. How is this analysis performed here? Can a 20cm interval produce satisfactory results when the layer thickness for these depths at NEEM is in the order of 50-60 cm? Also the term half–amplitude should be peak–amplitude or semi–amplitude.**

The reviewer is right, using a 20-cm interval is not adapted at NEEM. For layers below 10 meters at this site, using this window is ok, because then the cycles have a period smaller than 40 cm, therefore the half-period between a given maxima and the following minima is smaller than 20 cm. For layers shallower than 10 meters a window of 30 cm would be better. We will update our methodology and Figure n°4, to correct this mistake: the window will be changed to 30 cm for layers above 10 meters, at NEEM and at GRIP.

l. 558: 'For NEEM the values of the four cores are taken together. For NEEM and GRIP, the semi-amplitude is computed along the core. **In the first 10 meters, the maximum value every 30 cm is retained, and deeper in the firn, the maximum value every 20 cm is retained (Fig. 4).** Maximum semi-amplitudes every 5 m are also computed.'

Note nonetheless that the 'remaining amplitude' was computed based on maximas obtained with a 1-meter window. Thus, updating the size of the smaller window should not modify our results and conclusions.

We will replace the term half-amplitude by semi-amplitude as suggested.

**P22l513 GRIP is also slightly colder.**
OK. But the temperature difference is quite small.

**P22l520 This is a very good point. Temperature has a strong impact on the diffusivity coefficient. It is certainly relevant to consider various other processes that can be the cause to these discrepancies though a very simple test you could do here is to apply the Johnsen et al diffusivity parametrization in Crocus and compare the results.**
This has been done, following the reviewer suggestion, and the results are presented above and will be integrated into the manuscript. The main impact on the diffusivity is not through the temperature, but through the pressure.

**I am puzzled by the values that are given here for the firn diffusivity. These are much closer to air diffu values. Firn diffusivity values for ρ = 350kgm−3 around the temperature of 241 K are orders of magnitude lower. I attach a plot of the Johnsen et al diffusivity for a range of temperatures. What is the reason for such a large difference? Fig. 1. Diffusivity in firn for O18 at ρ = 350kgm−3**
Unfortunately, we cannot see the Figure 1 mentioned.

However, we suppose that the reviewer is comparing our diffusivity in vapor phase $D_{eff}$ to Johnsen at al.'s *isotopic* diffusivity in the ensemble {vapor + solid}, $\Omega_{fi}$. The values are indeed very different, because the ensemble considered is not the same. It is much easier to move molecules around in the porosity, and homogenize it, than to move molecules around in the crystal lattice and homogenize it. Johnsen's diffusivity integrates not only the diffusivity in vapor phase (which is very close to ours), but also an exchange step between vapor and solid, and evaluates in the end the timing of snow homogenization, not of vapor transport.

**P23l530 It should be mentioned here that the van der Wel (Van der Wel et al., 2015) study is made by spraying a layer of isotopically spiked artificial snow on top of the natural Summit snow. Such experiments are extremely challenging and the approach of using artificial snow can potentially introduce artifacts with respect to the diffusion processes.**
This is right. As stated by the authors themselves, the artificial snow obtained by snow gun may have different diffusivity properties compared to natural snow. However, the spike layer was only 2 cm thick and the diffusion process later continued into the natural snowpack, both downward and upward. So, at some distance from the artificial layer, the diffusion properties are probably back to natural.

The authors discuss this at length in the paper, and conclude that the discrepancy between their data and the Johnsen's et al. model prediction cannot be explained by reduced diffusion in the artificial layer. Indeed, they systematically remove the data from this layer from their computations. And while looking at the time evolution of diffusion length, the discrepancy increases, even if the layers considered are further and further away from the artificial layer region. So, it is not the artificial layer that is responsible for the discrepancy.

We will however add a note in the manuscript regarding this question, as recommended by the reviewer.

l. 594: '**Van der Wel et al. (2015) have compared the model results to a spike-layer experiment realized at Summit. Because an artificial snow layer cannot be representative of natural diffusion, they took care to evaluate diffusion based only on the natural layers present above and below the artificial layer.'**

**P23l548 Perhaps you can slightly rephrase as "..are required to observe significant change in densities due to vapor transport at the seasonal cycle".**

We have modified the text according to the reviewer suggestion.

**P24-25 The numbers of the experiments should be stated for clarity in the subsection titles or very soon after in the main body of each subsection.**

OK.

**P25l585 It is a little bit unclear here why and how the precipitation intermittency results in a biasing of the isotopic signal (from -53.2 ‰ to -49.8 ‰). I can see how the winter precipitation events are biased towards warmer temperatures and more enriched $\delta^{18}O$ values but cannot understand how this creates an additional bias in the isotopic composition of the snow.**

We were imprecise here. What we call (wrongly) initial signal in the precipitation is the expected values in the precipitation if precipitation was falling every day at the same rate. It is based on the temperature data, and not on the actual precipitation amount.

This is what we meant by 'constant precipitation throughout the year', l. 622.

We will correct the sentence to clarify our meaning:

l. 638: '**Based on the atmospheric temperature variations only, the isotopic composition in the precipitation should vary around an average value of -53.2 ‰, with a semi-amplitude of 8.6 ‰. The main reason for this difference is the precipitation amounts: large precipitation events in winter are associated with relatively high d$^{18}$O values. The vertical resolution chosen for the model of 2.5 cm may also contribute to the decrease of the semi-amplitude. Indeed, light snowfall events do not result in the production of a new surface layer, but are integrated into the old surface layer.'**

**P25l590 "As expected the maxima and the minima of $\delta$ 18O are further reduced as a result..." A more precise and careful writing would be very much appreciated. What does this sentence mean? Is this a decrease of the whole $\delta^{18}O$ signal, a decrease in the peak to peak amplitude of the signal or a decrease in only the minimum and the maximum of the signal?**

As above, what we meant was that the amplitude decreases, because minimas are increasing and maximas are decreasing. We will correct the sentence to clarify our meaning.

l. 643: 'As expected, the **peak to peak amplitude of $\delta^{18}O$ variations is** further reduced...'

**Additionally (see also general comments) you can technically not have isotopic diffusion because of temperature gradients. The latter can indeed create water vapor concentration gradients that will result in diffusive transport of all water vapor molecules. This is though not the same process as isotope diffusion.**

Here, both vapor diffusion **caused by temperature gradients** and vapor diffusion **caused by isotopic gradients** are active. While diffusion along isotopic gradients will certainly lead to attenuation, the influence of the second process is more difficult to predict. It is possible that this process contributes to the observed attenuation, at least partially, as is the case for Greenland. The global effect of the two vapor diffusion processes seems to be an attenuation of the original signal.

We will modify the sentence to remove ambiguity here:

l. 643: 'As expected, the **peak to peak amplitude of δ$^{18}$O variations** is further reduced as a result of the **two vapor diffusion processes and of associated vapor/solid exchanges.**'

**P26l600 It is very difficult for the reader to follow the discussion of this paragraph when no access is given to the δ 18O profiles pre and after diffusion. The approach of using contour plots or tracking single layers does not give a good picture of the initial conditions and the evolution of the simulation experiments. Even when those plots are presented they only cover the top 40-50 cm of the studied snow-firn column. As a result, referring to gradients of for example 24 ‰/m feels as an irrelevant piece of information.**

It is not clear to us what should be modified here.

1)	The initial δ$^{18}$O profile at the beginning of the simulation is without interest, since the initial snowpack has an homogeneous δ$^{18}$O value of -40 ‰.

2)	Moreover, vertical profiles of δ$^{18}$O are presented at the beginning of each new year on panel (b). By comparing these profiles together, it is already apparent that the maximas are reduced from one year to the next (and that the minimas are increased). Therefore, we ARE presenting profiles before and after diffusion. We also think that including the panel (d) help to better visualize this attenuation.

3)	Another possibility would be to present profiles that follow a diagonal on the 2D graph. For instance, we could present a diagonal profile for the isotopic composition over the first week after precipitation, for all the layers, to present 'original' isotopic composition, and then a second one, for the layer composition after say 5 years of existence. Is this what the reviewer is aiming at?

4)	Again, presenting what is happening downwards (below 40 cm) has no interest since the layers are isotopically homogeneous (so attenuation is unlikely).

5)	We can indeed convert the gradient into another unit, such as ‰/cm, to be more coherent with our layer thicknesses.

l. 658: 'low vertical gradients of δ$^{18}$O of the order of **0.24 ‰ cm$^{-1}$**, much smaller than the typical δ$^{18}$O gradients at Dome C (**1.10 ‰ cm$^{-1}$**).'

As stated above we will add in the Supplement a 2D plot of the temperature over the same period. This could help the reader to understand the conditions of the simulation.

**P26l605 Indeed lower temperatures will slow down diffusive fluxes. This though can only be modelled if the diffusivity coefficient is temperature dependent something that is not the case for this study. Can you comment on this?**

As explained above, the temperature also acts on diffusion by increasing the amount of water vapor available, and not only through diffusivity.

**P26l607 Which other parameters are loosely estimated? When the term "large uncertainty" is used it is only logical for the reader to ask how large is the uncertainty.**

The two parameters that are loosely estimated are $\tau$ and $\Delta t$ surf/center. We will clarify this in the text.

In this section we explore a range of possible values for both parameters, to evaluate how they affect the results and especially the attenuation. It was an error to state in advance that these parameters bring large uncertainty, because the situation is different for Dome C and GRIP, and because for GRIP the attenuation is increased only by one third. Thus, we will remove this assertion from the text, and simply say that these parameters lead to uncertainty on the simulation result. The rest of the section brings answers on this uncertainty.

l. 662: 'In parallel, **the parameters of the model associated to grain renewal ($\tau$ and $\Delta t_{gsurf/center}$),** could only loosely be estimated leading **to uncertainty** in the attenuation modeling.'

**P28l644 Replace badly with poorly.**
OK.

**P28l645 Being able to implement more processes in a model sounds in principle as a step forward. However I think that a discussion on improving on the knowledge, assumptions and parameters used in the more dominating processes of diffusion is missing here. Integration of more processes that are poorly implemented can be misleading and give the false impression of an improved approach for the description of the problem. With this in mind I think that a comment on proposed improvements, measurements and proper tests with real data would be most welcome in this manuscript especially if it focuses on the more dominating processes of the problem.**

Before testing them, it is difficult to decide which aspect of the problem is dominant. It may indeed be diffusion but we cannot be sure of that, especially since the model simulation falls short of reproducing attenuation observed in the data. Therefore, listing other processes potentially active (and maybe even dominant) seems logical here.

The present article aimed only at presenting the model modifications and its possible applications. But comparisons to real data is of course a necessary follow-up to the present paper. For instance, it would be nice to compare model results to on-site experiments on diffusion, such as the one by van der Wel et al. (2015).

We will add a remark on this in section 4.4, stating that the model should be improved not only by adding more processes but also by better constraining diffusion with real data.

l. 702: 'To **improve the model compatibility with data, two kinds of approaches are possible. On the one hand, it would be useful to realize simulations adapted to on-site experiments such as the one by van der Wel et al. (2015). This would allow to verify how diffusion can be improved in the model. For instance,** previous studies have suggested that water vapor diffusivity within the snow porosity may be underestimated by a factor of 5 (Colbeck, 1983), but this is debated (Calonne et al., 2014). **On the other hand, we also believe that other processes…'**

**P28l660 The top 10 m of snow may have been modelled in this study but results only the top 0.5 m are presented here. Thus I think this sentence should be rephrased in order to reflect the actual results presented in the study.**

We will modify the sentence:

l. 723: 'Water vapor transport and water isotopes have been implemented in the Crocus snow model enabling depicting the temporal $\delta^{18}$O variations in the **top 50 cm** of the snow in response to new precipitation, evolution of temperature gradient in the snow and densification.'

**P29l675 Refer to my general comments on the GRIP case.**

See above.

**4 Comments on figures**

**Figures of experiments results**

**The experiment number should be included in the captions and titles of all relevant figures.**

We added the information in the captions. Our figures do not have a title, only axes titles, which will not be modified.

**Color maps of figures The color maps of the density and $\delta$ $^{18}$O change plots can become more readable if there is also some information about where the zero value is. I assume it is the white but cannot tell with certainty.**

Yes, this is right. We have added a line to clarify this point.

l. 453: 'The variations of the considered variable are displayed as color levels. **The white color corresponds to an absence of change of the variable**. As'

**Density and O18 change plots I find these plots confusing and not intuitive. The meaning of the term "density change" and "$\delta$ 18O change" appears only in the caption of fig. 7 and 8. It is very hard for the reader to understand what this change refers to. My impression until I reached figure 7 and 8 was that these were rates ie change per time. Please clarify in the main text and on the legends of the figures.**

For Figure 2 and 3, the term '$\delta^{18}$O change' is already defined in the caption: it is the deviation to the original profile of $\delta^{18}$O.

For Figure 5: We indicated that the density change was 'cumulative'. This may not be very explicit for the reader. We will complete the explanation with the following sentence:

(in supplement) '**Here, 'density change' stands for the difference between density at t and at the beginning of the simulation for the selected layer.**'

We do the same for $\delta^{18}$O on Figure 6:

l. 1127: '**Here, '$\delta^{18}$O change' stands for the difference between $\delta^{18}$O at t and at the beginning of the simulation for the selected layer.**'

For Figure 7 to 10, the caption already contained the necessary information.

**Figure 11:** The original figure was indeed difficult to apprehend. Here, the initial profile was homogeneous, with $\delta^{18}O$ values of -40 ‰ at all depths. We had decided to use this value as reference and plot '$\delta^{18}O$ change' simply as the simulated value of $\delta^{18}O$ minus -40 ‰. Thus, the values of -18 and -2 present on the colorbar corresponded to $\delta^{18}O$ values of -58 ‰ and -42 ‰.

To make this figure easier to understand, we will remove the term '$\delta^{18}O$ change' and replace it by '$\delta^{18}O$'. We will therefore change the values to -58 ‰ and -42 ‰ on the colorbar.

The caption will be modified accordingly:

l. 1147: '**Simulation 6:** Evolution of $\delta^{18}O_{gcenter}$ values as a result of snowfall and vapor transport over 10 years (compaction is inactive; merging between layers is allowed but limited). (a) Temperature profiles at mid-January for each year. (b) $\delta^{18}O_{gcenter}$ profile at mid-January for each year. (c) **Repartition of $\delta^{18}O_{gcenter}$ values as a function of time and depth.** d) Evolution of $\delta^{18}O_{gcenter}$'

**C20 Figure 11 It is odd that while the slope for the 2000 winter layer is oposite to the other summer layers and you choose to comment on this, the scale of the axis for these data is inverted thus visually "masking" the event. I would really not mind of the lines end up crossing each other if all axis are plotted in the same way.**

OK, we will modify the Figure 11 as suggested.

**concentrations in our mass balance formulas (see Table 1 for symbol definitions).**

The initial **vapor mass concentration in air $C_v$** has already been computed in the vapor transport subroutine, and the volume of the porosity can be obtained from the snow **density $\rho_{sn}$ and the thickness of the layer $dz$.** By combining both, we obtain **Eq. (11)** which gives the initial mass of vapor in the layer $m_{vap\,ini}$.

$$m_{vap\,ini} = \mathbf{C_v} \times \left(1 - \frac{\rho_{sn}}{\rho_{ice}}\right) \times dz$$
(11)

This mass of vapor should be subtracted from the initial grain surface mass because vapor mass is not tracked outside of **the sub-routine (Fig. 1). The new** grain surface isotope composition, after vapor individualization is given by Eq.

**(12)**:

$$c^{18}_{surf\,new} = \frac{m^{18}_{surf\,new}}{m_{surf\,new}} = \frac{m^{18}_{surf\,ini} - m_{vap\,ini} \times c^{18}_{vap\,ini}}{m_{surf\,ini} - m_{vap\,ini}}$$
(12)

The diffusion of isotopes follows the same scheme as the water vapor diffusion described above in Sect. 3.1.2. and Eq.

**(5)**. In Eq. **(13)**, the gradient of **vapor mass concentrations** is replaced by a gradient of concentration of the studied isotopologue. The **kinetic fractionation during the diffusion** is realized with **the $D^i/D$ term where i stands** for $^{18}$O

or $^{17}$O or $^{2}$H (Barkan and Luz, 2007).

$$F^{18}(n+1 \rightarrow n) = \frac{-2 \times D_{eff}(t,n\rightarrow n+1)\left(C_v(t,n) \times c^{18}_{vap\,ini}(t,n) - C_v(t,n+1) \times c^{18}_{vap\,
[revised manuscript text omitted]
 vapor ($^x$ is $^{18}O$, $^{17}O$, $^{16}O$, $^1H$ or D) and the total mass of vapor (no unit). The mass balance is made separately and independently for H and O (i.e.: $c^{18}_{vap\ ini} + c^{17}_{vap\ ini} + c^{16}_{vap\ ini} = 1$ and $c^{1H}_{vap\ ini} + c^D_{vap\ ini} = 1$).** |
| | |
| $\alpha^i_{sub}$ | Fractionation coefficients at equilibrium during sublimation (**i** is either $^{18}O$, $^{17}O$ or D) |
| | Fractionation coefficients during condensation (**i** is either $^{18}O$, $^{17}O$ or D) |
| $\alpha^i_{cond}$ | No fractionation |
| $\alpha^i_{cond\ eff}$ | Effective (total) fractionation |

[revised manuscript text omitted]

***The Greenland snowpack has an initial sinusoidal profile of δ$^{18}$O defined using **Eq. (19):**

$\delta^{18}O = -35.5 - 8 \times \sin\left(\frac{2\pi \times z}{a \times \rho_{ice}/\rho_{sn}}\right)$

****The Dome C snowpack has an initial sinusoidal profile of δ$^{18}$O defined using **Eq. (21):**

$\delta^{18}O = -48.5 - 6.5 \times \sin\left(\frac{2\pi \times z}{a \times \rho_{ice}/\rho_{sn}}\right)$                                      **(21)**

[Figure]

**Figure 1.** Splitting of the snow layer into two compartments, grain center and grain surface, with a constant mass ratio
between them. The vapor compartment is a sub-compartment inside the grain surface compartment and is only defined
at specific steps of the model.

[Figure]

**Figure 2. Simulation 1**: Attenuation of the seasonal $\delta^{18}O_{gcenter}$ variation caused by diffusion **along** isotopic gradients
in vapor phase over 10 years (homogeneous and constant temperature of 241 K, original signal with mean value of -
35.5 ‰ and amplitude of 16 ‰). (a) Vertical homogeneous temperature profile; (b) $\delta^{18}O$ profile at the beginning and
end of the simulation; (c) Deviation of the $\delta^{18}O$ relative to the original profile, for 10 dates; (d) Evolution of the
deviation to the original profile of $\delta^{18}O$.

[Figure]

**Figure 3. Simulation 2**: Attenuation of the seasonal $\delta^{18}O_{gcenter}$ variation caused by diffusion in vapor phase over 10
years (with temperature evolution, original signal with mean value of -35.5 ‰ and amplitude of 16 ‰). (a) Vertical
temperature profile for each summer; (b) $\delta^{18}O_{gcenter}$ profile for each summer; (c) Evolution of the deviation to the
original profile of $\delta^{18}O_{gcenter}$. **Note that temperature evolves during the whole year (see Fig. S1).**

[Figure]

**Figure 4.** Evolution of the $\delta^{18}$O **semi-amplitude** with depth in shallow cores at NEEM, GRIP and NGRIP (Steen-Larsen et al., 2011 and Masson-Delmotte et al., 2015; White et al., 1997; Johnsen et al., 2000). The attenuation of the **semi**-amplitude values with depth was fitted using an exponential equation **(Eq. (22)):**

$$A = A0 \cdot \exp(-\gamma \cdot z) - b \tag{22}$$

With A0=4.976 ‰; $\gamma$=0.08094; b=-1.56 ‰ at GRIP, and A0=4.685 ‰; $\gamma$=0.06622; b=-2.44 ‰ at NEEM.

The dotted curve corresponds to the simulated attenuation at GRIP based on the Johnsen et al.'s model (diffusion length $\sigma$ from their Figure 2, and wavelength $\lambda$ fitted on the Eurocore core from GRIP, White et al., 1997).

[Figure]

**Figure 5. Simulation 3: Evolution of temperature and δ¹⁸O values from January to December 2001. (a)**
**Temperature profiles for the first day of each month; (b) Temperature evolution in the snowpack; (c) 'δ¹⁸O**
**change' in the grain surface compartment; (d) 'δ¹⁸O change' in the grain center compartment. Here, 'δ¹⁸O**
**change' stands for the difference between δ¹⁸O at t and at the beginning of the simulation for the selected layer.**

[Figure]

**Figure 6. Simulation 4:** Cumulative change in $\delta^{18}O_{gcenter}$ values (vapor transport, compaction and wind drift active).

[Figure]

**Figure 7. Simulation 5:** Cumulative change of $\delta^{18}O$ values at the grain center (relative to t0) over 6 months.
Simulation with snowfall with varying $\delta^{18}O$ (function of $T_{air}$), vapor transport active, wind and weight compaction
active.

[Figure]

**Figure 8. Simulation 6**: Evolution of $\delta^{18}O_{gcenter}$ values as a result of snowfall and vapor transport over 10 years (compaction is inactive; merging between layers is allowed but limited). (a) Temperature profiles at mid-January for each year. (b) $\delta^{18}O_{gcenter}$ profile at mid-January for each year. (c) Repartition of $\delta^{18}O_{gcenter}$ values as a function of time and depth. (d) Evolution of $\delta^{18}O_{gcenter}$ values after burial for 4 selected layers (deposited in winter 2000, and summer 2002, 2004, 2006). Note that we do not present the evolution of snow composition in the first year after deposition because the thin snow layers resulting from precipitation are getting merged.

[Figure]

**Figure 9.** Test of the sensitivity of the model to the ratio of mass between **grain surface compartments and total grain** and to the interval of mixing between the two compartments (GRIP).

[Figure]

**Figure 10.** Test of the sensitivity of the model to the ratio of mass between surface and grain center compartments
and to the interval of mixing between the two compartments (Dome C).

**Numerical experiments on vapor diffusion in polar snow and firn and its impact on isotopes using the multi-layer energy balance model Crocus in SURFEX V8.0**

Alexandra Touzeau[1], Amaëlle Landais[1], Samuel Morin[2], Laurent Arnaud[3], Ghislain Picard[3]

[1]LSCE, CNRS UMR8212, UVSQ, Université Paris-Saclay, Gif-sur-Yvette, 91191, France
[2]Météo-France - CNRS, CNRM UMR3589, Centre d'Etudes de la Neige, Grenoble, France
[3]IGE, CNRS UMR5183, Université Grenoble Alpes, Grenoble, France

*Correspondence to*: Alexandra Touzeau (alexandra.touzeau@uib.no)

**Abstract.**

To evaluate the impact of vapor diffusion onto isotopic composition variations in the snow pits and then in ice cores, we introduced water isotopes in the detailed snowpack model Crocus.  At each step and for each snow layer, 1) the initial isotopic composition of vapor is taken at equilibrium with solid phase, 2) a kinetic fractionation is applied during transport, and 3) vapor is condensed or snow is sublimated to compensate deviation to vapor pressure at saturation.

We study the different effects of temperature gradient, compaction, wind compaction and precipitation on the final vertical isotopic profiles. We also run complete simulations of vapor diffusion along isotopic gradients and of vapor diffusion driven by temperature gradients at GRIP, Greenland and at Dome C, Antarctica over periods of 1 or 10 years. The vapor diffusion tends to smooth the original seasonal signal, with an attenuation of 7 % to 12 % of the original signal over 10 years at GRIP. This is smaller than the observed attenuation in ice cores, indicating that the model attenuation due to diffusion is underestimated or that other processes, such as ventilation, influence attenuation. At Dome C, the attenuation is stronger (18 %), probably because of the lower accumulation and stronger $\delta^{18}O$ gradients.

**1 Introduction**

The isotopic ratios of oxygen or deuterium  measured in ice cores have been used for a long time to reconstruct the evolution of temperature over the Quaternary (EPICA

comm. members, 2004; Johnsen et al., 1995; Jones et al., 2018; Jouzel et al., 2007; Kawamura et al.,

2007; Lorius et al., 1985; Petit et al., 1999; Schneider et al., 2006; Stenni et al., 2004; Stenni et al.,

2011; Uemura et al., 2012; WAIS-Divide members, 2013). They are however subject to alteration during post-deposition through various processes. Consequently, even if the link between temperature and isotopic composition of the precipitations is quantitatively determined from measurements and modelling studies (Stenni et al., 2016; Goursaud et al., 2017), it cannot faithfully be applied to reconstruction of past temperature. Nevertheless, ice cores remain a primary climatic archive for the Southern Hemisphere where continental archives are rare (Mann and Jones, 2003). In Antarctica, where meteorological records only started in the 1950s (Genthon et al.,

2013), they provide useful information for understanding climate variability (e.g. EPICA comm. members, 2006;

Shaheen et al., 2013; Steig, 2006; Stenni et al., 2011

) and recent climate change (e.g. Altnau et al., 2015;

Schneider et al., 2006).

When using ice cores for past climate reconstruction, other parameters than temperature at condensation influence the isotopic compositions and must be considered. Humidity and temperature in the region of evaporation (Landais et al.,

2008; Masson-Delmotte et al., 2011; Vimeux et al., 2002), or the seasonality of precipitation (Delmotte et al., 2000;

Sime et al., 2008; Laepple et al., 2011) should be taken into account. In addition, uneven accumulation in time and space introduces randomness in the core stratigraphy ('stratigraphic noise', noise (Ekaykin et al., 2009). Indeed, records from adjacent snow pits have been shown to be markedly different, under the influence of decameter-scale local effects (such as wind redeposition of snow, erosion, compaction, and metamorphism) (Casado et al., 2016b; (Ekaykin et al., 2014; Ekaykin et al., 2002; Petit et al., 1982). These local effects reduce the signal /noise ratio, and then. Then only stacking a series of records from snow pits can eliminate this local variability and yield information relevant to recent climate variations (Altnau et al., 2015; Ekaykin et al., 2014; Ekaykin et al., 2002; Fisher and

Koerner, 1994; Hoshina et al., 2014; Ekaykin et al., 2014).; Altnau et al., 2015). This concern is particularly significant in central regions of east Antarctica characterized by very low accumulation rates (<lower than 100 mm w.e.water equivalent per year, (van de Berg et al., 2006) and). There, strong winds which can scour and erode snow layer over depths larger than the annual accumulation (Frezzotti et al., 2005; Libois et al., 2014; Morse et al., 1999; Magand et al., 2004; Epstein and Sharp, 1965; Town et al., 2008, Picard et al., 2016Libois et al., 2014). There is thus a strong need to study post-deposition effects in these cold and dry regions.

Additionally to mechanical shufflingreworking of the snow, the isotopic compositions are further modified in the snowpack. First, diffusion againstalong isotopic gradients can occur within the ice microstructure atsnow grains due to solid state ('solid diffusion', diffusion (Ramseier et al., 1967). Second, within the porosity, the vapor isotopic composition can change due to: 1) diffusion againstalong isotopic gradients (in gaseous state), oriented, 2) thermally induced vapor transport caused by vapor densitypressure gradients, 3) ventilation (also 
[revised manuscript text omitted]

Note that the Crocus model has a typical internal time step of 900 s (15 min), corresponding to the update frequency of layers properties. We only refer here to processes occurring in dry snow.

1)    Snow fall: The presence/absence of precipitation at a given time is determined from the atmospheric forcing inputs. When there is precipitation, a new layer of snow may be formed . Its thickness is deduced from the precipitation amount.

2)    Update of snow layering: At each step, the model may split one layer into two or merge two layers together to get closer to a target vertical profile for optimal calculations . This target profile has high resolution in the first layers to correctly simulate heat and matter exchanges. The layers that are merged together are the closest in terms of microstructure variables

.

3)    Metamorphism: The microstructure variables evolution follows empirical laws. These laws describe the change of grain parameters as a function of temperature, temperature gradient, snow density and liquid water content.

4)    Snow compaction:

Layer thickness decreases, and layer density increases under the burden of the overlying layers and resulting from metamorphism. In the original module, snow viscosity is parameterized using the layer density and also using information on the presence of hoar or liquid water.  However, this parameterization of the viscosity was designed for alpine snowpack (Vionnet et al., 2012) and may not be adapted to polar snow packs.

Moreover, since we are considering only the first 12 m of the snowpack in the present simulations, the compaction in the considered layers does not compensate the yearly accumulation, leading to rising snow level with time. To maintain a stable surface level in our simulations, we used a simplified compaction scheme, where the compaction rate $\varepsilon$ is the same for all the layers.

The compaction rate is obtained by dividing the accumulation rate at the site (see Sect. 3.3) by the total mass of the snow column (Eq. 3). It is then applied to all layers to obtain the density change per time step using Eq. 4.

$$\frac{1}{\rho_{sn}(t,n)} \times \frac{\rho_{sn}(t+dt,n)-\rho_{sn}(t,n)}{dt} = \frac{1}{m_{sn}(t,n)} \times \frac{dm_{sn}}{dt} \qquad \varepsilon = \frac{dm_{sn}}{dt} / \sum_{1}^{nmax}(\rho_{sn}(t,n) \times dz(t,n))$$

(3)

Note that symbols $\rho_{sn}$, $m_{sn}$ and $t$ are defined in Table 1.

$$\frac{\rho_{sn}(t+dt,n)-\rho_{sn}(t,n)}{dt} = \varepsilon \times \rho_{sn}(t,n) \hspace{5cm} (4)$$

5) Wind drift events: They modify the properties of the snow grains which tend to become more rounded, and. They also increase the density of the first layers through compaction (higher degree of packing of the grains).. An option allows snow to be partially sublimated during these wind drift events (Vionnet et al., 2012). This module was not modified.

6) Snow albedo and transmission of solar radiation: for each layerIn the first 3 cm of snow, snow albedo and absorption coefficient are computed from snow microstructure properties and impurity content (computed based on snow age). The average albedo value in three wavelength bands, using the properties of first 3 cm is used to determine the two uppermost snow layers. Incoming part of incoming solar radiation is partly reflected (using albedo value),at the surface. The rest of the radiation penetrates into the snowpack. Then, for each layer starting from the top, incoming radiation is partly absorbed (using absorption coefficient) and partly transmitted to the layer underneath. This module was not modified. is used to describe the rate of decay of the radiation as it is progressively absorbed by the layers downward, following an exponential law.

7) Latent and sensible surface energy and mass fluxes: The sensible heat flux and the latent heat flux are computed using the aerodynamic resistance and the turbulent exchange coefficients. This module was not modified.

8) Vertical snow temperature profile: It is deduced from the heat diffusion equation, using the snow conductivity, as well as the energy balance at the top (radiation, latent heat, sensible heat) and at the bottom of the snowpack. This module was not modified.

9)     Snow sublimation and condensation at the surface: The amount of snow sublimated/condensed is deduced from the latent heat flux, and the thickness of the first layer is updated. Other properties of the first layer (such as density, and SSA) are kept constant. .

**3.1.2 Implementation of water transfer**

The new vapor transport subroutine  has been inserted after the compaction (4) and wind drift (5) modules, and before the solar radiation module (6). In this section, the term 'interface' is used for the horizontal surface of exchange between two consecutive layers. The flux of vapor at the interface between two layers is obtained using the Fick's law of diffusion (Eq. (4$\underline{5}$)):

$F(n+1 \to n) = \frac{-2\,D_{eff}(t,n \to n+1)(\rho_v(t,n) - \rho_v(t,n+1))}{dz(t,n) + dz(t,n+1)} \quad \frac{-2\,D_{eff}(t,n \to n+1)(C_v(t,n) - C_v(t,n+1))}{dz(t,n) + dz(t,n+1)}$

(4$\underline{5}$)

where $dz(t, n)$ and $dz(t, n+1)$ are the thicknesses of the two layers considered in meters, $C_v(t, n)$ and $C_v(t, n+1)$ are the local vapor mass concentrations in the two layers (in kg m$^{-3}$)$_{,}$ and  $(D_{eff}(t, n \to n+1)$

in m$^2$ s$^{-1})$ is the effective diffusivity of water vapor in the snow at the interface. The thicknesses are known from the previous steps of the Crocus model , but the vapor mass concentrations and the interfacial diffusivities must be computed.

The effective diffusivity at the interface is obtained in two steps: first the effective diffusivities ($D_{eff}(t,n)$ and $D_{eff}(t,n+1)$)

in each layer are calculated (Eq. (5$\underline{6}$)), second, the interfacial diffusivity $(D_{eff}(t, n \to n+1))$ is computed as their harmonic mean (Eq. (6$\underline{7}$)). Effective diffusivity can be expressed as a function of the snow density using the relationship proposed by Calonne et al. (2014), for layers with relatively low density . In these circumstances, the compaction occurs by 'boundary sliding', meaning that the grains slide on each other, but that their shape is not modified. It is therefore applicable to our study where density is always below 600 kg m$^{-3}$. The equation of Calonne et al. (2014) is based on the numerical analysis of 3D tomographic images of different types of snow. It relates normalized effective diffusivity ($D_{eff}$ / $D_v$,  to the snow density $\rho_{sn}$

in the layer (Eq. (6)). $D_v$ is the vapor diffusivity in air and has a value that varies depending on the air pressure and air temperature (Eq. (19) in Johnsen et al.,  $\rho_{ice}$ corresponds to the density of ice ( 917 kg m⁻³

$$\frac{D_{eff}(t,n)}{D_v} = \frac{3}{2}\left(1 - \frac{\rho_{sn}(t,n)}{\rho_{ice}}\right) - \frac{1}{2} \tag{56}$$

$$D_{eff}(t, n \rightarrow n+1) = \frac{1}{\frac{1}{D_{eff}(t,n)} + \frac{1}{D_{eff}(t,n+1)}}$$

(6)

We assume that vapor is in general at saturation in the snow layers (Neumann et al., 2008; Neumann et al., 2009). The local mass concentration of vapor  $C_v$ (in kg m⁻³) in each layer is given by the Clausius-Clapeyron equation (Eq. (7)):

$$\rho_v C_v(t,n) = \rho_{v0} C_{v0}\exp\left(\frac{L_{sub}}{R_v \rho_{ice}}\left(\frac{1}{T_0} - \frac{1}{T(t,n)}\right)\right) \tag{7}$$

where $\rho_{v0}$ $C_{v0}$ is the mass concentration of vapor  at 273.16 K ( 2.173 10⁻³ kg m⁻³, $L_{sub}$ is the latent heat of sublimation ( 2.6 10⁹  m⁻³, $R_v$ is the vapor constant ( 462 , K⁻¹, $\rho_{ice}$ is the density of ice ( 917  m⁻³, $T_0$ is the temperature of the triple point of water ( 273.16 K and T is the temperature of the layer.

[revised manuscript text omitted]

$$\begin{cases} R^{D}_{vap\ ini} = \alpha^{D}_{sub} \times R^{D}_{surf\ ini} \\ R^{D}_{vap\ ini} + 1 = 1/c^{1H}_{vap\ ini} \end{cases} \qquad (910)$$

The equilibrium fractionation coefficients ($\alpha_{sub}^{*}$)($\alpha_{sub}^{i}$) are obtained using the temperature-based parameterization from

Ellehoj et al. (2013). Note that we make a slight approximation here, by replacing molar concentrations by mass concentrations in our mass balance formulas (see Table 1 for symbol definitions).

The initial vapor density $\rho_{vap\ ini}$ mass concentration in air $C_v$ has already been computed in the vapor transport subroutine, and the volume of the porosity can be obtained from the snow density ($\rho_{sn}$)$\rho_{sn}$ and the thickness of the layer ($dz$). By combining both, we obtain Eq. (10 11) which gives the initial mass of vapor in the layer ($m_{vap\ ini}$) $m_{vap\ ini}$:

$m_{vap\ ini} = \rho_{vap\ ini} C_v \times \left(1 - \frac{\rho_{sn}}{\rho_{ice}}\right) \times dz$

(10)

This mass of vapor should be subtracted from the initial grain surface mass (as vapor mass is not tracked outside of the sub-routine, see Fig. 1) so that the new grain surface isotope composition, after vapor individualization is given by Eq. (11):

$c_{surf\ new}^{18} = \frac{m_{surf\ new}^{18}}{m_{surf\ new}} = \frac{m_{surf\ ini}^{18} - m_{vap\ ini} \times c_{vap\ ini}^{18}}{m_{surf\ ini} - m_{vap\ ini}}$ (11)

This mass of vapor should be subtracted from the initial grain surface mass because vapor mass is not tracked outside of the sub-routine (Fig. 1). The new grain surface isotope composition, after vapor individualization is given by Eq.

(12):

$c_{surf\ new}^{18} = \frac{m_{surf\ new}^{18}}{m_{surf\ new}} = \frac{m_{surf\ ini}^{18} - m_{vap\ ini} \times c_{vap\ ini}^{18}}{m_{surf\ ini} - m_{vap\ ini}}$ Refer to Table 1 for the definition of symbols.

_______________________________________________ (12)

The diffusion of isotopes follows the same scheme as the water vapor diffusion described above in Sect. 3.1.2. and Eq.

(4 5). In Eq. (12 13), the gradient of vapor density mass concentrations is replaced by a gradient of concentration of the studied isotopologue. The kinetic fractionation during the diffusion is realized with the $D^{*}/D^{i}/D$ term (where *i stands for $^{18}O$ or $^{17}O$ or $^{2}H$, (Barkan and Luz, 2007).

$F^{18}(n + 1 \rightarrow n) =$

$$\frac{-2\times D_{eff,n\&n+1}\left(\rho_v(t,n)\times c^{18}_{vap\ ini}(t,n)-\rho_v(t,n+1)\times c^{18}_{vap\ ini}(t,n+1)\right)}{dz(t,n)+dz(t,n+1)}\frac{-2\times D_{eff}(t,n\rightarrow n+1)\left(C_v(t,n)\times c^{18}_{vap\ ini}(t,n)-C_v(t,n+1)\times c^{18}_{vap\ ini}(t,n+1)\right)}{dz(t,n)+dz(t,n+1)}\times$$

$$\frac{D^{18}}{D}$$                                        (13)

[revised manuscript text omitted]
) that could counteract this effect in natural conditions and we thus do not focus more on this first layer. In the layers below (first meter of snow) the half attenuation is of 0.76 ‰, corresponding to an attenuation of 1.6 ‰, and to a relative attenuation of 9.5 % of the initial amplitude. This result is very similar to the one obtained in the first, simplified simulation. . Thus, the depletion observed here may not occur in natural settings when these processes are active.

In conclusion, at GRIP, the diffusion of vapor as a result of temperature gradients has only a limiteddouble impact on isotopic compositions, and most of . It increases the simulated attenuation can be attributed to diffusion againstin 
[revised manuscript text omitted]
 (layer thickness ~of 2.5 cm). The values recorded for summer layers (above -45 ‰) reflect may also contribute to the average summer temperatures, whereasdecrease of the values recorded for winter layers (below -50 ‰)

are biased toward warm events (leading to an increase of +6 ‰ of $\delta^{18}O$ compared tosemi-amplitude. Indeed, light snowfall events do not result in the valueproduction of a new surface layer but are integrated into the old surface layer. As expected with constant precipitation throughout the year). , the peak to peak amplitude of $\delta^{18}O$

variations is then further reduced as a result of the two vapor diffusion processes and of associated vapor/solid exchanges. The effect of vapor transport is relatively small. To help its visualization, we selected four layers and displayed the evolution of $\delta^{18}O$ in these layers over the years (Fig. 8d). The selected layers were deposited during winter 2000, and during summer seasons 2002, 2004, and 2006.

As expected, the maxima and minima of $\delta^{18}O$ are further reduced as a result of diffusion driven by temperature gradients. The effect of vapor transport is relatively small so that to help its visualization, we selected three summers (2002, 2004, 2006) and one winter (2000), and followed the evolution of

$\delta^{18}O$ in the layers corresponding to the deposited snow during these seasons (Fig. 11d).

For the layer deposited during winter 2000, there is an increase in $\delta^{18}O$ values of about +0.8 ‰ over ten years.

The slope is irregular, with the strongest increases occurring during summers (Nov.-Feb.), between

November and February, when vapor transport is maximal. The slope is also stronger when the layer is still close to the surface, probably because of the stronger temperature gradients in the first centimeters of snow (Fig. 11a; Sect. 4.3.1.).8a). For the layers deposited during the summers, the evolution of $\delta^{18}O$ values is symmetric to the one observed for winter 2000.

Over 10 years (, i.e. between 2000 and 2009)7, the $\delta^{18}O$ amplitude thus decreases by about 1.6 ‰ (0.8 ‰ for the half-amplitude).‰. This corresponds to a decrease of 1418 % relative to the initial amplitude in the snow layers.

This is higher than the 87 % attenuation (or 9.5 %, modelled in Greenland for constant temperature, and to the 11.7 %

attenuation observed when including diffusion caused by temperature gradients) modelled in Greenland (Sect.

4.1.).). However, the comparison between the two sites is not straightforward, because of differences in temperature and accumulation counteracting each other. On the one hand, at GRIP, the diffusion is forced by low vertical gradients of $\delta^{18}$O of the order of 0.24 ‰ m cm$^{-1}$. These are much smaller than the typical $\delta^{18}$O gradients at Dome C (110 ‰ m$^{-}$

which are close to 1).. 10 ‰ cm$^{-1}$. On the other hand, the temperature of 241 K at GRIP is higher than the 220 K

measured at Dome C (241 K instead of 220 K),. thus favoring diffusion.

**4.3 Sensitivity tests for duration of recrystallization**

We have shown above that attenuation of the isotopic signal seems too small at least for the GRIP site. In parallel, some the parameters $\tau$ and $\Delta t_{gsurf/center}$ of the model, especially for the parameterization of associated to grain renewal, could only loosely be estimated leading to a large uncertainty in the attenuation modeling. In this section, we perform some sensitivity tests to quantify how $\delta^{18}$O attenuation can be increased by exploring the uncertainty range on the renewal of the snow grain. Indeed, the assumed values for the ratio between the mass of grain surface and the total mass of the grain ($\tau$), or may have been under or over-estimated. The same is true for the periodicity of mixing between these two compartments ($\Delta t_{surf/center}$) may have been under or over-estimated. .

The sensitivity tests are first designed for Greenland sites, run for 6 months, with initial amplitude of the sinusoidal

$\delta^{18}$O signal of 16 ‰, and a fixed temperature of 241 K in all the layers (Fig. 12 9). First, we use a periodicity of mixing

$\Delta t_{surf/center}$ of 15 days (more precisely, this mixing occurs on the second and 16$^{th}$ of each month) and vary the value for the mass ratio between the grain surface compartment and the total mass of the grain ($\tau$):: 1·10$^{-6}$,

5·10$^{-4}$, 3.3·10$^{-2}$. Then In practice, for $\Delta t_{surf/center}$=15 days, we realize mixing on the second and 16$^{th}$ of each month.

Second, we use the usual value of 5 10$^{-4}$ for $\tau$, and change the periodicity of the mixing to 2 days.

-        In the first case (, where $\tau$= 1 10$^{-6}$, and the mixing of occurs every 15 days, Fig. 12a), the grain surface compartment is very small, and its. Its original sinusoidal $\delta^{18}$O profile disappears in less than one day due to exchanges with vapor. (not shown). The impact on grain center is then very small (attenuation ofwith an increase of the first minimum by ~1.70 10⁻⁴ ‰ over 6 months (Fig. 9a). In this case, the attenuation due to diffusion is even reduced compared to the results displayed above.

-        In the second case (, where τ= 5 10⁻⁴, and the mixing ofoccurs every 15 days, Fig. 12b), the grain surface compartment is larger, and the attenuation is slower, and. Thus, in the grain surface compartment, half of the original amplitude still remains at the end of the 15 days.simulation (not shown). The impact on the grain center compartment is clearly visible (attenuationwith an increase of the first minimum by of 4.92.2 10⁻² ‰ after 6 months (Fig. 9b).

-        In the third case (, with τ= 3.3 10⁻², and mixing ofevery 15 days, Fig. 12c), the attenuation of the sinusoidal signal in the grain surface compartment is only of 1 % because the grain surface compartment is very large. On opposite, attenuation in the grain center is quite large, i.e. 7.2the first minimum increases by 4.0 10⁻² ‰ after 6 months.

(Fig. 9c).

-        In the fourth case (, with τ= 5 10⁻⁴, and mixing ofevery 2 days, Fig. 12d), the attenuation infirst minimum increases by 4.1 10⁻² ‰ after 6 months for the grain center compartment (Fig. 9d). It is similar to the attenuation observed in the third case (7.5 10⁻² ‰ in 6 months)..

The results of these sensitivity tests suggest that the impact of vapor transfer on the grain center isotopic compositions is maximized when the grain surface compartment is large (τ=3.3 10⁻²) and/or refreshed often (Δt$_{surf/center}$=2 days)..

They also show clearly that using a small grain surface compartment (such as τ= 1 10⁻⁶) drastically reduces the impact on the grain center isotopic values. However, our best estimates for τ and Δt$_{surf/center}$ were not chosen randomly (see

Sect. 3.1.3.), and moreover). Moreover, the use of τ=3.3 10⁻² or Δt$_{surf/center}$=2 days leadleads to an increasea near doubling of the δ¹⁸O attenuation by only a third (see above), which does). This is not bridgeyet 
[revised manuscript text omitted]

Epstein, S. and Sharp, R. P.: Six-year record of oxygen and hydrogen isotope variations in South Pole firn, Journal of

Geophysical Research, 70, 1809-1814, 1965.

Flanner, M. G. and Zender, C. S.: Linking snowpack microphysics and albedo evolution, Journal of Geophysical

Research: Atmospheres, 111, D12208, doi:10.1029/2005JD006834, 2006.

Fréville, H., Brun, E., Picard, G., Tatarinova, N., Arnaud, L., Lanconelli, C., Reijmer, C., and van den Broeke, M.:

Using MODIS land surface temperatures and the Crocus snow model to understand the warm bias of ERA-Interim reanalyses at the surface in Antarctica, The Cryosphere, 8, 1361-1373, 2014.

Frezzotti, M., Pourchet, M., Flora, O., Gandolfi, S., Gay, M., Urbini, S., Vincent, C., Becagli, S., Gragnani, R., and

Proposito, M.: Spatial and temporal variability of snow accumulation in East Antarctica from traverse data, Journal of Glaciology, 51, 113-124, 2005.

Friedman, I., Benson, C., and Gleason, J.: Isotopic changes during snow metamorphism, Stable isotope geochemistry: a tribute to Samuel Epstein, 1991. 211-221, 1991.

Gallet, J.-C., Domine, F., Arnaud, L., Picard, G., and Savarino, J.: Vertical profile of the specific surface area and density of the snow at Dome C and on a transect to Dumont D'Urville, Antarctica-albedo calculations and comparison to remote sensing products, The Cryosphere, 5, 631-649, 2011.

Gay, M., Fily, M., Genthon, C., Frezzotti, M., Oerter, H., and Winther, J.-G.: Snow grain-size measurements in Antarctica, Journal of Glaciology, 48, 527-535, 2002.

Genthon, C., Six, D., Gallée, H., Grigioni, P., and Pellegrini, A.: Two years of atmospheric boundary layer observations on a 45-m tower at Dome C on the Antarctic plateau, Journal of Geophysical Research: Atmospheres, 118, 3218-3232, 2013.

Genthon, C., Six, D., Scarchilli, C., Ciardini, V., and Frezzotti, M.: Meteorological and snow accumulation gradients across Dome C, East Antarctic plateau, International Journal of Climatology, 36, 455-466, 2016.

Gkinis, V., Simonsen, S. B., Buchardt, S. L., White, J. W. C., and Vinther, B. M.: Water isotope diffusion rates from the NorthGRIP ice core for the last 16,000 years – Glaciological and paleoclimatic implications, Earth and Planetary Science Letters, 405, 132-141, 2014.

Guillevic, M., Bazin, L., Landais, A., Kindler, Goursaud, S., Masson-Delmotte, V., Favier, V., Preunkert, S., Fily, M., Gallée, H., Jourdain, B., Legrand, M., Magand, O., Minster, B., Werner, M.: A 60-year ice-core record of regional climate from Adélie land, coastal Antarctica, The Cryosphere, 11, 343-362, 2017.

P., Orsi, A., Masson-Delmotte, V., Blunier, T., Buchardt, S., Capron, E., and Leuenberger, M.: Spatial gradients of temperature, accumulation and $\delta^{18}$O-ice in Greenland over a series of Dansgaard-Oeschger events, Climate of the Past, 9, 1029-1051, 2013.

Hoshina, Y., Fujita, K., Nakazawa, F., Iizuka, Y., Miyake, T., Hirabayashi, M., Kuramoto, T., Fujita, S., and Motoyama, H.: Effect of accumulation rate on water stable isotopes of near-surface snow in inland Antarctica, Journal of Geophysical Research, 119, 274-283, 2014.

Johnsen, S.: Stable isotope homogenization of polar firn and ice, Isotopes and impurities in snow and ice, 1, 210-219,

1977.

Johnsen, S. J., Dahl-Jensen, D., Dansgaard, W., and Gundestrup, N.: Greenland palaeotemperatures derived from GRIP

bore hole temperature and ice core profiles, Tellus, 47B, 624-629, 1995.

Johnsen, S. J., Clausen, H. B., Cuffey, K. M., Hoffmann, G., Schwander, J., and Creyts, T.: Diffusion of stable isotopes in polar firn and ice: the isotope effect in firn diffusion, Physics of ice core records, 159, 121-140, 2000.

Joos, F. and Spahni,Jones, T. R., Roberts, W. H. G., .: Rates of change in natural and anthropogenic radiative forcing over the past 20,000 years, Proceedings of the National Academy of Sciences, 105, 1425-1430, 2008.

Jouzel,Steig, E. J., Masson-Delmotte, V., Stiévenard,Cuffey, K. M., Markle, B. R.,Landais, A., Vimeux, F., Johnsen,

S. J., Sveinbjörnsdottir, A. E., and White, J. W. C.: Rapid deuterium-excess changes in GreenlandSouthern Hemisphere climate variability forced by Northern Hemisphere ice cores: a link between the ocean and the atmosphere, Comptes

Rendus Geoscience, 337, 957-969, 2005-sheet topography, Nature, 554, doi:10.1038/nature24669, 2018.

Jouzel, J., Masson-Delmotte, V., Cattani, O., Dreyfus, G., Falourd, S., Hoffmann, G., Minster, B., Nouet, J., Barnola,

J. M., Chappellaz, J., Fischer, H., Gallet, J. C., Johnsen, S., Leuenberger, M., Loulergue, L., Luethi, D., Oerter, H.,

Parrenin, F., Raisbeck, G., Raynaud, D., Schilt, A., Schwander, J., Selmo, E., Souchez, R., Spahni, R., Stauffer, B.,

Steffensen, J. P., Stenni, B., Stocker, T. F., Tison, J. L., Werner, M., and Wolff, E. W.: Orbital and millennial Antarctic climate variability over the past 800,000 years, Science, 317, 793-796, 2007.

Kaempfer, T. U., Schneebeli, M., and Sokratov, S. A.: A microstructural approach to model heat transfer in snow,

Geophysical Research Letters, 32, L21503, doi:10.1029/2005GL023873, 2005.

Kawamura, K., Parrenin, F., Lisiecki, L., Uemura, R., Vimeux, F., Severinghaus, J. P., Hutterli, M. A., Nakazawa, T.,

Aoki, S., Jouzel, J., Raymo, M. E., Matsumoto, K., Nakata, H., Motoyama, H., Fujita, S., Goto-Azuma, K., Fujii, Y., and Watanabe, O.: Northern Hemisphere forcing of climatic cycles in Antarctica over the past 360,000 years, Nature,

448, 912-916, 2007.

Krol, Q., and Löwe, H.: Relating optical and microwave grain metrics of snow: the relevance of grain shape, The

Cryosphere, 10, 2847-2863, 2016.

Laepple, T., Werner, M., and Lohmann, G.: Synchronicity of Antarctic temperatures and local solar insolation on orbital timescales, Nature, 471, 91, 2011.

Landais, A., Barkan, E., and Luz, B.: Record of $\delta^{18}O$ and $^{17}O$-excess in ice from Vostok Antarctica during the last

150,000 years, Geophysical Research Letters, 35, L02709, 2008.

Landais, A., Casado, C., Prié, F., Magand, O., Arnaud, L., Ekaykin, A., Petit, J.-R., Picard, G., Fily, M., Minster, B., Touzeau, A., Goursaud, S., Masson-Dlemotte, V., Jouzel, J., and Orsi, A.: Surface studies of water isotopes in Antarctica for quantitative interpretation of deep ice core data, Comptes Rendus Géosciences, 349, 139-150, 2017.

Legagneux, L., and Domine, F.: A mean field model of the decrease of the specific surface area of dry snow during isothermal metamorphism, Journal of Geophysical Research, 110, doi:10.1029/2004JF000181, 2005.

Lefebre, F., Gallée, H., van Ypersele, J.-P., and Greuell, W.: Modeling of snow and ice melt at ETH Camp (West Greenland): A study of surface albedo, Journal of Geophysical Research: Atmospheres, 108, D8-4231, doi:10.1029/2001JD001160, 2003.

Libois, Q., Picard, G., Arnaud, L., Morin, S., and Brun, E.: Modeling the impact of snow drift on the decameter-scale variability of snow properties on the Antarctic Plateau, Journal of Geophysical Research: Atmospheres, 119, 11662-11681, 2014.

Libois, Q., Picard, G., Arnaud, L., Dumont, M., Lafaysse, M., Morin, S., and Lefebvre, E.: Summertime evolution of snow specific surface area close to the surface on the Antarctic Plateau, The Cryosphere, 9, 2383-2398, 2015.

Lorius, C., Jouzel, J., Ritz, C., Merlivat, L., Barkov, N. I., Korotkevich, Y. S., and Kotlyakov, V. M.: A 150,000-year climatic record from Antarctic ice, Nature, 316, 591-596, 1985.

Lu, G. and DePaolo, D. J.: Lattice Boltzmann simulation of water isotope fractionation during ice crystal growth in clouds, Geochimica et Cosmochimica Acta, 180, 271-283, 2016.

Magand, O., Frezzotti, M., Pourchet, M., Stenni, B., Genoni, L., and Fily, M.: Climate variability along latitudinal and longitudinal transects in East Antarctica, Annals of Glaciology, 39, 351-358, 2004.

Mann, M. E. and Jones, P. D.: Global surface temperatures over the past two millennia, Geophysical Research Letters, 30, 1820, doi:10.1029/2003GL017814, 2003. Masson-Delmotte, V., Delmotte, M., Morgan, V., Etheridge, D., van Ommen, T., Tartarin, S., and Hoffmann, G.: Recent southern Indian Ocean climate variability inferred from a Law Dome ice core: new insights for the interpretation of coastal Antarctic isotopic records, Clim Dyn, 21, 153-166, 2003.

[revised manuscript text omitted]

van de Berg, W. J., van den Broeke, M. R., Reijmer, C. H., and van Meijgaard, E.: Reassessment of the Antarctic surface mass balance using calibrated output of a regional atmospheric climate model, Journal of Geophysical Research: Atmospheres, 111, D11104, doi:10.1029/2005JD006495, 2006.

van der Wel, L., Been, H., Smeets, P., and Meijer, H.: Constraints on the $\delta^2 H$ diffusion rate in firn from field measurements at Summit, Greenland, The Cryosphere, 9, 1089-1103, 2015.

Vimeux, F., Cuffey, K. M., and Jouzel, J.: New insights into Southern Hemisphere temperature changes from Vostok ice cores using deuterium excess correction, Earth and Planetary Science Letters, 203, 829-843, 2002.

Vionnet, V., Brun, E., Morin, S., Boone, A., Faroux, S., Le Moigne, P., Martin, E., and Willemet, J.: The detailed snowpack scheme Crocus and its implementation in SURFEX v7. 2, Geoscientific Model Development, 5, 773-791,

2012.

Waddington, E. D., Steig, E. J., and Neumann, T. A.: Using characteristic times to assess whether stable isotopes in polar snow can be reversibly deposited, Annals of Glaciology, 35, 118-124, 2002.

WAIS Divide Project Members: Onset of deglacial warming in West Antarctica driven by local orbital forcing, Nature,

500, 440-444, 2013.

White, J.W.C., Barlow, L.K., Fisher, D., Grootes, P., Jouzel, J., Johnsen, S.J., Stuiver, M., and Clausen, H.: The climate signal in the stable isotope of Summit, Greenland snow : Results of comparisons with modern climate observations.

Journal of Geophysical Research, 102, C12, 26425 - 26439, 1997.

**Table 1.** Definition of the symbols used.

| Symbol | Description |
|---|---|
| ***Constants*** | |
| $T_0$ | Temperature of the triple point of water (K) |
| $R_v$ | Vapor constant for water (J·kg$^{-1}$K$^{-1}$) |
| $L_{sub}$ | Latent heat of sublimation of water (J·m$^{-3}$) |
| $C_{v0}$ | Vapor mass concentration at 273.16 K (kg·m$^{-3}$ of air) |
| $D_{ice}$ | Diffusivity of water molecules in solid ice (m$^2$·s$^{-1}$) |
| $D_v$ | Diffusivity of vapor in air at 263 K (m$^2$·s$^{-1}$) (temperature dependency neglected) |
| $\rho_{ice}$ | Density of ice (kg·m$^{-3}$) |
| A | Accumulation (m i.e. per year) |
| $R_{moy}$ | Average snow grain radius (m) |
| $\Delta t_{sol}$ | Characteristic time for solid diffusion (s) |
| $\Delta t_{surf/center}$ | Periodicity of the mixing between grain center and grain surface, because of grain center translation (s) |
| ***1D-variables*** | |
| T | Time (s) |
| N | Layer number from top of the snowpack |
| $\delta^{18}O_{sf}(t)$ | Isotopic composition of oxygen in the snowfall (‰) |
| $T_{air}(t)$ | Temperature of the air at 2 m (K) |
| ***2D-variables*** | |
| $h(t,n)$ | Height of the center of the snow layer relative to the bottom of the snowpack (m) |
| $z(t, n)$ | Depth of the center of the snow layer (m from surface) |
| $dz(t, n)$ | Thickness of the snow layer (m) |
| $T(t, n)$ | Temperature of the snow layer (K) |
| $\rho_{sn}(t, n)$ | Density of the snow layer (kg·m$^{-3}$) |
| $m_{sn}(t, n)$ | Mass of the snow layer (kg) |
| $C_v(t,n)$ | Vapor mass concentration at saturation in the porosity of the snow layer (kg·m$^{-3}$ of air) |
| $D_{eff}(t,n)$ | Effective diffusivity of vapor in the layer (m$^2$·s$^{-1}$) |
| $\delta^{18}O(t, n)$ | Isotopic composition of oxygen in the snow layer (‰) |
| $F^{18}(n+1 \rightarrow n)$ | Flux of the heavy water molecules ($^{18}$O) from layer n+1 to layer n (kg· m$^{-2}$·s$^{-1}$) |
| $F(n+1 \rightarrow n)$ | Vapor flux from layer n+1 to layer n (kg· m$^{-2}$·s$^{-1}$) |
| $D_{eff}(t, n \rightarrow n+1)$ | Effective interfacial diffusivity between layers n and n+1 (m$^2$·s$^{-1}$) |
| $R^i_{vap\ ini}$ | Isotopic ratio in the initial vapor ((i is either $^{18}$O, $^{17}$O or D) |
| $R^i_{surf\ ini}$ | Isotopic ratio in the grain surface sub-compartment before vapor individualization |
| $c^x_{vap\ ini}$ | Ratio between the mass of a given isotopologue in the initial vapor (x is $^{18}$O, $^{17}$O $^{16}$O, $^1$H  or D) and the total mass of vapor (no unit). The mass balance is made separately and independently for H and O (i.e.: $c^{18}_{vap\ ini} + c^{17}_{vap\ ini} + c^{16}_{vap\ ini} = 1$ and $c^{1H}_{vap\ ini} + c^D_{vap\ ini} = 1$). |
| $\alpha^i_{sub}$ | Fractionation coefficients at equilibrium during sublimation ((i is either $^{18}$O, $^{17}$O or D) |

| | |
|---|---|
| $\alpha^i_{cond}$ | Fractionation coefficients during condensation (**i** is either $^{18}$O, $^{17}$O or D)
  No fractionation |
| $\alpha^i_{cond\ eff}$ |   Effective (total) fractionation |
| $\alpha^i_{cond\ kin}$ |   Kinetic fractionation only |
| $cond$ |  |
| $m_{vap\ ini}$ | Initial mass of vapor in the porosity (kg) |
| $m_{surf\ ini}$ | Mass of water in the grain surface sub-compartment before vapor individualization (kg) |
| $m_{surf\ new}$ | Mass of water in the grain surface sub-compartment after vapor individualization (kg) |
|  $T$ | Ratio of between the mass of the grain surface compartment and the mass of total grain |
| $m_{surf}$ | Mass of grain surface compartment |
| $m_{center}$ | Mass of grain center compartment |
| $m_{vap}$ | Mass of vapor in the porosity |
| | |
| $V_{tot}$ | Total volume of the considered layer |
| $\Phi$ | Porosity of the layer |
| | |
|  $m^{18}_{surf\ ini}$ | Mass of heavy water molecules ($^{18}$O) in the grain surface before vapor individualization (kg) |
|  $m^{18}_{surf\ new}$ | Mass of heavy water molecules ($^{18}$O) in the grain surface after vapor individualization (kg) |
| $D^{18}/D$ | Ratio of diffusivities between heavy isotope and light isotope |
| $\Delta m_{vap,exc}$ | Mass of vapor in excess in the porosity after vapor transport (kg) |
| | |
| $\rho_{sn\ ini}$ | Density of the snow layer before vapor transport |
| $\rho_{sn\ new}$ | Density of the snow layer after vapor transport |
| | |
| $T_{ini},\ T_{new}$ | Temperature of the snow layer before and after vapor transport |

GRIP

Accumulation    23 cm i.e./yr                                    Dahl-Jensen et al., 1993

Cellules supprimées

[revised manuscript text omitted]

(see Table 1 for symbols).

| | GRIP sensitivity tests | | | | Dome C sensitivity tests | | | | |
|---|---|---|---|---|---|---|---|---|---|
| N° | 1 | 2 | 3 | 4 | 1 | 2 | 3 | 4 | 5 |
| Section | 4.3. | 4.3. | 4.3. | 4.3. | 4.3. | 4.3. | 4.3. | 4.3. | 4.3. |
| Figures | Figure 9 | Figure 9 | Figure 9 | Figure 9 | Figure 10 | Figure 10 | Figure 10 | Figure 10 | Figure 10 |
| Duration | 6 months | 6 months | 6 months | 6 months | 3 years | 3 years | 3 years | 3 years | 3 years |
| Period | Jan-Jun 2000 | Jan-Jun 2000 | Jan-Jun 2000 | Jan-Jun 2000 | Jan 2000-Dec 2002 | Jan 2000-Dec 2002 | Jan 2000-Dec 2002 | Jan 2001-Dec 2003 | Jan 2001-Dec 2003 |
| *Atmospheric forcing applied* | | | | | | | | | |
| Air T | - | - | - | - | - | - | - | ERA-Interim | ERA-Interim |
| Specific humidity | - | - | - | - | - | - | - | ERA-Interim | ERA-Interim |
| Air pressure | - | - | - | - | - | - | - | ERA-Interim | ERA-Interim |
| Wind velocity | - | - | - | - | - | - | - | ERA-Interim | ERA-Interim |
| Snowfall | NO | NO | NO | NO | NO | NO | NO | NO | NO |
| $\delta^{18}O_{sf}$ | - | - | - | - | - | - | - | - | - |
| *Model configuration* | | | | | | | | | |
| Initial snow T | Flat profile (241 K) | Flat profile (241 K) | Flat profile (241 K) | Flat profile (241 K) | Flat profile (241 K) | Flat profile (220 K) | Flat profile (220 K) | One-year run initialization (Jan-Dec 2000) | One-year run initialization (Jan-Dec 2000) |
| Evolution of snow T | Constant  | Constant  | Constant  | Constant  | Constant  | Constant  | Constant  | Computed  | Computed  |
| Initial snow d18O | Sinusoidal profile*** | Sinusoidal profile*** | Sinusoidal profile*** | Sinusoidal profile*** | Sinusoidal profile**** | Sinusoidal profile**** | Sinusoidal profile**** | Sinusoidal profile**** | Sinusoidal profile**** |

| | | | | | | | | | |
|---|---|---|---|---|---|---|---|---|---|
| Wind drift | NO | NO | NO | NO | NO | NO | NO | NO | NO |
|  Homogeneous compaction | NO | NO | NO | NO | NO | NO | NO | NO | NO |
| Mass ratio τ within the grain | $1\cdot10^{-6}$ | $5\cdot10^{-4}$ | $3.3\cdot10^{-2}$ | $5\cdot10^{-4}$ | $5\cdot10^{-4}$ | $5\cdot10^{-4}$ | $5\cdot10^{-4}$ | $5\cdot10^{-4}$ | $3.3\cdot10^{-2}$ |
| Period for recrystallization Δtsurf/center | 15 days | 15 days | 15 days | 2 days | 15 days | 2 days | 15 days | 15 days | 15 days |

Table 5. List of the sensitivity tests performed at GRIP and at Dome C. The external atmospheric forcing used for Dome C is ERA-Interim reanalysis (see Table 4).

***The Greenland snowpack has an initial sinusoidal profile of $\delta^{18}$O defined using Eq. (19):

$$\delta^{18}O = -35.5 - 8 \times \sin\left(\frac{2\pi \times z}{a \times \rho_{ice}/\rho_{sn}}\right) \quad \text{}$$

****The Dome C snowpack has an initial sinusoidal profile of $\delta^{18}$O defined using Eq. (21):

$$\delta^{18}O = -48.5 - 6.5 \times \sin\left(\frac{2\pi \times z}{a \times \rho_{ice}/\rho_{sn}}\right) \quad \text{}$$

[Figure]

(21)

[Figure]

**Figure 1.** Splitting of the snow layer into two compartments, grain center and grain surface, with a constant mass ratio
between them. The vapor compartment is a sub-compartment inside the grain surface compartment, and is only defined
at specific steps of the model.

[Figure]

[Figure]

**Figure 2. Simulation** **1**: Attenuation of the seasonal δ¹⁸O$_{gcenter}$ variation caused by diffusion along isotopic gradients in vapor phase over 10 years (homogeneous and constant temperature of 241 K, original signal with mean value of -35.5 ‰ and amplitude of 16 ‰). (a) Vertical homogeneous temperature profile; (b) δ¹⁸O profile at the beginning and end of the simulation; (c) Deviation of the δ¹⁸O relative to the original profile, for 10 dates; (d) Evolution of the deviation to the original profile of δ¹⁸O.

[Figure]

**Figure 3. Simulation** 2: Attenuation of the seasonal $\delta^{18}O_{gcenter}$ variation caused by diffusion in vapor phase over 10 years (with temperature evolution, original signal with mean value of -35.5 ‰ and amplitude of 16 ‰). (a) Vertical temperature profile for each summer; (b) $\delta^{18}O_{gcenter}$ profile for each summer; (c)  Evolution of the deviation to the original profile of $\delta^{18}O_{gcenter}$. Note that temperature evolves during the whole year (see Fig. S1).

[Figure]

[Figure]

**Figure 4.** Evolution of the δ¹⁸O semi-amplitude with depth in shallow cores at NEEM, GRIP and NGRIP (Steen-
Larsen et al., 2011 and Masson-Delmotte et al., 2015; White et al., 1997; Johnsen et al., 2000). The attenuation of
the semi-amplitude values with depth was fitted using an exponential equation (Eq. (22)):

$$A = A0 \cdot \exp(-\gamma \cdot z) - b$$  (22)

With A0=4.976 ‰; γ=0.08094; b=-1.56 ‰ at GRIP, and A0=4.685 ‰; γ=0.06622; b=-2.44 ‰ at NEEM.

[Figure]

The dotted curve corresponds to the simulated attenuation at GRIP based on the Johnsen et al.'s model (diffusion
length σ from their Figure 2, and wavelength λ fitted on the Eurocore core from GRIP, White et al., 1997).

[Figure]

Figure 5. Change in snow density caused by vapor transfer over one year (cumulative). (a) Temperature profile on
the first day of each month, around 8 pm. The 1st of August corresponds to a short-term warm event within winter.
(b) Evolution of the snow densities. During summer, the first layer is gaining water whereas the layers immediately
below (11.58 to 11.63 m) are losing water. Thus the vapor departure region is not exactly at the top of the
snowpack in this simulation. Further down, layers are once again gaining water. During winter, temperature
gradients are generally reversed, but the amount of vapor is too small to visibly affect the layer densities (for
instance, the warming of Aug., 1st leads to a density loss of only 0.2 kg/m³ in the first layer).

[Figure]

. Simulation 3: Evolution of temperature and $\delta^{18}$O values from January to December 2001. (a) Temperature profiles
for the first day of each month; (b) Temperature evolution in the snowpack; (c) '$\delta^{18}$O change' in the grain surface
compartment; (d) '$\delta^{18}$O change' in the grain center compartment. Here, '$\delta^{18}$O change' stands for the difference between
$\delta^{18}$O at t and at the beginning of the simulation for the selected layer.

[Figure]

Figure 6. Change of $\delta^{18}$O values caused by vapor transfer from January to December 200. (a) grain
surface; (b) grain center. For both grain compartments: $\delta^{18}$O values change as a result of vapor transport
and therefore maximum change occurs during summer. The first layer that receives water has $\delta^{18}$O values
that decrease during summer, whereas $\delta^{18}$O values increase in the layers immediately below (zone of
water export). Further down (around 11.35 m) the $\delta^{18}$O values decrease (arrival of $^{18}$O-depleted vapor).
During the winter, short-lived warm events (like the one on the 1st of August) lead to small changes in the
$\delta^{18}$O values (the first layer $\delta^{18}$O increases). This is more visible for grain surface than for grain center.

[Figure]

 Figure 7. Evolution of the snow density over one year in a case with homogeneous compaction and wind drift, but
 without precipitation. The density change is taken as the difference relative to the first day for each layer (layer 1 is
 compared to layer 1, layer 5 to layer 5 etc...), even if they are not at the same height; thus density change may be
 overestimated compared to "horizontal" density change).

[Figure]

 Figure 8. Simulation 4: Cumulative change in $\delta^{18}O_{gcenter}$ values (vapor transport, compaction and wind drift active).

[Figure]

[Figure]

**Figure 7.** Snow density change (relative to original density profile at t0) over one year: precipitation
(snowfall) active, compaction (wind and weight) active, vapor transfer active.

[Figure]

**Simulation 5:** Cumulative change of $\delta^{18}$O values at the grain center (relative to t0) over 6 months.
Simulation with snowfall with varying $\delta^{18}$O (function of $T_{air}$), vapor transport active, wind and weight compaction
active.

[Figure]

**Figure 8. Simulation 6**: Evolution of δ[18]O$_{gcenter}$ values as a result of snowfall and vapor transport over 10 years (compaction is inactive; merging between layers is allowed but limited). (a) Temperature profiles at mid-January for each year. (b) δ[18]O$_{gcenter}$ profile at mid-January for each year. (c) Repartition of δ[18]Oδ[18]O$_{gcenter}$ values (expressed relative to -40‰) as a function of time and depth. (d) Evolution of δ[18]O$_{gcenter}$ values after burial for 4 selected layers (deposited in winter 2000, and summer 2002, 2004, 2006). Note that we do not present the evolution of snow composition in the first year after deposition because the thin snow layers resulting from precipitation are getting merged.

[Figure]

[Figure]

**Figure 9.** Test of the sensitivity of the model to the ratio of mass between grain surface and grain center compartments and total grain and to the interval of mixing between the two compartments (GRIP).

[Figure]

[Figure]

**Figure 10.** Test of the sensitivity of the model to the ratio of mass between surface and grain center compartments
and to the interval of mixing between the two compartments (Dome C).